# MGUP: A Momentum-Gradient Alignment Update Policy for Stochastic Optimization

**Da Chang**[134] , **Ganzhao Yuan**[21*]
[1]Shenzhen Institute of Advanced Technology, Chinese Academy of Sciences
[2]Shenzhen University of Advanced Technology, [3]Pengcheng Laboratory
[4]University of Chinese Academy of Sciences

## Abstract

Efficient optimization is essential for training large language models. Although intra-layer selective updates have been explored, a general mechanism that enables fine-grained control while ensuring convergence guarantees is still lacking. To bridge this gap, we propose **MGUP**, a novel mechanism for selective updates. **MGUP** augments standard momentum-based optimizers by applying larger step-sizes to a selected fixed proportion of parameters in each iteration, while applying smaller, non-zero step-sizes to the rest. As a nearly plug-and-play module, **MGUP** seamlessly integrates with optimizers such as AdamW, Lion, and Muon. This yields powerful variants such as **MGUP-AdamW**, **MGUP-Lion**, and **MGUP-Muon**. Under standard assumptions, we provide theoretical convergence guarantees for **MGUP-AdamW** (without weight decay) in stochastic optimization. Extensive experiments across diverse tasks, including MAE pretraining, LLM pretraining, and downstream fine-tuning, demonstrate that our **MGUP**-enhanced optimizers achieve superior or more stable performance compared to their original base optimizers. We offer a principled, versatile, and theoretically grounded strategy for efficient intra-layer selective updates, accelerating and stabilizing the training of large-scale models. The code is publicly available at `https://github.com/MaeChd/MGUP`.

## 1 Introduction

Recent studies reveal that the learning matrix during Large Language Model (LLM) training exhibits low-rank properties, suggesting that learning predominantly occurs in a low-dimensional space [1, 2]. This observation has catalyzed the development of methods such as Galore [3] and LDAdam [4], which use gradient low-rank decomposition to achieve performance comparable to full-rank updates while reducing memory consumption. Although low-rank properties do not directly imply sparsity, the insight that optimization occurs in a low-dimensional space provides a crucial foundation for selective parameter updates. This principle is exemplified by SIFT [5], which achieves efficient adaptation through gradient-based sparse parameter updates, leveraging the low intrinsic dimensionality and sparse gradient characteristics inherent in LLMs. Building on this foundation, several innovative layer-wise selective update methods have emerged, including AutoFreeze [6], LOMO [7], LISA [8], and BAdam [9]. By strategically freezing certain layers while updating others, these methods achieve performance comparable to, or even surpassing, that of full-parameter updates.

While layer-wise selective updates show promise, finer-grained parameter selection remains under-explored. Although SIFT [5] investigates sparse intra-layer updates, a systematic methodology for identifying the most critical parameters within each layer is still lacking. This research gap motivates the development of novel intra-layer sparse update strategies.

---

*Corresponding author: `yuanganzhao@foxmail.com`

39th Conference on Neural Information Processing Systems (NeurIPS 2025).

Recently, Liang et al. [10] have proposed Cautious Optimizers, a novel intra-layer sparse update strategy. This approach selectively updates only parameters where momentum and gradient are aligned (i.e., $\mathbf{m}_t \odot \mathbf{g}_t > 0$), enabling larger updates for aligned directions while skipping misaligned ones. Conceptually, it extends earlier adaptive optimizers like AdaBelief [11], which adjusts step sizes using $(\mathbf{m}_t - \mathbf{g}_t)^2$, but introduces parameter selection based on momentum-gradient alignment.

However, both methods have notable limitations. AdaBelief's update mechanism relies heavily on Adam's second-moment estimation, which restricts its applicability to optimizers that do not compute second moments (e.g., Lion [12] or Muon [13]). Furthermore, Cautious Optimizers lack rigorous theoretical convergence guarantees in the stochastic setting. Although the strategy offers theoretical insights in the deterministic case, its convergence properties remain unverified under stochastic conditions. This raises a crucial question:

> *Within the stochastic optimization setting, can the concept of intra-layer sparsity in updates, based on momentum-gradient direction consistency, truly serve as a plug-and-play mechanism?*

If so, what are the boundaries of its effectiveness? If not, what are the underlying reasons?

We explore this issue in detail in the theoretical analysis presented in Section 4. Specifically, we demonstrate that for Adam variants incorporating a mask, simply setting the update step to zero for parameters where momentum and gradient directions are misaligned significantly impacts the convergence properties of stochastic optimization. This motivates rethinking how to perform selective parameter updates more effectively in stochastic optimization settings to maintain favorable convergence properties. For example, without guided parameter selection, certain extreme cases can occur: *(i)* only a small fraction of parameters receive substantial updates (potentially leading to unstable training), or *(ii)* the updates for the vast majority of parameters are overly suppressed (potentially resulting in slow training). Therefore, we propose that a promising policy involves not only considering the alignment between momentum and gradient direction but also regulating the proportion of parameters receiving substantial versus minor updates, to strike a balance between training efficiency and stability.

Motivated by our theoretical analysis and resulting design considerations, we introduce a novel selective update method: **MGUP** (**M**omentum-**G**radient alignment **U**pdate **P**olicy). **MGUP** updates parameters selectively and differentially by sorting the values of the element-wise product $\mathbf{m}_t \odot \mathbf{g}_t$. Specifically, the top $K$ parameters ranked by $\mathbf{m}_t \odot \mathbf{g}_t$ receive a scaled step size $\alpha \cdot \eta_t$ ($\alpha > 1$), while the rest receive $\gamma \cdot \eta_t$ ($\gamma < 1$), where $\eta_t$ is the base step size from the original optimizer. **MGUP** is inspired by the cautious update strategy, refining it in line with the principles of AdaBelief and Cautious Optimizers by dynamically adjusting update strength based on momentum-gradient alignment.

Our contributions are summarized as follows:

- We develop a novel selective parameter update mechanism, **MGUP**, which assigns larger step sizes to a subset of parameters and smaller ones to the rest. As a plug-and-play mechanism, **MGUP** can be integrated into momentum-based optimizers such as AdamW, Lion, and Muon, yielding variants we refer to as **MGUP-AdamW**, **MGUP-Lion**, and **MGUP-Muon**.

- We establish the convergence of the Adam optimizer with the **MGUP** mechanism in the stochastic setting, providing theoretical guarantees for its reliability.

- We validate the proposed **MGUP** optimizers through key experiments, including MAE pretraining of ViT-27M on CIFAR-10; autoregressive pretraining of LLaMA2-71M and Qwen2.5-150M on Wikitext-103; and fine-tuning of RoBERTa-base on GLUE and LLaMA2-7B for GSM-8K. These results show the robustness and versatility of **MGUP** across diverse models and tasks.

## 2 Related Work

In this section, we review the basic principles of stochastic optimization methods relevant to the momentum-gradient approach. We consider minimizing the objective function as follows:

$$\min_{\mathbf{x} \in \mathbb{R}^d} f(\mathbf{x}), \text{ where } f(\mathbf{x}) = \mathbb{E}_{\xi \sim \mathcal{D}}[f(\mathbf{x}; \xi)]. \tag{1}$$

Here, $f : \mathbb{R}^d \to \mathbb{R}$ is a differentiable and possibly nonconvex function, $\xi$ represents a random vector, such as a training data point, sampled from an unknown data distribution $\mathcal{D}$.

In the context of solving problem (1), momentum-based methods are foundational in large-scale machine learning optimization, accumulating past gradient information to accelerate convergence and navigate complex loss landscapes. The standard momentum update, an exponentially weighted moving average (EWMA) of gradients, is given by

$$\mathbf{m}_t = \beta_1 \mathbf{m}_{t-1} + (1 - \beta_1) \mathbf{g}_t,$$

where $\beta_1$ is the decay factor, $\mathbf{m}_t$ denotes the momentum vector, and $\mathbf{g}_t$ denotes the gradient at the $t$-th iteration. This technique smooths gradient estimates, empirically and theoretically accelerating convergence and enhancing training stability [14, 15, 16, 17].

While standard momentum is a robust baseline, research has sought to improve it, primarily through: *(i)* reducing stochastic gradient estimate variance and *(ii)* adapting learning based on momentum and gradient characteristics.

Variance reduction techniques, such as SPIDER [18], STORM [19], SUPER-ADAM [20], and MARS [21], operate by substituting the original stochastic gradient $\mathbf{g}_t$ with a gradient estimator $\mathbf{g}_t'$ that exhibits lower variance. This refined estimator is then used in the momentum update: $\mathbf{m}_t = \beta \mathbf{m}_{t-1} + (1 - \beta) \mathbf{g}_t'$. While these methods theoretically accelerate convergence, they often necessitate additional computation or storage (e.g., storing past gradients). In contrast, **MGUP** adopts a distinct strategy, focusing on adaptively adjusting the update magnitude based on the characteristics of momentum and the current stochastic gradient, rather than directly altering the variance of the gradient estimation.

Another significant method involves adapting the optimization step based on the perceived reliability or characteristics of the momentum estimate. The intuition guiding this class of methods can be summarized as:

> *Increase step size for trustworthy momentum; Decrease step size for untrustworthy momentum.*

This adaptation is often implemented by modulating the momentum vector, which can be represented generally as:

$$\mathbf{x}_{t+1} = \mathbf{x}_t - \eta_t \mathbf{m}_t \odot \phi_t, \tag{2}$$

where $\phi_t$ is a scaling factor, often applied element-wise, determined by gradient statistics.

Early adaptive methods, like Adagrad [22], introduced per-parameter learning rates by accumulating squared gradients. The widely adopted Adam optimizer [23] builds on this by using EWMAs for both the first moment $\mathbf{m}_t$ and the second moment $\mathbf{v}_t$ of the gradients:

$$\mathbf{v}_t = \beta_2 \mathbf{v}_{t-1} + (1 - \beta_2) \mathbf{g}_t^2.$$

The update step is then element-wise scaled by $1/\sqrt{\hat{\mathbf{v}}_t + \epsilon}$, with $\hat{\mathbf{v}}_t$ being a bias-corrected $\mathbf{v}_t$ and $\epsilon > 0$ is a small constant. This enables Adam to adapt the learning rate per parameter based on historical gradient magnitudes. Subsequent research delved into various scaling factors, frequently investigating the interplay between the current gradient $\mathbf{g}_t$ and the accumulated momentum $\mathbf{m}_t$. The AdaBelief optimizer [11] modifies Adam's second moment by using the squared difference between momentum and the current gradient, $(\mathbf{m}_t - \mathbf{g}_t)^2$, instead of the raw squared gradient $\mathbf{g}_t^2$. The update rule for the second moment $\mathbf{v}_t$ is as follows, with the initial condition $\mathbf{v}_0 = 0$:

$$\mathbf{v}_t = \beta_2 \mathbf{v}_{t-1} + (1 - \beta_2)(\mathbf{m}_t - \mathbf{g}_t)^2 = (1 - \beta_2) \sum_{i=1}^{t} \beta_2^{t-i} (\mathbf{m}_i - \mathbf{g}_i)^2.$$

The term $(\mathbf{m}_t - \mathbf{g}_t)^2$ measures "belief" in the current gradient by its consistency with momentum. Significant deviation increases the corresponding element in $\mathbf{v}_t$, reducing that parameter's effective step size. This mechanism aims to merge Adam's rapid convergence with SGD's generalization. Denote $\mathbf{m}_{t,i}$ and $\mathbf{g}_{t,i}$ as the $i$-th elements of the momentum vector $\mathbf{m}_t$ and gradient vector $\mathbf{g}_t$, respectively. If $\mathbf{m}_{t,i}$ and $\mathbf{g}_{t,i}$ have different signs, $(\mathbf{m}_{t,i} - \mathbf{g}_{t,i})^2$ is typically larger than $\mathbf{g}_{t,i}^2$ (for similar magnitudes), increasing $\mathbf{v}_{t,i}$ and adaptively decreasing the step size. Meanwhile, a more direct approach to leveraging the sign consistency between momentum and gradient is taken by the Cautious Optimizers [10]. It employs an element-wise mask $\varphi_t$ to selectively apply momentum updates:

$$\varphi_t = \alpha \cdot \mathbb{I}(\mathbf{m}_t \odot \mathbf{g}_t > 0),$$

$$\mathbf{x}_{t+1} = \mathbf{x}_t - \eta_t \mathbf{m}_t \odot \varphi_t.$$

Here, $\mathbb{I}(\cdot)$ is the indicator function, which equals 1 when its argument is positive and 0 otherwise. If the signs $\mathbf{m}_{t,i}$ and $\mathbf{g}_{t,i}$ are align, the momentum component $\mathbf{m}_{t,i}$ is scaled by $\alpha > 1$; otherwise, the update for that component is nullified. This "Cautious Updating" strategy aims to prevent updates from potentially conflicting gradient information.

However, these advanced adaptive methods have notable limitations. AdaBelief's reliance on second-moment estimation restricts its applicability primarily to Adam-style optimizers, thereby rendering it incompatible with newer methods like Lion [12] and Muon [13] that perform well without this component. The Cautious Optimizer, while more broadly applicable, lacks formal stochastic convergence guarantees. As analyzed in Section 4, its binary masking mechanism can aggressively discard gradient information. This behavior may slow convergence, especially in scenarios where the signs of the momentum and gradient align infrequently. Furthermore, Cautious Adam exhibits non-convergent behavior, highlighting a critical flaw in its design (see the counterexample in Appendix A).

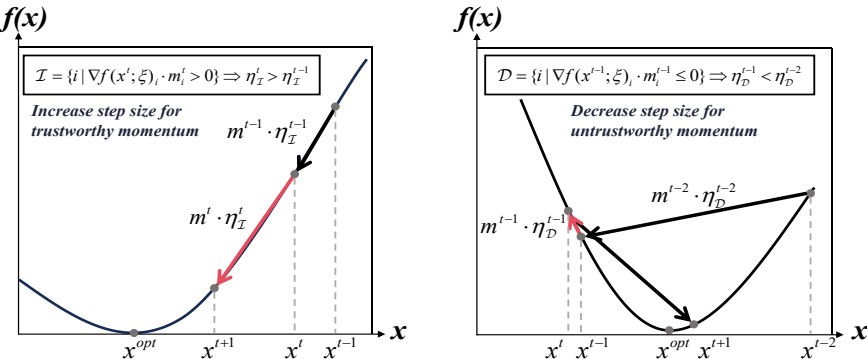

Figure 1: The key idea of MGUP involves adaptively adjusting the learning rate by leveraging the element-wise product of the stochastic gradient and momentum.

## 3    The Proposed Method

This section introduces the **MGUP** (**M**omentum-**G**radient alignment **U**pdate **P**olicy) mechanism for solving Problem (1). Our motivation is to address limitations observed in methods such as AdaBelief and Cautious Optimizers. Figure 1 provides a conceptual illustration of the **MGUP** idea. The pseudocode for a specific implementation variant, **MGUP-AdamW**, is detailed in Algorithm 1. The core steps of **MGUP** are as follows; see Algorithm 2 for the implementation.

▶ **Step 1** : Compute Alignment Scores. For each parameter $i$, calculate its alignment score $\mathbf{s}_{t,i} = \mathbf{m}_{t,i} \cdot \mathbf{g}_{t,i}$.

▶ **Step 2**: Top $K$ Selection. Sort all parameters based on their alignment scores $\mathbf{s}_{t,i}$ in decreasing order, and identify the index set $\mathcal{I}_{\text{topK}}$ of the top $K$ entries, where $K = \lfloor \tau \cdot d \rfloor$ with $\tau \in (0, 1)$.

▶ **Step 3**: Differentiated Update. Adjust the step size $\eta_{t,i}$ computed by the original optimizer as follows: *(i)* If parameter $i \in \mathcal{I}_{\text{topK}}$, its effective step size is set to $\alpha \cdot \eta_{t,i}$. *(ii)* If parameter $i \notin \mathcal{I}_{\text{topK}}$, its effective step size is set to $\gamma \cdot \eta_{t,i}$. Here, $\alpha > 1$ represents the amplification factor, while $\gamma$ denotes the decay factor. In practice, $\alpha$ and $\gamma$ can be set to $1/\tau$ and $\tau$, respectively, where $\tau \in (0, 1)$.

An adjustment based on sign judgment is a concept from prior work (e.g., Cautious Optimizers [10]). Similarly, we can define the Cautious-MGUP mechanism as:

$$\phi_{t,i} = \begin{cases} 1/\tau & \text{if } \mathbf{m}_{t,i} \cdot \mathbf{g}_{t,i} > 0 \\ \tau & \text{if } \mathbf{m}_{t,i} \cdot \mathbf{g}_{t,i} \leq 0. \end{cases} \tag{3}$$

In contrast, **MGUP** offers a more flexible and robust adjustment strategy by introducing a top-$K$ selection and sorting mechanism based on the element-wise product between the momentum and gradient.

We clarify the selection criterion associated with the momentum. While intuitively related to momentum-gradient consistency, **MGUP-AdamW**'s implementation in Algorithm 1 can use the

product of the final update vector $\mathbf{u}_t$ (typically $\mathbf{m}_t/(\sqrt{\mathbf{v}_t} + \epsilon)$) and the gradient $\mathbf{g}_t$, not just momentum $\mathbf{m}_t$ and gradient $\mathbf{g}_t$. This is because, in specific contexts, especially when training large language models, the difference between the selections based on $\mathbf{u}_{t,i} \cdot \mathbf{g}_{t,i}$ and $\mathbf{m}_{t,i} \cdot \mathbf{g}_{t,i}$ may be negligible. Research [24, 25, 26, 27] suggests that within certain model layers, the second moment $\mathbf{v}_t$'s adaptive scaling might be relatively uniform. This implies an approximation where $(\mathbf{m}_1/\sqrt{\mathbf{v}_1}, \ldots, \mathbf{m}_d/\sqrt{\mathbf{v}_d}) \approx (\mathbf{m}_1/c, \ldots, \mathbf{m}_d/c)$ for some constant $c$. Consequently, the sign and relative magnitude ordering from $\mathbf{u}_{t,i} \cdot \mathbf{g}_{t,i}$ would closely mirror that from $\mathbf{m}_{t,i} \cdot \mathbf{g}_{t,i}$. Thus, **MGUP-AdamW** can be intuitively seen as a selection strategy guided by momentum-gradient alignment.

**Remark 3.1.** *For optimizers with simpler update structures, such as Lion, Muon, or standard SGD+Momentum, $\mathbf{m}_{t,i} \cdot \mathbf{g}_{t,i}$ can be directly used as the alignment score.*

**Remark 3.2.** *We explain why **MGUP** is expected to accelerate convergence. (i) **MGUP's mechanism uses a greedy strategy (due to sorting).** Greedy strategies play a key role in accelerating heuristic algorithms. When the stochastic gradient and the update share the same sign, their positive product favors larger step sizes; when their signs differ, the negative product favors smaller ones. Large steps drive acceleration, while small nonzero steps ensure convergence. The necessity of small yet nonzero step sizes is analyzed in Section 4, with a counterexample in Appendix A illustrating the failure of zero step sizes. (ii) **MGUP increases the average update magnitude.** In MGUP, a fraction $\tau$ of parameters update with an increased learning rate of $(1/\tau)lr$, while the remaining $1 - \tau$ use a reduced rate of $\tau lr$. The resulting average learning rate is $\tau \cdot lr \cdot (1/\tau) + (1 - \tau) \cdot lr \cdot \tau = lr \cdot (1 + \tau - \tau^2) > lr$. When individual step sizes are roughly uniform—for example, when Adam's early updates approximate $sign(\mathbf{g}_t)$—the overall update magnitude of **MGUP** increases by a factor of $(1 + \tau - \tau^2)$ compared to Adam. This provides an intuitive explanation for its acceleration effect. We recommend $\tau = \frac{1}{2}$ as the default, since $\arg\max_{\tau \in (0,1)}(1 + \tau - \tau^2) = \frac{1}{2}$.*

---

**Algorithm 1 MGUP-AdamW**

**Input:** Learning rate $\eta > 0$, initial solution $\mathbf{x}_0 \in \mathbb{R}^d$, momentum factors $\beta_1, \beta_2 \in [0, 1)$, weight decay coefficient $\lambda$, stability term $\epsilon > 0$, ratio $\tau \in (0, 1)$.
Set $\mathbf{m}_0 = 0$, $\mathbf{v}_0 = 0$.
**for** $t = 1$ **to** $T$ **do**
    Compute the stochastic gradient $\mathbf{g}_t = \nabla f(\mathbf{x}_t; \xi_t)$
    $\mathbf{m}_t = \beta_1 \mathbf{m}_{t-1} + (1 - \beta_1)\mathbf{g}_t$
    $\mathbf{v}_t = \beta_2 \mathbf{v}_{t-1} + (1 - \beta_2)(\mathbf{g}_t \odot \mathbf{g}_t)$
    $\mathbf{u}_t = \frac{\mathbf{m}_t}{\sqrt{\mathbf{v}_t} + \epsilon}$, $\eta_t = \eta \frac{\sqrt{1 - \beta_2^t}}{1 - \beta_1^t}$
    $\phi_t = \mathbf{MGUP}(\mathbf{u}_t \odot \mathbf{g}_t)$
    $\mathbf{x}_t = (1 - \eta_t \lambda)\mathbf{x}_t$
    $\mathbf{x}_{t+1} = \mathbf{x}_t - \eta_t \phi_t \odot \mathbf{u}_t$
**end for**

---

**Algorithm 2 MGUP**

**Input:** Alignment score vector $\mathbf{s}_t = \mathbf{u}_t \odot \mathbf{g}_t \in \mathbb{R}^d$, ratio $\tau \in (0, 1)$.
**(S1)** Let $\mathcal{I}_{\text{topK}}$ be the index set of the largest $K$ elements of $\mathbf{s}_t \in \mathbb{R}^d$ with $K = \lfloor \tau \cdot d \rfloor$.
**(S2)** Set $\phi_{t,i} = \begin{cases} 1/\tau, & i \in \mathcal{I}_{\text{topK}}; \\ \tau, & \text{else.} \end{cases}$
**return** $\phi_t$

---

**Remark 3.3.** *The **MGUP** method can be easily plugged into existing momentum-based optimization algorithms in a plug-and-play manner. Examples include Lion [12] and Muon [13] (see Appendix H for the pseudocode of **MGUP-Lion** and **MGUP-Muon**).*

## 4 Convergence Analysis

In this section, we rigorously establish both the expected convergence and high-probability convergence guarantees for Algorithm 1 in the stochastic setting.

For the convergence analysis of Algorithm 1, we make the following assumptions:

**Assumption 4.1.** *The function $f$ is bounded from below. There exists $f^* > -\infty$ such that $f(\mathbf{x}) \geq f^*$, for all $\mathbf{x} \in \mathbb{R}^d$.*

**Assumption 4.2.** *The function $f$ is $L$-smooth: $\|\nabla f(\mathbf{y}) - \nabla f(\mathbf{x})\| \leq L\|\mathbf{y} - \mathbf{x}\|$.*

Assumptions 4.1 and 4.2 are standard in the analysis of nonconvex optimization [28, 29, 20, 30, 31].

We first show that, under the following standard assumption, the **MGUP-AdamW**(without weight decay) can achieve the expected convergence rate of $\mathcal{O}(\log(T)/\sqrt{T})$.

**Assumption 4.3.** *The stochastic gradient is unbiased with bounded variance. That is, there exists $\sigma > 0$ such that for all $\mathbf{x} \in \mathbb{R}^d$, $\mathbb{E}[\nabla f(\mathbf{x}; \xi)] = \nabla f(\mathbf{x})$, and $\mathbb{E}[\|\nabla f(\mathbf{x}; \xi) - \nabla f(\mathbf{x})\|_2^2] \leq \sigma^2$. Additionally, we assume that $f(\mathbf{x}; \xi)$ is $M$-Lipschitz, i.e., $\|\nabla f(\mathbf{x}; \xi)\| \leq M$ for all $\mathbf{x}$ and $\xi$.*

Assumption 4.3 is very common in the literature [31, 32, 33]. Theorem 4.1 states our general non-convex convergence result.

**Theorem 4.1.** *Let $\beta_{1,t} = 1 - t^{-1/2}$, $0 < \beta_2 \leq 1$, and $\eta_t = \eta t^{-1/2}/\rho$. We define the following: $\varepsilon_1 = \frac{\sigma^2}{L}$, $\varepsilon_2 = \frac{1}{\rho}\left(\frac{u_{\min}^2}{2u_{\max}^3} - \frac{5L}{\rho u_{\min}^2}\right)$, and $\varepsilon_3 = \frac{1}{2L}$. Here, $u_{\min} = \frac{\epsilon}{\eta}$ and $u_{\max} = \frac{M}{\eta\gamma}$ for some constant learning rate $\eta$ and any $\epsilon > 0$. Let $\rho > \frac{10Lu_{\max}^3}{u_{\min}^4}$ so that $\varepsilon_2 > 0$, and define $\varepsilon_{\min} = \min(\varepsilon_1, \varepsilon_2, \varepsilon_3)$. Under Assumptions 4.1, 4.3, and 4.2, for Algorithm 1 (without weight decay), it holds that:*

$$\frac{1}{T}\sum_{t=1}^{T}\mathbb{E}\|\nabla f(\mathbf{x}_{t+1})\|_2^2 \leq \hat{G}.$$

*where $\hat{G} = \frac{3L^2\eta^2 + 3\rho^2\epsilon^2}{\rho^2\epsilon^2 T}\left(\frac{f(\mathbf{x}_1) - f(\mathbf{x}^*) + 2\sigma^2 L^{-1}\log(T+1)}{\varepsilon_{\min}}\sqrt{T} - 2(\sqrt{T} - 1)\right).$*

**Remark 4.1.** *Convergence relies on the condition $\rho > \frac{10Lu_{\max}^3}{u_{\min}^4}$. Notably, if $\gamma$ can be 0, $u_{\max}$ approaches infinity, making $\frac{10Lu_{\max}^3}{u_{\min}^4}$ unbounded. This renders the condition $\rho > \frac{10Lu_{\max}^3}{u_{\min}^4}$ ill-defined, as $\rho$ would need to be infinitely large, which is unattainable and adversely affects convergence.*

**Remark 4.2.** *While our analysis assumes global Lipschitz continuity, the algorithm can be implemented using $M_T = \max_{j \in [T]} \|\nabla f(\mathbf{x}_j; \xi)\|$ instead of a global bound $M$. This approach only requires bounded gradients along the optimization trajectory, typically yields tighter bounds, and remains fully compatible with our theoretical guarantees. Furthermore, setting $\eta_t = \eta \cdot t^{-1/2}/\rho$ instead of $\eta \frac{\sqrt{1-\beta_2^t}}{1-\beta_1^t} \cdot t^{-1/2}/\rho$ is justified since $\frac{\sqrt{1-\beta_2^t}}{1-\beta_1^t}$ is bounded and can be absorbed into the constant $\eta$ without loss of generality. See Appendix C for details.*

Next, under the assumption of coordinate-wise random noise, we show that the **MGUP-AdamW**(without weight decay) also achieves a rate of $\mathcal{O}(\text{poly}(\log(T))/\sqrt{T})$ with high probability.

**Assumption 4.4.** *The stochastic gradient is unbiased with coordinate-wise bounded variance. That is, there exists $\sigma_i > 0$ such that for all $\mathbf{x} \in \mathbb{R}^d$, $\mathbb{E}[\nabla f(\mathbf{x}; \xi)] = \nabla f(\mathbf{x})$, and $\mathbb{E}[(\nabla f(\mathbf{x}; \xi)_i - \nabla f(\mathbf{x})_i)^2] \leq \sigma_i^2$ for all $i$.*

Assumption 4.4 is commonly used in the literature [34, 35, 36, 37, 38, 32]. Note that the coordinate-wise noise bound in Assumption 4.4 is stronger than the standard bound $\mathbb{E}\|\nabla f(\mathbf{x}; \xi) - \nabla f(\mathbf{x})\|_2^2 \leq \sigma^2$, as the latter can be readily derived from the former. This relaxed choice is made to facilitate the application of probabilistic inequalities, thereby achieving improved convergence properties.

**Theorem 4.2.** *Let $0 \leq \beta_1 < \beta_2 < 1$, $\beta_2 = 1 - 1/T$, $\eta = C_0\sqrt{1-\beta_2}$, $\omega = (\sqrt{1 + 1/\beta_2} + 1)\max\{1, \gamma, 1/\gamma\}$, $\gamma \in (\frac{2}{\beta}, 1)$, and $\beta_3 = \max\left\{\frac{1-\beta_2}{\sqrt{1-\beta_2}}, \frac{2-\gamma^2(1+\beta_2)}{\gamma\sqrt{1-\beta_2}}, \frac{|\beta_2-\gamma^2|+1-\gamma^2}{\gamma\sqrt{1-\beta_2}}\right\}$ for some constants $C_0 > 0$, $\beta > 2$. Under Assumptions 4.1, 4.2, and 4.4, for Algorithm 1(without weight decay), then for any given $\delta \in (0, 1/2)$, it holds that with probability at least $1 - 2\delta$,*

$$\frac{1}{T}\sum_{t=1}^{T}\|\nabla f(\mathbf{x}_t)\|_2^2 \leq \tilde{\mathcal{O}}(T^{-1/2}).$$

**Remark 4.3.** *Setting $\gamma > 0$ is crucial for ensuring the stable convergence of the algorithm. The convergence proof relies on surrogate stepsizes (defined in equations (10) and (12)) to manage the complex interplay between stochastic gradients and adaptive stepsizes. The theoretical framework for employing these surrogate stepsizes within the proof is informed by the methodologies presented in [35, 39, 40], as follows:*

$$\mathbf{y}_{t+1} = \mathbf{y}_t - \eta_t\phi_t \odot \frac{\mathbf{g}_t}{\mathbf{b}_t} + \frac{\beta_1}{1-\beta_1}\left(\frac{\eta_t\mathbf{b}_{t-1}\odot\phi_t}{\eta_{t-1}\mathbf{b}_t\odot\phi_{t-1}} - \mathbf{1}_d\right) \odot (\mathbf{x}_t - \mathbf{x}_{t-1}).$$

*where the precise definitions of all terms are provided in Appendix E. Notably, if $\gamma$ were set to 0, the ratio $\frac{\phi_{t,i}}{\phi_{t-1,i}}$ could approach infinity for some component $i$ when $\phi_{t,i} = \alpha$ and $\phi_{t-1,i} = \gamma = 0$. Such occurrences might prevent parameter updates in certain iterations, thereby hindering convergence. Consequently, $\gamma$ is set to a positive value instead of 0. For a more detailed discussion, please refer to Appendix E.*

**Remark 4.4.** *Theorem 4.1 and Theorem 4.2 are independent of the specific mask selection mechanism.*

Combining Remark 4.1 and Remark 4.3, for Adam variants employing a mask, simply nullifying the update step when momentum and gradient directions misalign (tantamount to setting $\gamma = 0$) markedly alters the convergence properties of stochastic optimization.

## 5 Experiments

In this section, we evaluate the performance of the proposed **MGUP** optimizers on both pretraining and supervised fine-tuning (SFT) tasks. All experiments are conducted using two NVIDIA V100 (32GB) GPUs and four NVIDIA RTX 4090 (24GB) GPUs. Detailed experimental settings are provided in Appendix G.

▶ **Datasets.** We use the image dataset CIFAR-10, the text dataset Wikitext-103, and the language model fine-tuning benchmarks GLUE and GSM-8K.

▶ **Compared Methods.** We compare **MGUP-AdamW**, **MGUP-Lion**, **MGUP-Muon** with (*i*) AdamW [41], (*ii*) Cautious Optimizers(C-AdamW, C-Lion, C-Muon) [10], (*iii*) Lion [12], (*iv*) Muon [13, 42], as well as other state-of-the-art memory-efficient optimization methods such as (*v*) GaLore [3], (*vi*) LDAdam [4], (*vii*) Adam-mini [27] and (*viii*) Adam-8Bit [43].

Unless stated otherwise, the default setting for **MGUP**-enhanced optimizers is $\tau = \gamma = 1/\alpha = \frac{1}{2}$.

### 5.1 Pretraining

▶ **Image MAE Pretraing** We pre-train a straightforward ViT model [44] with the Masked Autoencoder (MAE) framework [45] on the CIFAR-10 dataset. For this experiment, we set the learning rate to $1.5 \times 10^{-4}$, the MAE mask rate to 75%, and train for 200 epochs. We compare **MGUP-AdamW** with the standard AdamW and C-AdamW optimizers by evaluating their training and validation losses. The results, presented in Figure 2, show that **MGUP-AdamW** consistently achieves lower training and validation loss throughout the training process. In contrast, the performance of C-AdamW gradually falls behind that of AdamW.

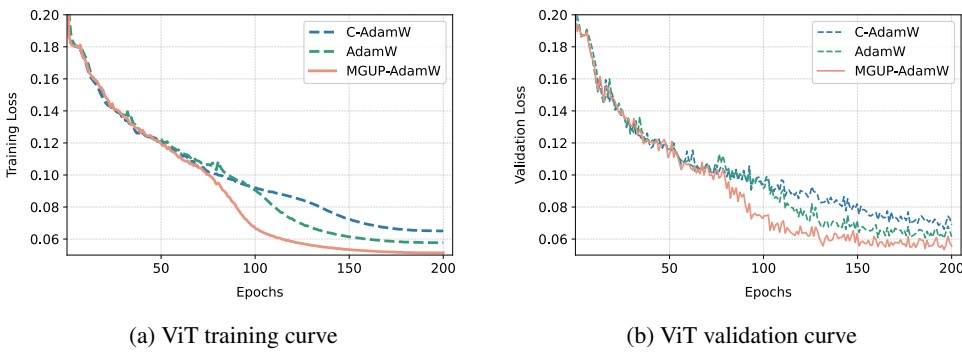

(a) ViT training curve                    (b) ViT validation curve

Figure 2: ViT MAE training and validation curves on CIFAR-10

▶ **Language Modeling** We employ a straightforward LLaMA2-71M model [46] and a Qwen2.5-150M model [47], both of which are pretrained on the WikiText-103 dataset.

**LLaMA2-71M on WikiText-103.** To assess optimizer performance on a smaller language model, we train LLaMA2-71M on WikiText-103, evaluating validation loss. We compare AdamW, Lion, and Muon variants using a learning rate of 3e-4, a batch size of 480, and 2000 training steps. As shown in Figure 3a, the results highlight several key differences. Among the Adam-type optimizers, **MGUP-AdamW** achieves a 1.6x speedup over standard AdamW and exhibits superior generalization compared to C-AdamW. For the Lion-type optimizers, **MGUP-Lion** demonstrates a 2.5x speedup over standard Lion; unlike the unstable C-Lion which shows early loss spikes, **MGUP-Lion** maintains training stability. With the Muon-type optimizers, **MGUP-Muon** yields a ~1.2x speedup relative to Muon and delivers better generalization than C-Muon.

While $\tau$ serves as the primary hyperparameter in our approach, it is essential to examine how variations in $\gamma$ influence the performance of **MGUP-AdamW**. We conduct experiments with $\tau \in \{0.3, 0.5, 0.7\}$

and $\gamma \in \{0, 0.1, 0.5, 0.9\}$ to evaluate this relationship across different hyperparameter configurations. The comparative results are presented in Figure 3b. The analysis indicates the following: *(i)* with $\gamma$ fixed, increasing $\tau$ beyond a certain threshold degraded performance; *(ii)* with $\tau$ fixed, a larger $\gamma$ generally improved performance. The findings in *(ii)* precisely corroborate the discussion on the setting of $\gamma$ in Section 4.

**Qwen2.5-150M on WikiText-103.** We also evaluate optimizers on a larger Qwen2.5-150M model using WikiText-103 (Figure 4). For these experiments, we use a learning rate of 1e-3, a batch size of 160, and 1500 training steps. For Adam-type optimizers, **MGUP-AdamW** demonstrates a higher speedup than standard AdamW and better generalization than C-AdamW. For Muon-type optimizers, **MGUP-Muon** achieves a 1.1x speedup over standard Muon and superior generalization compared to C-Muon.

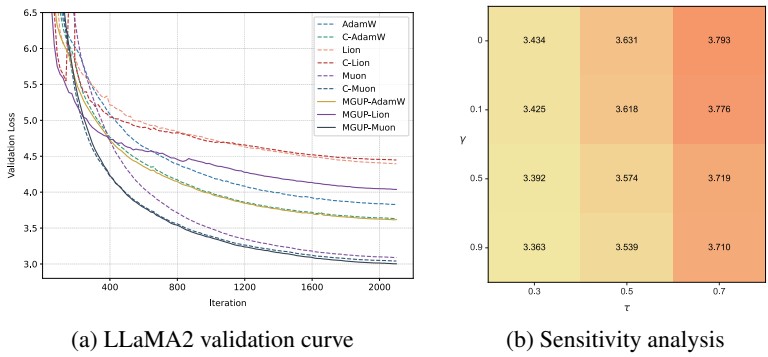

(a) LLaMA2 validation curve

(b) Sensitivity analysis

Figure 3: LLaMA2-71M validation curve and MGUP-AdamW sensitivity analysis on WikiText-103

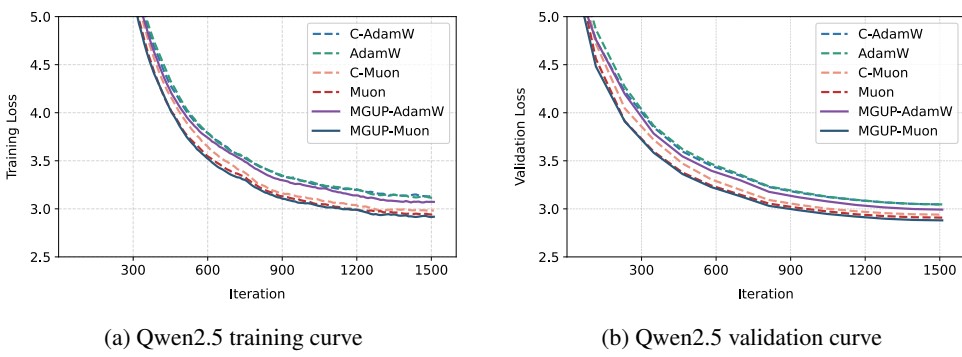

(a) Qwen2.5 training curve

(b) Qwen2.5 validation curve

Figure 4: Qwen2.5-150M training and validation curves on WikiText-103

## 5.2 Fine-Tuning

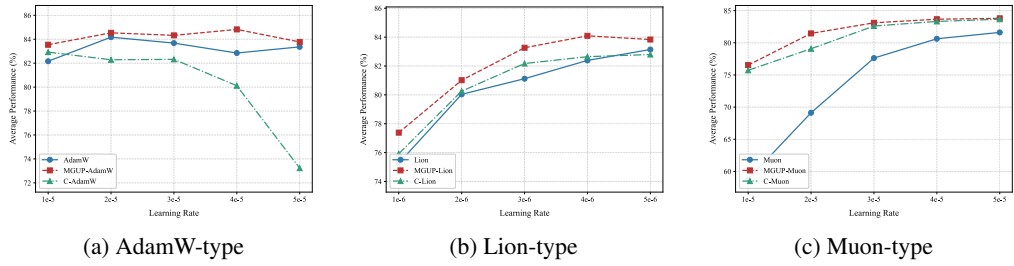

(a) AdamW-type

(b) Lion-type

(c) Muon-type

Figure 5: Adamw-type, Lion-type, Muon-type optimizers average performance across GLUE tasks

We conduct comprehensive experiments on downstream tasks, with particular emphasis on supervised fine-tuning (SFT) scenarios. Our evaluation encompassed two representative tasks: fine-tuning

the RoBERTa-base model [48] on the GLUE benchmark and the LLaMA2-7B model [46] on the GSM-8K.

▶ **GLUE Benchmark Evaluation.** To evaluate performance and generalization on diverse Natural Language Understanding (NLU) tasks, we experiment on the GLUE benchmark, which comprises tasks varying in dataset size and complexity. We perform a learning rate search within the range of 1e-5 to 5e-5 for most optimizers, and within the range of 1e-6 to 5e-6 for Lion-type optimizers. The best performance for each task is reported in Table 1. On most tasks, **MGUP-AdamW** and **MGUP-Muon** achieve state-of-the-art results. Notably, **MGUP-AdamW** reaches an average optimal performance of **85.15** across all GLUE tasks.

Figure 5 shows how the average GLUE score changes across the tested learning rates. **MGUP-AdamW**, **MGUP-Lion**, and **MGUP-Muon** consistently outperform their standard counterparts AdamW, Lion, and Muon across this range. Additionally, all **MGUP**-enhanced optimizers demonstrate greater robustness compared to the cautious variants: C-AdamW, C-Lion, and C-Muon.

▶ **GSM-8K Fine-tuning.** We further evaluate **MGUP-AdamW** by fine-tuning LLaMA2-7B on the challenging GSM-8K dataset, a critical indicator of fine-tuning effectiveness due to typically low zero-shot accuracy [49]. We conduct a learning rate grid search (from 1e-5 to 5e-5), consistent with [4]. As shown in Table 2, **MGUP-AdamW** achieves lower training loss per epoch and the highest validation accuracy **34.96%**, outperforming baseline optimizers.

Table 1: Comparison of best results of fine-tuning RoBERTa-base model on GLUE benchmark.

| Method | RTE 2.5k | MRPC 3.7k | STS-B 7k | CoLA 8.5k | SST-2 67k | QNLI 105k | QQP 364k | Avg. |
|---|---|---|---|---|---|---|---|---|
| AdamW [41] | 72.93 | **90.44** | **90.55** | 60.32 | 94.84 | 92.79 | 91.34 | 84.74 |
| Lion [12] | 67.15 | 87.50 | 89.39 | 60.57 | 94.84 | 93.00 | 91.32 | 83.39 |
| Muon [13] | 64.62 | 81.13 | 87.33 | 59.34 | 94.27 | 93.11 | 91.72 | 81.65 |
| Adam-mini [27] | 56.32 | 87.01 | 89.49 | 56.32 | 93.35 | 92.02 | 89.58 | 80.44 |
| GaLore(r=8) [3] | 69.45 | 86.19 | 88.97 | 55.12 | 94.15 | 92.01 | 89.86 | 82.25 |
| LDAdamW(r=8) [4] | 67.58 | 88.32 | 90.03 | 60.60 | 94.49 | 92.82 | 91.23 | 83.58 |
| C-AdamW [10] | 71.12 | 89.22 | 90.25 | 57.29 | 93.92 | 92.62 | 91.39 | 83.69 |
| C-Lion [10] | 67.87 | 88.73 | 89.58 | 57.78 | 94.50 | 92.81 | 91.41 | 83.23 |
| C-Muon [10] | 70.04 | 88.24 | 90.04 | 59.81 | 94.84 | 93.19 | 91.75 | 83.98 |
| **MGUP-Lion** | 71.12 | 88.24 | 90.07 | **61.23** | 94.27 | 93.04 | 91.33 | 84.18 |
| **MGUP-Muon** | 70.40 | 88.24 | 89.84 | 61.07 | 94.61 | **93.24** | **91.78** | 84.17 |
| **MGUP-AdamW** | **75.81** | **90.44** | 90.54 | 59.83 | **94.95** | 93.08 | 91.43 | **85.15** |

Table 2: Fine-tuning results for LLaMA-2 on GSM-8k.

| Model | Metric | AdamW | AdamW-8b | LDAdamW ($rank = 512$) | GALore ($rank = 512$) | C-AdamW | MicroAdamW ($m = 10$) | MGUP-AdamW |
|---|---|---|---|---|---|---|---|---|
| 7B | Accuracy | 34.53 | 34.42 | 34.88 | 34.62 | 34.68 | 34.58 | **34.96** |
| | Train loss | 0.064 | 0.069 | 0.073 | 0.070 | 0.081 | 0.057 | **0.056** |

# 6 Conclusion

We introduce **MGUP**, a novel intra-layer parameter selection mechanism based on momentum-gradient alignment, and integrated it into AdamW, Lion, and Muon yields **MGUP-AdamW**, **MGUP-Lion**, and **MGUP-Muon**. Empirically, **MGUP** Optimizers demonstrate competitive convergence speeds and superior generalization over their base versions across diverse tasks, including large language model training. Theoretically, we establish stochastic convergence guarantees for **MGUP-AdamW**(without weight decay) under standard non-convex assumptions, achieving a rate near the known optimum. Limitations include the pre-selection of $\tau$, inviting future work on adaptive methods. Our theoretical analysis also primarily covers **MGUP-AdamW** (without weight decay). Thus, while empirically effective with optimizers like Lion and Muon, **MGUP**'s theoretical properties (e.g., the necessity of $\gamma > 0$) in these diverse frameworks require further study.

## Acknowledgments

This work was supported by NSFC (61772570), and Guangdong Natural Science Funds for Distinguished Young Scholar (2018B030306025).

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

# Appendix

The appendices are structured as follows:

- Appendix A gives a counterexample showing that Cautious Adam may diverge.
- Appendix B summarizes additional related work.
- Appendix C provides the definitions and lemmas related to Theorem 4.1.
- Appendix D offers the formal proof of Theorem 4.1.
- Appendix E presents the definitions and lemmas related to Theorem 4.2.
- Appendix F includes the formal proof of Theorem 4.2.
- Appendix G supplies additional details regarding the experimental setup.
- Appendix H contains the pseudocode for other MGUP-type algorithms.
- Appendix I presents more experimental results.

## A  Motivating Counterexample: The Necessity of $\gamma > 0$

To intuitively demonstrate the necessity of a non-zero decayed step size ($\gamma > 0$) for misaligned updates, we present a counterexample where optimizers that nullify updates (i.e., $\gamma = 0$), such as Cautious Adam (C-Adam) [10], fail to converge. We adapt a classic construction from [50].

Consider the one-dimensional objective function $f(x) = \sum_{i=0}^{n-1} f_i(x)$, where the stochastic components are defined as:

$$f_i(x) = \begin{cases} nx, & x \geq -1 \\ \frac{n}{2}(x+2)^2 - \frac{3n}{2}, & x < -1 \end{cases} \quad \text{for } i = 0.$$

$$f_i(x) = \begin{cases} -x, & x \geq -1 \\ -\frac{1}{2}(x+2)^2 + \frac{3}{2}, & x < -1 \end{cases} \quad \text{for } i > 0.$$

The full objective is $f(x) = \sum_{i=0}^{n-1} f_i(x)$, which simplifies to:

$$f(x) = \begin{cases} x, & x \geq -1 \\ \frac{1}{2}(x+2)^2 - \frac{3}{2}, & x < -1. \end{cases}$$

We analyze the behavior of C-Adam and MGUP-Adam on a counterexample with its global minimum at $x^* = -2$, starting from an initial point $x_0 = -0.5$. In this environment, the optimizer encounters frequent, small negative gradients $g_t = -1$ and rare, large positive gradients $g_t = n$.

▶ **Analysis of C-Adam's Failure.** The stochastic nature of the gradients induces a "pulse-decay" dynamic in the momentum term $m_t$. A rare positive gradient pulse pushes $m_t$ to a high value, after which the frequent negative gradients cause it to decay. This leads to persistent oscillations of the momentum around zero, a behavior empirically confirmed in Figure 6b.

This momentum instability is detrimental to C-Adam ($\gamma = 0$). When $m_t > 0$, the frequent, correctly-signed gradients $g_t = -1$ are misaligned with the momentum $m_t g_t < 0$, causing the optimizer to skip the update. When $m_t < 0$, these same gradients are aligned $m_t g_t > 0$, but they produce an incorrect update, pushing the parameter $x$ away from the optimum $x^* = -2$. Figure 6c provides clear evidence for this dysfunction: C-Adam's updates are either null ($\Delta x_t = 0$) or strictly positive ($\Delta x_t > 0$), moving in the wrong direction. As a result, the iterates not only stagnate but actively diverge from the minimum, as illustrated by the trajectory in Figure 6a.

▶ **MGUP's Advantage.** In stark contrast, **MGUP-Adam** leverages its safeguarding mechanism ($\gamma > 0$) to overcome this failure mode. The critical scenario is when $m_t > 0$ and the gradient is $g_t = -1$. Instead of inaction, MGUP performs a small, corrective update of size $\gamma \eta_t$ in the proper negative direction. This ensures a persistent, albeit small, push towards the optimum. The scatter plot in Figure 6c demonstrates that **MGUP-Adam** consistently performs updates in the correct direction ($\Delta x_t < 0$). These small but steady corrective steps enable the optimizer to escape the challenging region and successfully converge to the true minimum at $x^* = -2$, as shown by its trajectory in Figure 6a.

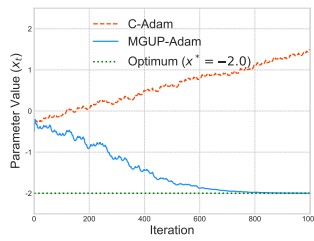 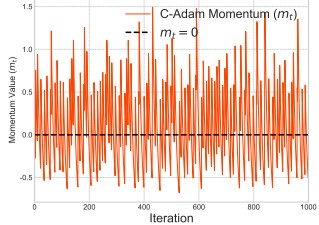 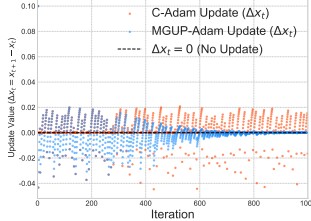

| (a) Parameter Trajectory | (b) Momentum Dynamics of C-Adam | (c) Analysis of Update Steps |

Figure 6: Analysis of C-Adam's failure and **MGUP-Adam**'s success in the counterexample. (a) **MGUP-Adam** converges to the optimum $x^* = -2$ while C-Adam diverges. (b) The momentum in C-Adam oscillates around zero, leading to unstable update decisions. (c) A scatter plot of update steps shows C-Adam either skips updates ($\Delta x_t = 0$) or updates in the wrong direction ($\Delta x_t > 0$), whereas **MGUP-Adam** consistently updates in the correct direction ($\Delta x_t < 0$).

## B  More Related Work

### B.1  Efficient Training and Parameter Update Strategies

Large Language Model (LLM) training often exhibits gradient updates with inherent low-rank characteristics [1, 2]. This observation has spurred the development of methods aiming to enhance training efficiency and reduce memory footprint. Techniques like Adafactor significantly cut memory needs by applying low-rank decomposition to Adam's second-order moments [51], while Adam-mini achieves further optimization using Transformer-specific Hessian-based storage strategies [27]. More directly exploiting gradient structure, GaLore enhances memory and computational efficiency through low-rank projection of gradient matrices [3], and LDAdam complements this by optimizing within low-dimensional gradient subspaces [4]. Furthermore, Q-Galore integrates quantization with low-rank projections, boosting efficiency, particularly in resource-constrained scenarios [52].

While the aforementioned methods focus on compressing or projecting the update information, another significant avenue for efficiency involves selectively updating only a subset of model parameters. This strategy operates on the principle that not all parameters contribute equally to learning at every stage. Such selective approaches have been particularly explored for accelerating the fine-tuning or adaptation phase of LLMs, although their applicability to large-scale pre-training is less established compared to full parameter updates.

For instance, SIFT [5] achieves efficient adaptation through gradient-based sparse parameter updates, leveraging the low intrinsic dimensionality and sparse gradient characteristics observed in LLMs during fine-tuning. Applying the selective principle at a coarser granularity, layer-wise and block-wise strategies have also proven effective, primarily in fine-tuning contexts. Building upon early unsupervised pre-training concepts [53, 54], the LOMO method enables efficient gradient calculation and grouped updates, allowing for full-parameter fine-tuning with reduced memory [7], later enhanced by AdaLOMO with adaptive learning rates [55]. The LISA method introduces an innovative layer selection strategy based on parameter norms to further optimize the update process during fine-tuning [8]. Similarly, BAdam implemented a block coordinate optimization framework, selecting parameter blocks for Adam updates, thereby reducing memory and computation specifically for adaptation tasks [9].

These diverse approaches highlight prominent pathways towards more efficient training and adaptation: leveraging low-rank approximations, or strategically selecting which parameters or parameter groups receive updates, with many selective methods currently specialized for post-pre-training stages.

### B.2  Evolution of Adaptive Optimization Methods

First-order optimization methods play a critical role in deep learning. Building on early foundational work, the Momentum method accelerates the optimization process by accumulating historical gradi-

ents [56]. Subsequently, the RMSprop algorithm introduced the concept of adaptive learning rates, which enables distinct update steps for different parameters [57]. The Adam algorithm merges the benefits of Momentum and RMSprop, adaptively adjusting both first and second moments, and is widely used for its effective adaptive moment estimation [23].

Building upon Adam, researchers have proposed several enhanced variants. The AMSGrad method [58] aims to strengthen optimization stability by utilizing the maximum of historical second-order moments, while the NAdam approach [59] incorporates Nesterov momentum to enhance performance. To address challenges with weight decay and learning rate adjustment in Adam, the AdamW method [41] enhances L2 regularization through decoupled weight decay, and the AdaBound method [60] introduces learning rate bounds to prevent excessively large or small update steps. The RAdam method [61] is designed to enhance convergence stability by rectifying variance estimation during the early stages of training. Finally, the AdaBelief algorithm [11] refines the computation of second-order moments by employing the exponential moving average of gradient deviations from their mean, a design choice intended to improve generalization performance.

Recent advances have further expanded the capabilities of adaptive optimization methods. Xie et al. [62] presents the Adai framework from the perspective of dynamical systems, accelerating training and improving minima selection by decoupling the effects of adaptive learning rates and momentum. In [33], the Adan method is introduced. This method incorporates a novel Nesterov momentum estimation approach designed to accelerate convergence without incurring additional gradient computation overhead. More recently, the C-AdamW method is proposed in [10], which employs a masking strategy aimed at enhancing optimization efficiency.

Beyond Adam variants, other optimizers also demonstrate considerable advantages. The Sophia method, detailed in [63], enhances Adam's second-moment estimation through efficient diagonal Hessian approximation combined with coordinate-wise clipping. This approach has demonstrated superior performance in language model pretraining. In contrast, the Lion optimizer, presented in [12], is designed to optimize memory efficiency and computational speed. It achieves this by tracking momentum exclusively and employing a sign-based operator to standardize update magnitudes. Notably, the Muon method, proposed in [13] and originating from the framework of Shampoo [64], incorporates Newton-Schulz-iterated orthogonalization of gradient momentum. This technique is intended to enhance convergence dynamics through adaptation to parameter curvature.

### B.3 A Brief Review on the Convergence of Adam

The convergence theory for the Adam optimizer has evolved from early uncertainty to rigorous proof. Initially, Reddi et al. [58] revealed its risk of non-convergence with a convex counterexample. Early work to address this established conditional guarantees; for instance, Chen et al. [29] provided the first convergence rate in non-convex settings, while Zou et al. [65] identified necessary hyperparameter coupling conditions to ensure stability. A key turning point was the work of Zhang et al. [50], who first proved that the unmodified Adam algorithm is convergent, attributing prior failures to a mismatch between hyperparameters and the specific problem rather than an inherent algorithmic flaw. Building on this, He et al. [66] strengthened the guarantee from ergodic to the more practical last-iterate convergence. Finally, Wang et al. [32] resolved the debate by proving that Adam achieves an optimal iteration complexity of $\mathcal{O}(\epsilon^{-4})$, matching the theoretical lower bound and providing a firm theoretical foundation for its excellent empirical performance.

## C  Expectation Convergence Lemmas

### C.1  Definition

Recall the form of Algorithm 1. Let $\mathbf{g}_t$ denote the stochastic gradient. Let's consider $\beta_{1,t} = 1 - t^{-1/2}$, $\eta_t = \eta t^{-1/2}/\rho$. Therefore, we can rewrite the formal definition of the algorithm,

$$\mathbf{m}_t = \beta_{1,t}\mathbf{m}_{t-1} + (1 - \beta_{1,t})\mathbf{g}_t,$$
$$\mathbf{v}_t = \beta_2\mathbf{v}_{t-1} + (1 - \beta_2)\mathbf{g}_t^2,$$
$$\mathbf{v}_t' = \rho\sqrt{t}\max(\epsilon, \sqrt{\mathbf{v}_t}), \rho > 0,$$
$$\eta_t = \eta,$$
$$\mathbf{h}_t = \frac{\mathbf{v}_t'}{\eta_t\phi_t}, \tag{4}$$
$$\mathbf{x}_{t+1} = \mathbf{x}_t - \frac{\mathbf{m}_t}{\mathbf{h}_t}.$$

Here, we incorporate $\eta_t = \eta t^{-1/2}/\rho$ into $\mathbf{v}_t'$ and set $\eta_t = \eta$. Without loss of generality, for $\phi_t \in \{\alpha, \gamma\}$, we set $\alpha = 1$ and $\gamma \in (0, 1)$. Then, we define that

$$u_{\min} = \frac{\epsilon}{\eta}, u_{\max} = \frac{M}{\eta\gamma}, \kappa = \frac{u_{\max}}{u_{\min}},$$
$$\mathbf{r}_t = \mathbf{h}_t \odot (\mathbf{x}_{t+1} - \mathbf{x}_t), \tag{5}$$
$$\mathbf{s}_t = \mathbf{m}_t - \nabla f(\mathbf{x}_t).$$

## C.2  Lemma C.1

**Lemma C.1.** *Suppose that $\{E_i, A_i\}$ are two nonnegative sequences. Assume $E_{t+1} \le (1 - (t + 1)^{-1/2})E_t + A_{t+1}$ , and $\delta \ge 1/2$. Then we have:*

$$t^{-1/2}E_t \le 2^\delta(E_t - E_{t+1} + A_{t+1}).$$

*Proof.* We derive:

$$t^{-1/2}E_t - c\left(E_t - E_{t+1} + A_{t+1}\right)$$
$$\overset{(\bullet)}{\le} t^{-1/2}E_t - c\left(E_t + A_{t+1}\right) + c \cdot \left(E_t - (t+1)^{-1/2}E_t + A_{t+1}\right)$$
$$= E_t\left(t^{-1/2} - c(t+1)^{-1/2}\right)$$
$$= E_t \cdot (t+1)^{-1/2} \cdot \left(\left(\frac{t}{t+1}\right)^{-1/2} - c\right)$$
$$\overset{(\circ)}{\le} E_t \cdot (t+1)^{-1/2} \cdot (2^{1/2} - c)$$
$$\overset{(\star)}{\le} 0.$$

where ($\bullet$) follows from $E_{t+1} \le (1 - (t+1)^{-1/2})E_t + A_{t+1}$; ($\circ$) is due to $(\frac{t}{t+1})^{-1/2} \le 2^{1/2}$; ($\star$) is due to our choice $c = 2^\delta$. $\qquad\square$

## C.3  Lemma C.2

**Lemma C.2.** *We have the following results for all $t \ge 1$, $\rho\sqrt{t}u_{\min} \le \min(\mathbf{h}_t) \le \rho\sqrt{t}u_{\max}$.*

*Proof.* We obtain:

$$\mathbf{v}_{t,i} = (1 - \beta_2) \sum_{j=1}^{t} \beta_2^{t-j} \mathbf{g}_{j,i}^2$$

$$\leq (1 - \beta_2)(\max_{j \in [t]} \mathbf{g}_{j,i}^2) \sum_{j=1}^{t} \beta_2^{t-j}$$

$$\overset{(\circ)}{\leq} \max_{j \in [t]} |\mathbf{g}_j|^2$$

$$\overset{(\star)}{\leq} M^2,$$

where $(\circ)$ is due to $\sum_{i=1}^{t} \beta_2^{t-i} = \frac{1 - \beta_2^t}{1 - \beta_2} \leq \frac{1}{1 - \beta_2}$; $(\star)$ is due to we assume that $f(\mathbf{x}; \xi)$ is M-Lipschitz for all $\mathbf{x}$. Additionally, we note that

$$\mathbf{v}_{t,i} \geq 0.$$

Thus, we conclude:

$$\mathbf{v}_{t,i} \in [0, M^2].$$

This implies:

$$\mathbf{v}'_{t,i} \in [\rho\sqrt{t}\epsilon, \rho\sqrt{t}M].$$

Next, according to the definition of $u_{\min}$ and $u_{\max}$:

$$u_{\min} = \frac{\epsilon}{\eta}, u_{\max} = \frac{M}{\eta\gamma}.$$

Therefore, we have:

$$\mathbf{h}_{t,i} \in \left[\frac{\rho\sqrt{t}\epsilon}{\eta}, \frac{\rho\sqrt{t}M}{\eta\gamma}\right] = \left[\rho\sqrt{t}u_{\min}, \rho\sqrt{t}u_{\max}\right].$$

$\square$

## C.4   Lemma C.3

**Lemma C.3.** *Let $u_{\min}, u_{\max}, \mathbf{r}_t, \mathbf{s}_t$ be given in (5). We have the following inequality:*

$$\mathbb{E}[f(\mathbf{x}_{t+1})] \leq f(\mathbf{x}_t) - \left(\frac{1}{2\kappa^2 \rho u_{\max}} - \frac{L}{\rho^2 u_{\min}}\right) \frac{\mathbb{E}\|\mathbf{r}_t\|_2^2}{\sqrt{t}} + \frac{\mathbb{E}\|\mathbf{s}_t\|_2^2}{2\sqrt{t}L}.$$

*Proof.* First, we have the following inequalities:

$$\|\mathbf{x}_{t+1} - \mathbf{x}_t\|_{\mathbf{h}_t}^2 = \|\mathbf{x}_{t+1} - \mathbf{x}_t\|_{\mathbf{h}_t}^2 \cdot \frac{\max(\mathbf{h}_t)^2}{\min(\mathbf{h}_t)^2} \cdot \frac{1}{\kappa^2}$$

$$\geq \|\mathbf{x}_{t+1} - \mathbf{x}_t\|_2^2 \cdot \min(\mathbf{h}_t) \cdot \frac{\max(\mathbf{h}_t)^2}{\min(\mathbf{h}_t)^2} \cdot \frac{1}{\kappa^2}$$

$$= \|\mathbf{x}_{t+1} - \mathbf{x}_t\|_2^2 \cdot \frac{\max(\mathbf{h}_t)^2}{\min(\mathbf{h}_t)} \cdot \frac{1}{\kappa^2} \qquad (6)$$

$$\geq \frac{1}{\kappa^2 \min(\mathbf{h}_t)} \|\mathbf{h}_t \odot (\mathbf{x}_{t+1} - \mathbf{x}_t)\|_2^2$$

$$= \frac{1}{\kappa^2 \min(\mathbf{h}_t)} \|\mathbf{r}_t\|_2^2.$$

Applying the descent Lemma to the algorithm, we have

$$f(\mathbf{x}_{t+1}) \le f(\mathbf{x}_t) + \langle \nabla f(\mathbf{x}_t), \mathbf{x}_{t+1} - \mathbf{x}_t \rangle + \frac{L}{2} \|\mathbf{x}_{t+1} - \mathbf{x}_t\|_2^2$$

$$= f(\mathbf{x}_t) + \langle \mathbf{m}_t, \mathbf{x}_{t+1} - \mathbf{x}_t \rangle - \langle \mathbf{m}_t - \nabla f(\mathbf{x}_t), \mathbf{x}_{t+1} - \mathbf{x}_t \rangle + \frac{L}{2} \|\mathbf{x}_{t+1} - \mathbf{x}_t\|_2^2$$

$$\overset{(\bullet)}{\le} f(\mathbf{x}_t) - \frac{1}{2}\|\mathbf{x}_{t+1} - \mathbf{x}_t\|_{\mathbf{h}_t}^2 - \langle \mathbf{m}_t - \nabla f(\mathbf{x}_t), \mathbf{x}_{t+1} - \mathbf{x}_t \rangle + \frac{L}{2} \|\mathbf{x}_{t+1} - \mathbf{x}_t\|_2^2$$

$$\overset{(\circ)}{\le} f(\mathbf{x}_t) - \frac{1}{2}\|\mathbf{x}_{t+1} - \mathbf{x}_t\|_{\mathbf{h}_t}^2 + \frac{1}{2\beta L}\|\mathbf{m}_t - \nabla f(\mathbf{x}_t)\|_2^2 + \frac{(\beta+1)L}{2}\|\mathbf{x}_{t+1} - \mathbf{x}_t\|_2^2$$

$$\le f(\mathbf{x}_t) - \frac{1}{2}\|\mathbf{x}_{t+1} - \mathbf{x}_t\|_{\mathbf{h}_t}^2 + \frac{1}{2\beta L}\|\mathbf{m}_t - \nabla f(\mathbf{x}_t)\|_2^2 + \frac{(\beta+1)L}{2\min(\mathbf{h}_t)^2}\|\mathbf{h}_t \odot (\mathbf{x}_{t+1} - \mathbf{x}_t)\|_2^2$$

$$\overset{(\star)}{\le} f(\mathbf{x}_t) - \frac{1}{2\kappa^2 \min(\mathbf{h}_t)}\|\mathbf{r}_t\|_2^2 + \frac{(\beta+1)L}{2\min(\mathbf{h}_t)^2}\|\mathbf{r}_t\|_2^2 + \frac{1}{2\beta L}\|\mathbf{s}_t\|_2^2$$

$$\overset{(*)}{\le} f(\mathbf{x}_t) - \left( \frac{1}{2\kappa^2 \rho\sqrt{t}u_{\max}} - \frac{(\beta+1)L}{2\rho^2 t u_{\min}^2} \right)\|\mathbf{r}_t\|_2^2 + \frac{\|\mathbf{s}_t\|_2^2}{2\beta L},$$

where $(\bullet)$ follows from the equality $\langle \mathbf{m}_t, \mathbf{x}_{t+1} - \mathbf{x}_t \rangle + \|\mathbf{x}_{t+1} - \mathbf{x}_t\|_{\mathbf{h}_t}^2 = 0$; $(\circ)$ is due to Young's inequality; $(\star)$ results from Inequality (6); and $(*)$ is derived from Lemma C.2.

Then, by setting $\beta = \sqrt{t}$ and taking the expectation of both sides, we obtain:

$$\mathbb{E}[f(\mathbf{x}_{t+1})] \le f(\mathbf{x}_t) - \left( \frac{1}{2\kappa^2 \rho\sqrt{t}u_{\max}} - \frac{(\sqrt{t}+1)L}{2\rho^2 t u_{\min}^2} \right)\mathbb{E}\|\mathbf{r}_t\|_2^2 + \frac{1}{2\sqrt{t}L}\mathbb{E}\|\mathbf{s}_t\|_2^2$$

$$\le f(\mathbf{x}_t) - \left( \frac{1}{2\kappa^2 \rho\sqrt{t}u_{\max}} - \frac{L}{\rho^2 \sqrt{t} u_{\min}^2} \right)\mathbb{E}\|\mathbf{r}_t\|_2^2 + \frac{1}{2\sqrt{t}L}\mathbb{E}\|\mathbf{s}_t\|_2^2 \qquad (7)$$

$$= f(\mathbf{x}_t) - \left( \frac{1}{2\kappa^2 \rho u_{\max}} - \frac{L}{\rho^2 u_{\min}^2} \right)\frac{\mathbb{E}\|\mathbf{r}_t\|_2^2}{\sqrt{t}} + \frac{\mathbb{E}\|\mathbf{s}_t\|_2^2}{2\sqrt{t}L}.$$

$\square$

## C.5   Lemma C.4

**Lemma C.4.** *Let* $\mathbf{r}_t, \mathbf{s}_t$ *be given in (5). We define* $S_t = \mathbb{E}\|\mathbf{s}_t\|$, $R_t = \mathbb{E}\|\mathbf{r}_t\|$, $P_t = f(\mathbf{x}_t) - f(\mathbf{x}^*) + \frac{2}{L}S_t^2$, $\varepsilon_1 = \frac{\sigma^2}{L}$, $\varepsilon_2 = \frac{1}{\rho}\left(\frac{1}{2\kappa^2 u_{\max}} - \frac{5L}{\rho u_{\min}^2}\right)$, *and* $\varepsilon_3 = \frac{1}{2L}$. *Assume that* $\rho$ *is sufficiently large such that* $\varepsilon_2 > 0$. *Let* $\varepsilon_{\min} = \min(\varepsilon_1, \varepsilon_2, \varepsilon_3)$. *The following result holds:*

$$\sum_{t=1}^{T} R_t^2 + S_t^2 \le \frac{P_1 + 2\sigma^2 L^{-1}\log(T+1)}{\varepsilon_{\min}} \cdot \sqrt{T} - 2(\sqrt{T}-1).$$

*Proof.* First, we derive the following equalities:

$$\mathbf{s}_t = \mathbf{m}_t - \nabla f(\mathbf{x}_t)$$

$$= \beta_{1,t}\mathbf{m}_{t-1} + (1 - \beta_{1,t})\mathbf{g}_t - \nabla f(\mathbf{x}_t)$$

$$= \beta_{1,t}(\mathbf{m}_{t-1} - \nabla f(\mathbf{x}_{t-1})) + (1 - \beta_{1,t})\mathbf{g}_t + \beta_{1,t}\nabla f(\mathbf{x}_{t-1}) - \nabla f(\mathbf{x}_t)$$

$$= \underbrace{\beta_{1,t}\mathbf{s}_{t-1} + \beta_{1,t}(\nabla f(\mathbf{x}_{t-1}) - \nabla f(\mathbf{x}_t))}_{\mathbf{z}_t} + (1 - \beta_{1,t})(\mathbf{g}_t - \nabla f(\mathbf{x}_t)).$$

Then, we have:

$$
\begin{aligned}
\mathbb{E}\|\mathbf{z}_t\|_2^2 &= \|\beta_{1,t}\mathbf{s}_{t-1} + \beta_{1,t}(\nabla f(\mathbf{x}_{t-1}) - \nabla f(\mathbf{x}_t))\|_2^2 \\
&\overset{(\bullet)}{\leq} (1+\beta)\beta_{1,t}^2\|\mathbf{s}_{t-1}\|_2^2 + \left(1+\frac{1}{\beta}\right)\beta_{1,t}^2\|\nabla f(\mathbf{x}_{t-1}) - \nabla f(\mathbf{x}_t))\|_2^2 \\
&\overset{(\circ)}{\leq} (2-\beta_{1,t})\beta_{1,t}^2\|\mathbf{s}_{t-1}\|_2^2 + \left(1+\frac{1}{1-\beta_{1,t}}\right)\beta_{1,t}^2\|\nabla f(\mathbf{x}_{t-1}) - \nabla f(\mathbf{x}_t))\|_2^2 \\
&\overset{(\star)}{\leq} (2\beta_{1,t}-\beta_{1,t}^2)\beta_{1,t}\|\mathbf{s}_{t-1}\|_2^2 + \frac{2\beta_{1,t}-\beta_{1,t}^2}{1-\beta_{1,t}}\beta_{1,t}L^2\|\mathbf{x}_{t-1}-\mathbf{x}_t\|_2^2 \\
&\overset{(*)}{=} \beta_{1,t}\|\mathbf{s}_{t-1}\|_2^2 + \frac{L^2}{1-\beta_{1,t}}\|\mathbf{x}_{t-1}-\mathbf{x}_t\|_2^2,
\end{aligned}
$$

where $(\bullet)$ is due to Young's inequality for any $\beta > 0$; $(\circ)$ follows from setting $\beta = 1 - \beta_{1,t}$; $(\star)$ is due to Assumption 4.2; and $(*)$ results from the fact that $2\beta_{1,t} - \beta_{1,t}^2 - 1 = -(\beta_{1,t}-1)^2 \leq 0$, which implies $2\beta_{1,t} - \beta_{1,t}^2 \leq 1$.

Therefore, we have:

$$
\begin{aligned}
\mathbb{E}\|\mathbf{s}_t\|_2^2 &\overset{(\circ)}{=} \mathbb{E}\|(1-\beta_{1,t})(\mathbf{g}_t - \nabla f(\mathbf{x}_t))\|_2^2 + \mathbb{E}\|\mathbf{z}_t\|_2^2 \\
&\leq \sigma^2(1-\beta_{1,t})^2 + \beta_{1,t}\|\mathbf{s}_{t-1}\|_2^2 + \frac{L^2}{1-\beta_{1,t}}\|\mathbf{x}_{t-1}-\mathbf{x}_t\|_2^2 \\
&\leq \sigma^2(1-\beta_{1,t})^2 + \beta_{1,t}\|\mathbf{s}_{t-1}\|_2^2 + \frac{L^2}{(1-\beta_{1,t})\min(\mathbf{h}_{t-1})^2}\mathbb{E}\|\mathbf{h}_{t-1}\odot(\mathbf{x}_t-\mathbf{x}_{t-1})\|_2^2,
\end{aligned}
$$

where $(\circ)$ is due to Assumption 4.3 $\mathbb{E}[\nabla f(x;\xi)] = \nabla f(x)$. Then, we define

$$
A_t = \sigma^2(1-\beta_{1,t})^2 + \frac{L^2}{(1-\beta_{1,t})\min(\mathbf{h}_{t-1})^2}R_{t-1}^2.
$$

Therefore, we have

$$
S_t^2 \leq \beta_{1,t}S_{t-1}^2 + A_t.
$$

Then, using Lemma C.1 and $\beta_{1,t} = 1 - t^{-1/2}$, we obtain:

$$
\begin{aligned}
t^{-1/2}S_t^2 &\leq 2(S_t^2 - S_{t+1}^2 + A_{t+1}) \\
&= 2(S_t^2 - S_{t+1}^2) + 2\sigma^2(t+1)^{-1} + \frac{2L^2}{(t+1)^{-1/2}\min(\mathbf{h}_t)^2}R_t^2 \\
&\overset{(\circ)}{\leq} 2(S_t^2 - S_{t+1}^2) + \frac{2\sigma^2}{t+1} + 2\left(\frac{t+1}{t}\right)^{1/2}\frac{L^2R_t^2}{\rho^2 u_{\min}^2\sqrt{t}} \\
&\overset{(\star)}{\leq} 2(S_t^2 - S_{t+1}^2) + \frac{2\sigma^2}{t+1} + \frac{4L^2R_t^2}{\rho^2 u_{\min}^2\sqrt{t}},
\end{aligned}
\tag{8}
$$

where $(\circ)$ is due to Lemma C.2; $(\star)$ relies on $(\frac{t+1}{t})^{1/2} \leq 2^{1/2} \leq 2$.

Then, from Lemma C.3, it follows that:

$$
\begin{aligned}
\mathbb{E}[f(\mathbf{x}_{t+1})] - f(\mathbf{x}_t) &\leq -\left(\frac{1}{2\kappa^2\rho u_{\max}} - \frac{L}{\rho^2 u_{\min}^2}\right)\frac{\mathbb{E}\|\mathbf{r}_t\|_2^2}{\sqrt{t}} + \frac{\mathbb{E}\|\mathbf{s}_t\|_2^2}{2\sqrt{t}L} \\
&= -\left(\frac{1}{2\kappa^2\rho u_{\max}} - \frac{L}{\rho^2 u_{\min}^2}\right)\frac{R_t^2}{\sqrt{t}} + \frac{S_t^2}{2\sqrt{t}L}.
\end{aligned}
$$

Adding both sides by $\varepsilon_1 t^{-1} + \frac{t^{-1/2}}{2L} S_t^2$ yields:

$$\mathbb{E}[f(\mathbf{x}_{t+1})] - f(\mathbf{x}_t) + \varepsilon_1 t^{-1} + \frac{t^{-1/2}}{2L} S_t^2$$

$$\leq \left( -\frac{1}{2\kappa^2 \rho u_{\max}} + \frac{L}{\rho^2 u_{\min}^2} \right) \frac{R_t^2}{\sqrt{t}} + \varepsilon_1 t^{-1} + \frac{t^{-1/2}}{L} S_t^2$$

$$\overset{(\circ)}{\leq} \left( -\frac{1}{2\kappa^2 \rho u_{\max}} + \frac{L}{\rho^2 u_{\min}^2} \right) \frac{R_t^2}{\sqrt{t}} + \varepsilon_1 t^{-1} + \frac{2}{L}(S_t^2 - S_{t+1}^2) + \frac{2\sigma^2}{L(t+1)} + \frac{4L R_t^2}{\rho^2 u_{\min}^2 \sqrt{t}}$$

$$= - \underbrace{\left( \frac{1}{2\kappa^2 \rho u_{\max}} - \frac{5L}{\rho^2 u_{\min}^2} \right)}_{\triangleq \varepsilon_2} \frac{R_t^2}{\sqrt{t}} + \varepsilon_1 t^{-1} + \frac{2}{L}(S_t^2 - S_{t+1}^2) + \frac{2\sigma^2}{L(t+1)},$$

where $(\circ)$ is due to Inequality (8).

Using the definition of $P_t, \varepsilon_1, \varepsilon_2, \varepsilon_3$, we further derive:

$$\varepsilon_1 t^{-1} + \varepsilon_2 t^{-1/2} R_t^2 + \varepsilon_3 t^{-1/2} S_t^2 \leq P_t - P_{t+1} + \frac{2\sigma^2}{L}(t+1)^{-1}.$$

This leads to

$$\varepsilon_{\min} t^{-1/2}(t^{-1/2} + R_t^2 + S_t^2) \leq P_t - P_{t+1} + \frac{2\sigma^2}{L}(t+1)^{-1}. \tag{9}$$

Summing Inequality (9) over $t$ from $1$ to $T$ yields:

$$\varepsilon_{\min} \sum_{t=1}^{T} t^{-1/2} \cdot (t^{-1/2} + R_t^2 + S_t^2)$$

$$\leq \sum_{t=1}^{T} (P_t - P_{t+1} + 2\sigma^2 L^{-1}(t+1)^{-1})$$

$$\overset{(\circ)}{\leq} P_1 + 2\sigma^2 L^{-1} \cdot \log(T+1),$$

where $(\circ)$ is due to $\sum_{t=1}^{T} \frac{1}{t+1} \leq \log(T+1)$.

This further leads to

$$\sum_{t=1}^{T} R_t^2 + S_t^2 \leq \frac{P_1 + 2\sigma^2 L^{-1} \cdot \log(T+1)}{\varepsilon_{\min}} \cdot \sqrt{T} - \sum_{t=1}^{T} t^{-1/2}$$

$$\overset{(\circ)}{\leq} \frac{P_1 + 2\sigma^2 L^{-1} \cdot \log(T+1)}{\varepsilon_{\min}} \cdot \sqrt{T} - 2(\sqrt{T} - 1) = \mathcal{O}(\log(T)\sqrt{T}),$$

where $(\circ)$ is due to $\sum_{t=1}^{T} t^{-1/2} \geq 2(\sqrt{T} - 1)$.

$\square$

# D    Proofs of Theorem 4.1

*Proof.* First, we have the following inequalities:

$$\sum_{t=1}^{T}\|\nabla f(\mathbf{x}_{t+1})\|_2^2$$

$$=\sum_{t=1}^{T}\|\nabla f(\mathbf{x}_{t+1})-\mathbf{m}_t-\mathbf{h}_t\odot(\mathbf{x}_{t+1}-\mathbf{x}_t)\|_2^2$$

$$=\sum_{t=1}^{T}\|\nabla f(\mathbf{x}_{t+1})-\nabla f(\mathbf{x}_t)+\nabla f(\mathbf{x}_t)-\mathbf{m}_t-\mathbf{h}_t\odot(\mathbf{x}_{t+1}-\mathbf{x}_t)\|_2^2$$

$$=\sum_{t=1}^{T}\|\nabla f(\mathbf{x}_{t+1})-\nabla f(\mathbf{x}_t)-\mathbf{s}_t-\mathbf{h}_t\odot(\mathbf{x}_{t+1}-\mathbf{x}_t)\|_2^2$$

$$=\sum_{t=1}^{T}[\|\nabla f(\mathbf{x}_{t+1})-\nabla f(\mathbf{x}_t)\|_2^2+\|\mathbf{s}_t\|_2^2+\|\mathbf{h}_t\odot(\mathbf{x}_{t+1}-\mathbf{x}_t)\|_2^2+2\langle\mathbf{s}_t,\mathbf{h}_t\odot(\mathbf{x}_{t+1}-\mathbf{x}_t)\rangle$$

$$-2\langle\nabla f(\mathbf{x}_{t+1})-\nabla f(\mathbf{x}_t),\mathbf{s}_t\rangle-2\langle\nabla f(\mathbf{x}_{t+1})-\nabla f(\mathbf{x}_t),\mathbf{h}_t\odot(\mathbf{x}_{t+1}-\mathbf{x}_t)\rangle]$$

$$\overset{(\bullet)}{\leq}\sum_{t=1}^{T}3(\|\nabla f(\mathbf{x}_{t+1})-\nabla f(\mathbf{x}_t)\|_2^2+\|\mathbf{s}_t\|_2^2+\|\mathbf{r}_t\|_2^2)$$

$$\overset{(\circ)}{\leq}\sum_{t=1}^{T}3L^2\|\mathbf{x}_{t+1}-\mathbf{x}_t\|_2^2+3\|\mathbf{s}_t\|_2^2+3\|\mathbf{r}_t\|_2^2$$

$$\leq\sum_{t=1}^{T}\frac{3L^2}{(\min(\mathbf{h}_t))^2}\|\mathbf{r}_t\|_2^2+3\|\mathbf{s}_t\|_2^2+3\|\mathbf{r}_t\|_2^2,$$

where $(\bullet)$ is due to Young's inequality; $(\circ)$ is due to Assumption 4.2.

Let's take the expectation of both sides,

$$\sum_{t=1}^{T}\mathbb{E}\|\nabla f(\mathbf{x}_{t+1})\|_2^2\leq\sum_{t=1}^{T}\frac{3L^2}{(\min(\mathbf{h}_t))^2}R_t^2+3S_t^2+3R_t^2$$

$$\overset{(\bullet)}{\leq}\sum_{t=1}^{T}\frac{3L^2}{\rho^2tu_{\min}^2}R_t^2+3(R_t^2+S_t^2)$$

$$=\sum_{t=1}^{T}\frac{3L^2}{\rho^2tu_{\min}^2}R_t^2+3(R_t^2+S_t^2)$$

$$\overset{(\circ)}{\leq}\frac{3L^2\eta^2}{\rho^2\epsilon^2}\sum_{t=1}^{T}R_t^2+3\sum_{t=1}^{T}(R_t^2+S_t^2)$$

$$\leq\left(\frac{3L^2\eta^2}{\rho^2\epsilon^2}+3\right)\left(\sum_{t=1}^{T}R_t^2+S_t^2\right)$$

$$\overset{(\star)}{\leq}\frac{3L^2\eta^2+3\rho^2\epsilon^2}{\rho^2\epsilon^2}\left(\frac{f(\mathbf{x}_1)-f(\mathbf{x}^*)+2\sigma^2L^{-1}\log(T+1)}{\varepsilon_{\min}}\sqrt{T}-2(\sqrt{T}-1)\right)$$

$$=\mathcal{O}(\log(T)\sqrt{T})=\tilde{\mathcal{O}}(\sqrt{T}),$$

where $(\bullet)$ results from Lemma C.2; $(\circ)$ is due to $\frac{1}{t}\leq 1$; $(\star)$ follows from Lemma C.4. Thus, we have

$$\min_{t=1,\cdots,T}\mathbb{E}\|\nabla f(\mathbf{x}_{t+1})\|_2^2\leq\frac{1}{T}\sum_{t=1}^{T}\mathbb{E}\|\nabla f(\mathbf{x}_{t+1})\|_2^2\leq\frac{\tilde{\mathcal{O}}(T^{1/2})}{T}=\tilde{\mathcal{O}}(T^{-1/2}).$$

$\square$

# E  Probability Convergence Lemmas

This proof refers to the following literature [35, 37, 36, 38].

## E.1  Definition

Let $\mathbf{g}_s$ denote the stochastic gradient. We define the noise as $\xi_s = \mathbf{g}_s - \nabla f(\mathbf{x}_s)$ and the coordinate-wise noise as $\xi_{s,i} = \mathbf{g}_{s,i} - \nabla f(\mathbf{x}_s)_i$. Furthermore, we define two auxiliary sequences $\{\mathbf{p}_s\}_{s \geq 1}$ and $\{\mathbf{y}_s\}_{s \geq 1}$.

$$
\begin{aligned}
\mathbf{p}_1 = \mathbf{0}_d, \mathbf{y}_1 = \mathbf{x}_1, \mathbf{p}_s &= \frac{\beta_1}{1 - \beta_1}(\mathbf{x}_s - \mathbf{x}_{s-1}), \\
\mathbf{y}_s &= \mathbf{p}_s + \mathbf{x}_s, \forall s \geq 2, \\
\xi_s = \mathbf{g}_s - \nabla f(\mathbf{x}_s), \xi_{s,i} &= \mathbf{g}_{s,i} - \nabla f(\mathbf{x}_s)_i.
\end{aligned}
\tag{10}
$$

Then by the definition of Assumption 4.4 , we continue to define useful notations:

$$
\begin{aligned}
D_T &= \sqrt{\log\left(\frac{eT}{\delta}\right)}, \quad G_s = \max_{j \in [s]} \|\nabla f(\mathbf{x}_j)\|, \\
G_T(s) &= D_T \sqrt{\|\sigma\|^2 + 2G_s^2}, \quad G_T = D_T \sqrt{\|\sigma\|^2 + 2G^2}, \\
\hat{\mathbf{m}}_s &= \frac{\mathbf{m}_s}{1 - \beta_1^s}, \hat{\mathbf{v}}_s = \frac{\mathbf{v}_s}{1 - \beta_2^s}, \\
\mathbf{b}_s &= \sqrt{\mathbf{v}_s} + \epsilon = \sqrt{\beta_2 \mathbf{v}_{s-1} + (1 - \beta_2)\mathbf{g}_s^2} + \epsilon, \\
\mathbf{a}_s &= \sqrt{\tilde{\mathbf{v}}_s} + \epsilon = \sqrt{\beta_2 \mathbf{v}_{s-1} + (1 - \beta_2)(G_T(s)\mathbf{1}_d)^2} + \epsilon.
\end{aligned}
\tag{11}
$$

It is noted that $y_t$ can also be expressed in the following form:

$$
\mathbf{y}_{s+1} = \mathbf{y}_s - \eta_s \phi_s \odot \frac{\mathbf{g}_s}{\mathbf{b}_s} + \frac{\beta_1}{1 - \beta_1}\left(\frac{\eta_s \mathbf{b}_{s-1} \odot \phi_s}{\eta_{s-1}\mathbf{b}_s \odot \phi_{s-1}} - \mathbf{1}_d\right) \odot (\mathbf{x}_s - \mathbf{x}_{s-1}).
\tag{12}
$$

Without loss of generality, for $\phi_s \in \{\alpha, \gamma\}$, we set $\alpha = 1$ and $\gamma \in (0, 1)$.

Then, we define $\Delta_s = \left(\frac{\eta_s b_{s-1} \odot \phi_s}{\eta_{s-1} b_s \odot \phi_{s-1}} - \mathbf{1}_d\right)$.

## E.2  Lemma E.1

**Lemma E.1.** *Suppose that $\{\alpha_s\}_{s \geq 1}$ is a real number sequence. Given $0 \leq \beta_1 < \beta_2 \leq 1, \epsilon > 0$, we define $c_s = \sum_{j=1}^{s} \beta_1^{s-j}\alpha_j, d_s = \frac{1}{1-\beta_1^s}\sum_{j=1}^{s}\beta_1^{s-j}\alpha_j$ and $e_s = \sum_{j=1}^{s}\beta_2^{s-j}\alpha_j^2$, then*

$$
\sum_{s=1}^{t} \frac{c_s^2}{\epsilon + e_s} \leq \frac{1}{(1 - \beta_1)(1 - \beta_1/\beta_2)}\left(\log\left(1 + \frac{e_t}{\epsilon}\right) - t\log\beta_2\right), \quad \forall t \geq 1,
$$

$$
\sum_{s=1}^{t} \frac{d_s^2}{\epsilon + e_s} \leq \frac{1}{(1 - \beta_1)^2(1 - \beta_1/\beta_2)}\left(\log\left(1 + \frac{e_t}{\epsilon}\right) - t\log\beta_2\right), \quad \forall t \geq 1.
$$

*Proof.* See the proof of [[35], Lemma A.2]. $\qquad\square$

## E.3  Lemma E.2

**Lemma E.2.** *Suppose $\{Z_s\}_{s \in [T]}$ is a martingale difference sequence with respect to $\zeta_1, \cdots, \zeta_T$. Assume that for each $s \in [T], \sigma_s$ is a random variable only dependent by $\zeta_1, \cdots, \zeta_T$ and satisfies that*

$$
\mathbb{E}\left[\exp(Z_s^2/\sigma_s^2) \mid \zeta_1, \cdots, \zeta_{s-1}\right] \leq e,
$$

*then for any $\lambda > 0$, and for any $\delta \in (0, 1)$, it holds that*

$$
\mathbb{P}\left(\sum_{s=1}^{T} Z_s > \frac{1}{\lambda}\log\left(\frac{1}{\delta}\right) + \frac{3}{4}\lambda\sum_{s=1}^{T}\sigma_s^2\right) \leq \delta.
$$

*Proof.* See the proof of [[36], Lemma 1]. □

## E.4 Lemma E.3

**Lemma E.3.** *Let $\mathbf{g}_s, \mathbf{m}_s, \hat{\mathbf{m}}_s$ be given in Algorithm 1 and $\mathbf{b}_s$, be defined in (11). If $0 \leq \beta_1 < \beta_2 < 1$ and $\mathcal{F}_i(t) = 1 + \frac{1}{\epsilon^2} \sum_{s=1}^{t} \mathbf{g}_{s,i}^2$ then $\forall t \geq 1$, we have,*

$$\sum_{s=1}^{t} \left\| \frac{\mathbf{g}_s}{\mathbf{b}_s} \right\|^2 \leq \frac{1}{1 - \beta_2} \sum_{i=1}^{d} \log\left( \frac{\mathcal{F}_i(t)}{\beta_2^t} \right),$$

$$\sum_{s=1}^{t} \left\| \frac{\mathbf{m}_s}{\mathbf{b}_s} \right\|^2 \leq \frac{1 - \beta_1}{(1 - \beta_2)(1 - \beta_1/\beta_2)} \sum_{i=1}^{d} \log\left( \frac{\mathcal{F}_i(t)}{\beta_2^t} \right),$$

$$\sum_{s=1}^{t} \left\| \frac{\mathbf{m}_s}{\mathbf{b}_{s+1}} \right\|^2 \leq \frac{1 - \beta_1}{\beta_2(1 - \beta_2)(1 - \beta_1/\beta_2)} \sum_{i=1}^{d} \log\left( \frac{\mathcal{F}_i(t)}{\beta_2^t} \right),$$

$$\sum_{s=1}^{t} \left\| \frac{\hat{\mathbf{m}}_s}{\mathbf{b}_s} \right\|^2 \leq \frac{1}{(1 - \beta_2)(1 - \beta_1/\beta_2)} \sum_{i=1}^{d} \log\left( \frac{\mathcal{F}_i(t)}{\beta_2^t} \right).$$

*Proof.* First, we have:

$$\epsilon^2 \geq \epsilon^2(1 - \beta_2^s) \geq \epsilon^2(1 - \beta_2).$$

Next, the following inequalities and equality hold:

$$\mathbf{b}_{s,i}^2 \geq \mathbf{v}_{s,i}^2 + \epsilon^2 \geq (1 - \beta_2)\left( \sum_{j=1}^{s} \beta_2^{s-j} \mathbf{g}_{j,i}^2 + \epsilon^2 \right), \quad \mathbf{m}_{s,i} = (1 - \beta_1) \sum_{j=1}^{s} \beta_1^{s-j} \mathbf{g}_{j,i}.$$

For the first expression, it follows that:

$$\sum_{s=1}^{t} \frac{\mathbf{g}_{s,i}^2}{\mathbf{b}_{s,i}^2} \leq \frac{1}{1 - \beta_2} \sum_{s=1}^{t} \frac{\mathbf{g}_{s,i}^2}{\epsilon^2 + \sum_{j=1}^{s} \beta_2^{s-j} \mathbf{g}_{j,i}^2}.$$

$$\overset{(\circ)}{\leq} \frac{1}{1 - \beta_2} \left[ \log\left( 1 + \frac{1}{\epsilon^2} \sum_{s=1}^{t} \beta_2^{t-s} \mathbf{g}_{s,i}^2 \right) - t \log \beta_2 \right]$$

$$\leq \frac{1}{1 - \beta_2} \log\left( \frac{\mathcal{F}_i(t)}{\beta_2^t} \right),$$

where $(\circ)$ is due to using Lemma E.1.

For the second expression, we have:

$$\sum_{s=1}^{t} \frac{\mathbf{m}_{s,i}^2}{\mathbf{b}_{s,i}^2} \leq \frac{(1 - \beta_1)^2}{1 - \beta_2} \cdot \sum_{s=1}^{t} \frac{\left( \sum_{j=1}^{s} \beta_1^{s-j} \mathbf{g}_{j,i} \right)^2}{\epsilon^2 + \sum_{j=1}^{s} \beta_2^{s-j} \mathbf{g}_{j,i}^2}$$

$$\overset{(\circ)}{\leq} \frac{(1 - \beta_1)^2}{1 - \beta_2} \cdot \frac{1}{(1 - \beta_1)(1 - \beta_1/\beta_2)} \left[ \log\left( 1 + \frac{1}{\epsilon^2} \sum_{s=1}^{t} \beta_2^{t-s} \mathbf{g}_{s,i}^2 \right) - t \log \beta_2 \right]$$

$$= \frac{1 - \beta_1}{(1 - \beta_2)(1 - \beta_1/\beta_2)} \log\left( \frac{\mathcal{F}_i(t)}{\beta_2^t} \right),$$

where $(\circ)$ is due to using Lemma E.1 and setting $\beta_2 < 1$.

For the third inequality, we derive:

$$\sum_{s=1}^{t} \frac{\mathbf{m}_{s,i}^2}{\mathbf{b}_{s+1,i}^2} \le \sum_{s=1}^{t} \frac{\left[(1-\beta_1)\sum_{j=1}^{s}\beta_1^{s-j}\mathbf{g}_{j,i}\right]^2}{\epsilon^2(1-\beta_2)+(1-\beta_2)\sum_{j=1}^{s+1}\beta_2^{s+1-j}\mathbf{g}_{j,i}^2}$$

$$= \sum_{s=1}^{t} \frac{(1-\beta_1)^2\left(\sum_{j=1}^{s}\beta_1^{s-j}\mathbf{g}_{j,i}\right)^2}{\epsilon^2(1-\beta_2)+(1-\beta_2)\beta_2\sum_{j=1}^{s}\beta_2^{s-j}\mathbf{g}_{j,i}^2}$$

$$= \frac{(1-\beta_1)^2}{(1-\beta_2)\beta_2}\cdot\sum_{s=1}^{t}\frac{\left(\sum_{j=1}^{s}\beta_1^{s-j}\mathbf{g}_{j,i}\right)^2}{\frac{\epsilon^2}{\beta_2}+\sum_{j=1}^{s}\beta_2^{s-j}\mathbf{g}_{j,i}^2}$$

$$\overset{(\circ)}{\le} \frac{(1-\beta_1)^2}{(1-\beta_2)\beta_2}\cdot\frac{1}{(1-\beta_1)(1-\beta_1/\beta_2)}\left[\log\left(1+\frac{\beta_2}{\epsilon^2}\sum_{s=1}^{t}\beta_2^{t-s}\mathbf{g}_{s,i}^2\right)-t\log\beta_2\right]$$

$$\le \frac{1-\beta_1}{\beta_2(1-\beta_2)(1-\beta_1/\beta_2)}\log\left(\frac{\mathcal{F}_i(t)}{\beta_2^t}\right),$$

where $(\circ)$ is due to using Lemma E.1 and setting $\beta_2 < 1$.

For the fourth inequality, we derive:

$$\sum_{s=1}^{t}\frac{\hat{\mathbf{m}}_{s,i}^2}{\mathbf{b}_{s,i}^2} \le \frac{(1-\beta_1)^2}{1-\beta_2}\cdot\sum_{s=1}^{t}\frac{\left(\frac{1}{1-\beta_1^s}\sum_{j=1}^{s}\beta_1^{s-j}\mathbf{g}_{j,i}\right)^2}{\epsilon^2+\sum_{j=1}^{s}\beta_2^{s-j}\mathbf{g}_{j,i}^2}$$

$$\le \frac{(1-\beta_1)^2}{1-\beta_2}\cdot\frac{1}{(1-\beta_1)^2(1-\beta_1/\beta_2)}\left[\log\left(1+\frac{1}{\epsilon^2}\sum_{s=1}^{t}\beta_2^{t-s}\mathbf{g}_{s,i}^2\right)-t\log\beta_2\right]$$

$$\le \frac{1}{(1-\beta_2)(1-\beta_1/\beta_2)}\log\left(\frac{\mathcal{F}_i(t)}{\beta_2^t}\right),$$

where $(\circ)$ is due to using Lemma E.1 and setting $\beta_2 \le 1$.

$\square$

## E.5   Lemma E.4

**Lemma E.4.** *Let* $\eta_s, \eta_{s-1}, \gamma, b_s, \mathbf{b}_{s-1}$ *be given in Algorithm 1 and (10), then we have*

$$\left|\frac{\eta_s\mathbf{b}_{s-1,i}\phi_{s,i}}{\eta_{s-1}\mathbf{b}_{s,i}\phi_{s-1,i}}-1\right|\le\omega,\quad\forall t\ge 2,$$

*where* $\omega = c_0\sqrt{\frac{1+\beta_2}{\beta_2}}+1, c_0 = \max\{1,\gamma,1/\gamma\}$

*Proof.* To proceed with the proof, we first establish the following. For all $t \ge 2$, given the conditions $0 \le 1-\beta_1^{s-1} < 1-\beta_1^s$ and $\frac{\beta_2^{s-1}}{1-\beta_2^{s-1}} \le \frac{\beta_2}{1-\beta_2}$, it follows that:

$$\frac{\eta_s}{\eta_{s-1}} = \sqrt{\frac{1-\beta_2^s}{1-\beta_2^{s-1}}\cdot\frac{1-\beta_1^{s-1}}{1-\beta_1^s}}$$

$$\le \sqrt{1+\frac{\beta_2^{s-1}(1-\beta_2)}{1-\beta_2^{s-1}}} \le \sqrt{1+(1-\beta_2)\cdot\frac{\beta_2}{1-\beta_2}} = \sqrt{1+\beta_2}.$$

Next, We have:

$$\frac{\mathbf{b}_{s-1,i}}{\mathbf{b}_{s,i}} = \frac{\epsilon+\sqrt{\mathbf{v}_{s-1,i}}}{\epsilon+\sqrt{\beta_2\mathbf{v}_{s-1,i}+(1-\beta_2)\mathbf{g}_{s,i}^2}} \le \frac{\epsilon+\sqrt{\mathbf{v}_{s-1,i}}}{\epsilon+\sqrt{\beta_2\mathbf{v}_{s-1,i}}} \le \frac{1}{\sqrt{\beta_2}}.$$

For $\phi_{s-1,i}, \phi_{s,i} \in \{1, \gamma\}$, the ratio satisfies $\frac{\phi_{s,i}}{\phi_{s-1,i}} \leq \max\left\{1, \frac{1}{\gamma}, \gamma\right\}$. Defining $c = \max\left\{1, \frac{1}{\gamma}, \gamma\right\}$, it follows that:

$$\left|\frac{\eta_s \mathbf{b}_{s-1,i} \phi_{s,i}}{\eta_{s-1} \mathbf{b}_{s,i} \phi_{s-1,i}} - 1\right| \leq \left|c \frac{\eta_s \mathbf{b}_{s-1,i}}{\eta_{s-1} \mathbf{b}_{s,i}}\right| + 1 \leq c\sqrt{\frac{1 + \beta_2}{\beta_2}} + 1.$$

$\square$

## E.6 Lemma E.5

**Lemma E.5.** *Let $\mathbf{m}_s, \mathbf{b}_s$ be given in Algorithm 1 and 11 with $0 \leq \beta_1 < \beta_2 < 1$, respectively. Then,*

$$\left\|\frac{\mathbf{m}_s}{\mathbf{b}_s}\right\|_\infty \leq \sqrt{\frac{(1 - \beta_1)(1 - \beta_1^s)}{(1 - \beta_2)(1 - \beta_1/\beta_2)}}, \quad \forall t \geq 1.$$

*Consequently, if $f$ is L-smooth and we set $\eta = C_0\sqrt{1 - \beta_2}$ for some constant $C_0 > 0$, then we have :*

$$\|\nabla f(\mathbf{x}_s)\| \leq \|\nabla f(\mathbf{x}_1)\| + LC_0 s\sqrt{\frac{d}{1 - \beta_1/\beta_2}}, \quad \forall t \geq 1.$$

*Proof.* First, we derive:

$$
\begin{aligned}
\left|\frac{\mathbf{m}_{s-1,i}}{\mathbf{b}_{s-1,i}}\right| &= \sqrt{\frac{(1 - \beta_1)^2 \left(\sum_{j=1}^{t-1} \beta_1^{s-1-j} \mathbf{g}_{j,i}\right)^2}{(1 - \beta_2) \sum_{j=1}^{s-1} \beta_2^{s-1-j} \mathbf{g}_{j,i}^2}} \\
&\overset{(\circ)}{\leq} \frac{1 - \beta_1}{\sqrt{1 - \beta_2}} \sqrt{\sum_{j=1}^{s-1} \beta_1^{s-1-j} \cdot \frac{\sum_{j=1}^{s-1} \beta_1^{s-1-j} \mathbf{g}_{j,i}^2}{\sum_{j=1}^{s-1} \beta_2^{s-1-j} \mathbf{g}_{j,i}^2}} \\
&= \frac{1 - \beta_1}{\sqrt{1 - \beta_2}} \sqrt{\sum_{j=1}^{s-1} \left(\frac{\beta_1}{\beta_2}\right)^{s-1-j} \cdot \frac{\beta_2^{s-1-j} \mathbf{g}_{j,i}^2}{\sum_{k=1}^{s-1} \beta_2^{s-1-k} \mathbf{g}_{k,i}^2}} \\
&\overset{(\star)}{\leq} \frac{1 - \beta_1}{\sqrt{1 - \beta_2}} \sqrt{\sum_{j=1}^{s-1} \beta_1^{s-1-j} \cdot \sum_{j=1}^{s-1} \left(\frac{\beta_1}{\beta_2}\right)^{s-1-j}} \\
&= \frac{1 - \beta_1}{\sqrt{1 - \beta_2}} \sqrt{\frac{1 - \beta_1^{s-1}}{1 - \beta_1} \cdot \frac{1 - \left(\frac{\beta_1}{\beta_2}\right)^{s-1}}{1 - \frac{\beta_1}{\beta_2}}} \\
&\leq \sqrt{\frac{(1 - \beta_1)(1 - \beta_1^{s-1})}{(1 - \beta_2)\left(1 - \frac{\beta_1}{\beta_2}\right)}},
\end{aligned}
$$

where $(\circ)$ follows from applying Cauchy-Schwarz inequality, which gives us $\left(\sum_{j=1}^{s} \beta_1^{s-1-j} \mathbf{g}_{j,i}\right)^2 \leq \sum_{j=1}^{s} \beta_1^{s-1-j} \sum_{j=1}^{s} \beta_1^{s-1-j} \mathbf{g}_{j,i}^2$ ; $(\star)$ is due to $\frac{\beta_2^{s-1-j} \mathbf{g}_{j,i}^2}{\sum_{k=1}^{s-1} \beta_2^{s-1-k} \mathbf{g}_{k,i}^2} \leq 1$.

For the second conclusion, we have:

$$\|\nabla f(\mathbf{x}_s)\| \leq \|\nabla f(\mathbf{x}_s)\| + \|\nabla f(\mathbf{x}_s) - \nabla f(\mathbf{x}_{s-1})\| \leq \|\nabla f(\mathbf{x}_s)\| + L\|\mathbf{x}_s - \mathbf{x}_{s-1}\|.$$

Furthermore, it follows that:

$$\|\mathbf{x}_s - \mathbf{x}_{s-1}\| \leq \sqrt{d}\|\mathbf{x}_s - \mathbf{x}_{s-1}\|_\infty = \eta_{s-1}\sqrt{d}\left\|\frac{\phi_{s-1} \odot \mathbf{m}_{s-1}}{\mathbf{b}_{s-1}}\right\|_\infty$$

$$\leq \eta_{s-1}\sqrt{d}\left\|\frac{\mathbf{m}_{s-1}}{\mathbf{b}_{s-1}}\right\|_\infty \leq \eta\sqrt{\frac{d}{(1 - \beta_2)\left(1 - \frac{\beta_1}{\beta_2}\right)}} = C_0\sqrt{\frac{d}{1 - \frac{\beta_1}{\beta_2}}}.$$

So,we obtain:

$$\|\nabla f(\mathbf{x}_s)\| \le \|\nabla f(\mathbf{x}_1)\| + LC_0\sqrt{\frac{d}{1-\frac{\beta_1}{\beta_2}}} \le \|\nabla f(\mathbf{x}_1)\| + LC_0 s\sqrt{\frac{d}{1-\frac{\beta_1}{\beta_2}}}.$$

$\square$

## E.7 Lemma E.6

**Lemma E.6.** *Suppose that $f$ is $L$-smooth and Assumption 4.1 holds, then for any $\mathbf{x} \in \mathbb{R}^d$, we have*

$$\|\nabla f(\mathbf{x})\|^2 \le 2L(f(\mathbf{x}) - f^*).$$

*If $\eta = C_0\sqrt{1-\beta_2}$, $0 \le \beta_1 < \beta_2 < 1$, let any $\mathbf{x}_t, \mathbf{y}_t$ be defined in (10), then we have*

$$\|\nabla f(\mathbf{x}_t)\|^2 \le 2\|\nabla f(\mathbf{y}_t)\| + \frac{2L^2 C_0^2 d}{(1-\beta_1)^2(1-\beta_1/\beta_2)}, \quad \forall s \ge 1.$$

*Proof.* For the first conclusion, we define $\hat{\mathbf{x}} = \mathbf{x} - \frac{1}{L}\nabla f(\mathbf{x})$. According to the descent Lemma for $L$-smooth functions, we have:

$$f(\hat{\mathbf{x}}) \le f(\mathbf{x}) + \langle \nabla f(\mathbf{x}), \hat{\mathbf{x}} - \mathbf{x}\rangle + \frac{L}{2}\|\hat{\mathbf{x}} - \mathbf{x}\|^2 \le f(\mathbf{x}) - \frac{1}{2L}\|\nabla f(\mathbf{x})\|^2.$$

Rearranging the terms and noting that $f(\hat{\mathbf{x}}) \ge f^*$, where $f^*$ denotes the optimal value of $f$, it follows that:

$$\|\nabla f(\mathbf{x})\|^2 \le 2L\left(f(\mathbf{x}) - f(\hat{\mathbf{x}})\right) \le 2L\left(f(\mathbf{x}) - f^*\right).$$

For the second conclusion, utilizing the norm inequality and the $L$-smoothness property, we obtain:

$$\begin{aligned}
\|\nabla f(\mathbf{x}_t)\|^2 &\le 2\|\nabla f(\mathbf{y}_t)\|^2 + 2\|\nabla f(\mathbf{x}_t) - \nabla f(\mathbf{y}_t)\|^2 \\
&\le 2\|\nabla f(\mathbf{y}_t)\|^2 + 2L^2\|\mathbf{y}_t - \mathbf{x}_t\|^2 \\
&= 2\|\nabla f(\mathbf{y}_t)\| + \frac{2L^2\beta_1^2}{(1-\beta_1)^2}\|\mathbf{x}_t - \mathbf{x}_{t-1}\|^2 \\
&\le 2\|\nabla f(\mathbf{y}_t)\|^2 + 2\left(\frac{L\beta_1\sqrt{d}\eta_{t-1}}{1-\beta_1}\right)^2 \left\|\frac{\mathbf{m}_{t-1}}{\mathbf{b}_{t-1}}\right\|_\infty^2 \\
&\overset{(\star)}{\le} 2\|\nabla f(\mathbf{y}_t)\|^2 + \frac{2L^2 C_0^2 d}{(1-\beta_1)^2(1-\beta_1/\beta_2)},
\end{aligned}$$

where $(\circ)$ is due to Young's inequality; $(\star)$ relies on the inequality $\eta_t \le \frac{\eta}{1-\beta_1} \le \frac{C_0\sqrt{1-\beta_2}}{1-\beta_1}$, followed by Lemma E.5.

$\square$

## E.8 Lemma E.7

**Lemma E.7.** *Given $T \ge 1$, suppose that for any $s \in [T]$, coordinate-wise $\xi_{s,i} = \mathbf{g}_{s,i} - \nabla f(\mathbf{x}_s)_i$ satisfies Assumption 4.4. Then for any given $\delta \in (0,1)$, it holds that with probability at least $1 - \delta$,*

$$\xi_{s,i}^2 \le D_T^2 \sigma_i^2, \quad \forall s \in [T].$$

*Furthermore, we have*

$$\max_{j \in [s]} \|\xi_j\| \le G_T(s), \quad \max_{j \in [s]} \|\mathbf{g}_j\| \le G_T(s), \quad \max_{j \in [s]} \|\mathbf{v}_j\|_\infty \le (G_T(s))^2, \quad \forall s \in [T].$$

*Proof.* First, we define $\omega_{s,i} = \frac{\xi_{s,i}^2}{\sigma_i^2}$ for all $s \in [T]$. According to Assumption 4.4, taking the full expectation yields:

$$\mathbb{E}\left[\exp(\omega_{s,i})\right] \le \exp(1).$$

By applying the Markov inequality, for any $\delta \in (0, 1)$, we have:

$$\mathbb{P}\left(\max_{s\in[T]} \omega_{s,i} \geq \delta\right) = \mathbb{P}\left(\exp\left(\max_{s\in[T]} \omega_{s,i}\right) \geq \exp(\delta)\right)$$

$$\leq \exp(-\delta)\mathbb{E}\left[\exp\left(\max_{s\in[T]} \omega_{s,i}\right)\right]$$

$$\leq \exp(-\delta)\mathbb{E}\left[\sum_{s=1}^{T} \exp(\omega_{s,i})\right]$$

$$\leq \exp(-\delta)T\exp(1).$$

This implies that, with probability at least $1 - \delta$,

$$\xi_{s,i}^2 \leq \log\left(\frac{eT}{\delta}\right)\sigma_i^2 \quad \forall s \in [T].$$

Consequently, it follows that:

$$\|\xi_s\|^2 \leq D_T^2\left(\|\sigma\|^2 + 2G_s^2\right) \leq G_T^2(s),$$

where $D_T$ and $G_T(s)$ are appropriately defined constants or functions in 11.

Next, applying Young's inequality and given $D_T \geq 1$, for any $j \in [s]$, we obtain:

$$\|\mathbf{g}_j\|^2 \leq 2\|\nabla f(\mathbf{x}_j)\|^2 + 2\|\xi_j\|^2 \leq 2D_T^2\left(\|\sigma\|^2 + \|\nabla f(\mathbf{x}_j)\|^2\right) \leq (G_T(s))^2.$$

Finally, we employ mathematical induction to establish the concluding result. For any $i \in [d]$, note that the base case holds since:

$$\mathbf{v}_{1,i} = (1 - \beta_2)\mathbf{g}_{1,i}^2 \leq (G_T(s))^2.$$

Assume that for some $s' \in [s]$, the inequality $\mathbf{v}_{j,i} \leq (G_T(s))^2$ holds for all $j \in [s']$. Then, for the inductive step at $j = s' + 1$,

$$\mathbf{v}_{s'+1,i} = \beta_2\mathbf{v}_{s',i} + (1 - \beta_2)\mathbf{g}_{s',i}^2 \leq \beta_2(G_T(s))^2 + (1 - \beta_2)(G_T(s))^2 = (G_T(s))^2.$$

Thus, by induction, it follows that:

$$\mathbf{v}_{j,i} \leq (G_T(s))^2 \quad \forall j \in [s].$$

Since this inequality holds for all $i \in [d]$, we conclude that the desired result is obtained.

$\square$

## E.9 Lemma E.8

**Lemma E.8.** *Given $T \geq 1$. If $\mathbf{b}_s = (\mathbf{b}_{s,i})_i$ and $\mathbf{a}_s = (\mathbf{a}_{s,i})_i$ follow the definitions in 11, and Lemma E.7 holds, then for all $s \in [T], i \in [d], c \in \{1, \gamma, 1/\gamma\}, \gamma$ is given by Algorithm 1,*

$$\left|\frac{1}{\mathbf{a}_{s,i}} - \frac{1}{\mathbf{b}_{s,i}}\right| \leq \frac{G_T(s)\sqrt{1 - \beta_2}}{\mathbf{a}_{s,i}\mathbf{b}_{s,i}},$$

$$\left|\frac{1}{\mathbf{a}_{s,i}} - \frac{1}{\mathbf{b}_{s-1,i}}\right| \leq \frac{(G_T(s) + \epsilon)\sqrt{1 - \beta_2}}{\mathbf{a}_{s,i}\mathbf{b}_{s-1,i}},$$

$$\left|\frac{1}{\mathbf{a}_{s,i}} - \frac{c}{\mathbf{b}_{s-1,i}}\right| \leq \frac{(G_T(s))\beta_3}{\mathbf{a}_{s,i}\mathbf{b}_{s-1,i}},$$

*where $\beta_3 = \frac{|c^2\beta_2 - 1| + |c^2 - 1|}{c\sqrt{1 - \beta_2}}$.*

*Proof.* First, we prove the first inequality:

$$\left|\frac{1}{\mathbf{a}_{s,i}} - \frac{1}{\mathbf{b}_{s,i}}\right| = \frac{\left|\sqrt{\mathbf{v}_{s,i}} - \sqrt{\tilde{v}_{s,i}}\right|}{\mathbf{a}_{s,i}\mathbf{b}_{s,i}} = \frac{1 - \beta_2}{\mathbf{a}_{s,i}\mathbf{b}_{s,i}} \frac{\left|\mathbf{g}_{s,i}^2 - (G_T(s))^2\right|}{\sqrt{\mathbf{v}_{s,i}} + \sqrt{\tilde{v}_{s,i}}}$$

$$\overset{(\circ)}{\leq} \frac{1 - \beta_2}{\mathbf{a}_{s,i}\mathbf{b}_{s,i}} \cdot \frac{(G_T(s))^2}{\sqrt{\mathbf{v}_{s,i}} + \sqrt{\beta_2\mathbf{v}_{s-1,i} + (1 - \beta_2)(G_T(s))^2}}$$

$$\overset{(\star)}{\leq} \frac{G_T(s)\sqrt{1 - \beta_2}}{\mathbf{a}_{s,i}\mathbf{b}_{s,i}},$$

where $(\circ)$ applies the result from Lemma E.7, $\mathbf{g}_{s,i}^2 \le \|\mathbf{g}_s\|^2 \le (G_T(s))^2$, and $(\star)$ is due to $\sqrt{\tilde{\mathbf{v}}_{s,i}} \ge \sqrt{1-\beta_2}G_T(s)$.

Next, we prove the second inequality using $\sqrt{a} - \sqrt{b} \le \sqrt{a-b}$ for $0 \le b \le a$:

$$
\left| \frac{1}{\mathbf{b}_{s-1,i}} - \frac{1}{\mathbf{a}_{s,i}} \right| = \frac{\left| \sqrt{\tilde{\mathbf{v}}_{s,i}} - \sqrt{\mathbf{v}_{s-1,i}} \right|}{\mathbf{b}_{s-1,i}\mathbf{a}_{s,i}}
$$

$$
\le \frac{1}{\mathbf{b}_{s-1,i}\mathbf{a}_{s,i}} \frac{(1-\beta_2)\left|(G_T(s))^2 - \mathbf{v}_{s-1,i}\right|}{\sqrt{\tilde{\mathbf{v}}_{s,i}} + \sqrt{\mathbf{v}_{s-1,i}}}
$$

$$
\overset{(\circ)}{\le} \frac{1}{\mathbf{b}_{s-1,i}\mathbf{a}_{s,i}} \cdot \frac{(1-\beta_2)(G_T(s))^2}{\sqrt{\tilde{\mathbf{v}}_{s,i}} + \sqrt{\mathbf{v}_{s-1,i}}}
$$

$$
\overset{(\star)}{\le} \frac{G_T(s)\sqrt{1-\beta_2}}{\mathbf{b}_{s-1,i}\mathbf{a}_{s,i}},
$$

where $(\circ)$ is due to the fact that $\mathbf{v}_{s-1,i} \le (G_T(s))^2$, as stated in Lemma E.7; and $(\star)$ follows from the inequality $\sqrt{1-\beta_2}G_T^2(s) \le \tilde{v}_{s,i}$.

Finally, we prove the third inequality, similar to the proof of the second inequality:

$$
\left| \frac{c}{\mathbf{b}_{s-1,i}} - \frac{1}{\mathbf{a}_{s,i}} \right| = \frac{\left| c\sqrt{\tilde{\mathbf{v}}_{s,i}} - \sqrt{\mathbf{v}_{s-1,i}} \right|}{\mathbf{b}_{s-1,i}\mathbf{a}_{s,i}}
$$

$$
\le \frac{1}{\mathbf{b}_{s-1,i}\mathbf{a}_{s,i}} \frac{\left| c^2\tilde{\mathbf{v}}_{s,i} - \mathbf{v}_{s-1,i} \right|}{c\sqrt{\tilde{\mathbf{v}}_{s,i}} + \sqrt{\mathbf{v}_{s-1,i}}}
$$

$$
\le \frac{1}{\mathbf{b}_{s-1,i}\mathbf{a}_{s,i}} \frac{\left| c^2\beta_2(\mathbf{v}_{s-1,i} - G_T^2(s)) + (c^2-1)(G_T^2(s) - \mathbf{v}_{s-1,i}) \right|}{c\sqrt{\tilde{\mathbf{v}}_{s,i}} + \sqrt{\mathbf{v}_{s-1,i}}}
$$

$$
\le \frac{1}{\mathbf{b}_{s-1,i}\mathbf{a}_{s,i}} \frac{(|c^2\beta_2 - 1| + |c^2 - 1|)G_T^2(s)}{c\sqrt{1-\beta_2}G_T(s)}.
$$

Let $\beta_3 = \frac{|c^2\beta_2-1|+|c^2-1|}{c\sqrt{1-\beta_2}}$, then we have:

$$
\left| \frac{c}{\mathbf{b}_{s-1,i}} - \frac{1}{\mathbf{a}_{s,i}} \right| \le \frac{\beta_3 G_T(s)}{\mathbf{b}_{s-1,i}\mathbf{a}_{s,i}}.
$$

This completes the proof of all three inequalities. $\qquad\square$

### E.10   Lemma E.9

**Lemma E.9.** *Given $T \ge 1$ and $\delta \in (0,1)$. If Assumptions 4.4 holds, then for any $\beta > 0, \lambda = \frac{2(1-\beta_1)\sqrt{1-\beta_2}}{3\eta G_T \beta}$, with probability at least $1-\delta$,*

$$
A.1.1 \le \frac{1}{2\beta}\sum_{s=1}^{t}\eta_s \left\| \frac{\nabla f(\mathbf{x}_s)}{\sqrt{\mathbf{a}_s}} \right\|^2 + \frac{d}{\lambda}\log(\frac{T}{\delta})
$$

$$
A.1.2 \le \frac{1}{2\beta}\sum_{s=1}^{t}\eta_s \left\| \frac{\nabla f(\mathbf{x}_s)}{\sqrt{\mathbf{a}_s}} \right\|^2 + \frac{\eta\beta G_T(t)\sqrt{1-\beta_2}}{2(1-\beta_1)}\sum_{s=1}^{t}\left\| \frac{\mathbf{g}_s}{\mathbf{b}_s} \right\|^2.
$$

*Proof.* Recalling the definitions of $a_s$ and $G_T(s)$, for any $s \in [T]$ and $i \in [d]$, we have:

$$
\frac{1}{\mathbf{a}_{s,i}} \le \frac{1}{(G_T(s)+\epsilon)\sqrt{1-\beta_2}}
$$

$$
\le \frac{1}{G_T(s)\sqrt{1-\beta_2}} \le \frac{1}{\sigma_i\sqrt{1-\beta_2}}.
$$

Then, we define:

$$
\hat{X}_{s,i} = -\frac{\eta_s\phi_{s,i}\nabla f(\mathbf{x}_s)_i\xi_{s,i}}{\mathbf{a}_{s,i}}, \quad X_{s,i} = -\frac{\eta_s\nabla f(\mathbf{x}_s)_i\xi_{s,i}}{\mathbf{a}_{s,i}}, \quad w_{s,i} = \frac{\eta_s\nabla f(\mathbf{x}_s)_i}{\mathbf{a}_{s,i}}\sigma_i,
$$

where $\nabla f(\mathbf{x}_s)_i$ and $\mathbf{a}_{s,i}$ are measurable with respect to $\mathcal{F}_{s-1,i} = \sigma(\xi_{1,i} \cdots, \xi_{s-1,i})$ and $\xi_{s,i}$ is the noise at step $s$. Thus:

$$\mathcal{F}_{s,i} = \sigma(\xi_{1,i} \cdots, \xi_{s,i}).$$

Next, applying the Cauchy-Schwarz inequality, we obtain:

$$\left\langle \nabla f(\mathbf{x}_s), \frac{\phi_s \odot \xi_s}{a_s} \right\rangle^2 \leq \left\langle \nabla f(\mathbf{x}_s), \frac{\xi_s}{a_s} \right\rangle^2 \leq \left\| \frac{\nabla f(\mathbf{x}_s)}{a_s} \right\|^2 \|\xi_s\|^2.$$

Given Assumption 4.4:$\mathbb{E}_\xi[\nabla f(\mathbf{x}; \xi)] = \nabla f(\mathbf{x})$, and:

$$(\nabla f(\mathbf{x}; \xi)_i - \nabla f(\mathbf{x})_i)^2 \leq \sigma_i^2,$$

almost surely, it follows that:

$$\mathbb{E}\left[ \exp\left( \frac{\hat{X}_{s,i}^2}{w_{s,i}^2} \right) \mid \mathcal{F}_{s-1,i} \right] \leq \mathbb{E}\left[ \exp\left( \frac{X_{s,i}^2}{w_{s,i}^2} \right) \mid \mathcal{F}_{s-1,i} \right]$$

$$\leq \mathbb{E}\left[ \exp\left( \frac{\eta_s \nabla f(\mathbf{x}_s)_i \xi_{s,i}^2}{\eta_s \nabla f(\mathbf{x}_s)_i \sigma_i^2} \right) \mid \mathcal{F}_{s-1,i} \right] \leq \exp(1).$$

Then, by invoking Lemma E.2, we derive:

$$\sum_{s=1}^t \hat{X}_{s,i} \leq \frac{3\lambda}{4} \sum_{s=1}^t w_{s,i}^2 + \frac{1}{\lambda} \log\left( \frac{T}{\delta} \right)$$

$$= \frac{3\lambda}{4} \sum_{s=1}^t \frac{\eta_s^2 \nabla f(\mathbf{x}_s)_i^2}{\mathbf{a}_{s,i}^2} \sigma_i^2 + \frac{1}{\lambda} \log\left( \frac{T}{\delta} \right)$$

$$\overset{(\circ)}{\leq} \frac{3\lambda\eta}{4(1-\beta_1)\sqrt{1-\beta_2}} \sum_{s=1}^t \frac{\eta_s \nabla f(\mathbf{x}_s)_i^2}{\mathbf{a}_{s,i}} \sigma_i + \frac{1}{\lambda} \log\left( \frac{T}{\delta} \right)$$

$$\overset{(\star)}{\leq} \frac{3\lambda\eta G_T(t)}{4(1-\beta_1)\sqrt{1-\beta_2}} \sum_{s=1}^t \frac{\eta_s \nabla f(\mathbf{x}_s)_i^2}{\mathbf{a}_{s,i}} + \frac{1}{\lambda} \log\left( \frac{T}{\delta} \right),$$

where $(\circ)$ follows from $\frac{1}{\mathbf{a}_{s,i}} \leq \frac{1}{\sigma_i \sqrt{1-\beta_2}}$ and $\eta_s \leq \frac{\eta}{1-\beta_1}$;$(\star)$ follows from $\sigma_i \leq G_T$.

Setting $\lambda = \frac{2(1-\beta_1)\sqrt{1-\beta_2}}{3\eta G_T \beta}$, we obtain:

$$A.1.1 = \sum_{s=1}^t -\eta_s \left\langle \nabla f(\mathbf{x}_s), \frac{\phi_s \odot \xi_s}{\mathbf{a}_s} \right\rangle \leq \frac{1}{2\beta} \sum_{s=1}^t \sum_{i=1}^d \frac{\eta_s \nabla f(\mathbf{x}_s)_i^2}{\mathbf{a}_{s,i}} + \frac{d}{\lambda} \log\left( \frac{T}{\delta} \right)$$

$$= \frac{1}{2\beta} \sum_{s=1}^t \eta_s \left\| \frac{\nabla f(\mathbf{x}_s)}{\sqrt{\mathbf{a}_s}} \right\|^2 + \frac{d}{\lambda} \log\left( \frac{T}{\delta} \right).$$

Next, we bound A.1.2 as follows:

$$A.1.2 = \sum_{s=1}^t \eta_s \left\langle \nabla f(\mathbf{x}_s), \left( \frac{1}{a_s} - \frac{1}{\mathbf{b}_s} \right) \phi_s \odot \mathbf{g}_s \right\rangle$$

$$\leq \sum_{i=1}^d \sum_{s=1}^t \eta_s \left| \frac{1}{\mathbf{a}_{s,i}} - \frac{1}{\mathbf{b}_{s,i}} \right| \cdot |\nabla f(\mathbf{x}_s)_i \mathbf{g}_{s,i}|$$

$$\overset{(\circ)}{\leq} \sum_{i=1}^d \sum_{s=1}^t \eta_s \cdot \frac{G_T(s)\sqrt{1-\beta_2}}{\mathbf{a}_{s,i}\mathbf{b}_{s,i}} \cdot |\nabla f(\mathbf{x}_s)_i \mathbf{g}_{s,i}|$$

$$\overset{(\star)}{\leq} \frac{1}{2\beta} \sum_{i=1}^d \sum_{s=1}^t \frac{\eta_s \nabla f(\mathbf{x}_s)_i^2}{\mathbf{a}_{s,i}} + \frac{(1-\beta_2)\beta}{2} \sum_{i=1}^d \sum_{s=1}^t \frac{(G_T(s))^2}{\mathbf{a}_{s,i}} \cdot \frac{\eta_s \mathbf{g}_{s,i}^2}{\mathbf{b}_{s,i}^2}$$

$$\overset{(\bullet)}{\leq} \frac{1}{2\beta} \sum_{s=1}^t \eta_s \left\| \frac{\nabla f(\mathbf{x}_s)}{\sqrt{\mathbf{a}_s}} \right\|^2 + \frac{\eta\beta G_T(t)\sqrt{1-\beta_2}}{2(1-\beta_1)} \sum_{s=1}^t \left\| \frac{\mathbf{g}_s}{\mathbf{b}_s} \right\|^2,$$

where $(\circ)$ follows from the result of Lemma E.8, $(\star)$ is due to Young's inequality, and $(\bullet)$ relies on $\frac{1}{\mathbf{a}_{s,i}} \leq \frac{1}{\sigma_i \sqrt{1-\beta_2}}$ and $\eta_s \leq \frac{\eta}{1-\beta_1}$. $\qquad \square$

### E.11 Lemma E.10

**Lemma E.10.** *Given $T \geq 1$, if Lemma E.7 holds, then for all $t \in [T]$,*

$$B.1 \leq \sum_{s=1}^{t} \frac{\eta_s}{\beta} \left\| \frac{\nabla f(\mathbf{x}_s)}{\sqrt{\mathbf{a}_s}} \right\|^2 + \sum_{s=1}^{t} \frac{\beta(G_T(t))\eta\sqrt{1-\beta_2}}{2(1-\beta_1)^3} \left\| \frac{\mathbf{m}_{s-1}}{\mathbf{b}_s} \right\|^2$$

$$+ \sum_{s=1}^{t} \frac{\beta G_T(t)\eta\beta_3^2}{2(1-\beta_1)^3} \left\| \frac{\mathbf{m}_{s-1}}{\mathbf{b}_s} \right\|^2 + \frac{2\sqrt{d}\eta G_t}{\sqrt{(1-\beta_1)^3(1-\beta_2)(1-\beta_1/\beta_2)}}.$$

*Proof.* Decompose $\Delta_s \odot (\mathbf{x}_s - \mathbf{x}_{s-1})$ as follows:

$$\Delta_s \odot (\mathbf{x}_s - \mathbf{x}_{s-1}) = \left( \frac{\eta_s \mathbf{b}_{s-1} \odot \phi_s}{\eta_{s-1} \mathbf{b}_s \odot \phi_{s-1}} - 1_d \right) \odot \left( \eta_{s-1} \frac{\mathbf{m}_{s-1} \odot \phi_{s-1}}{\mathbf{b}_{s-1}} \right)$$

$$= -\phi_s \odot \left( \frac{\eta_s}{\mathbf{b}_s} - \frac{\eta_s}{\mathbf{a}_s} \right) \odot \mathbf{m}_{s-1}$$

$$- \left( \frac{\eta_s \phi_s}{\mathbf{a}_s} - \frac{\eta_s \phi_{s-1}}{\mathbf{b}_{s-1}} \right) \odot \mathbf{m}_{s-1} - (\eta_s - \eta_{s-1}) \frac{\mathbf{m}_{s-1} \odot \phi_{s-1}}{\mathbf{b}_{s-1}}.$$

Then, we have

$$B.1 \leq \frac{\beta_1}{1-\beta_1} \cdot \left| \left\langle \Delta_s \odot \frac{\eta_{s-1} \mathbf{m}_{s-1} \odot \phi_{s-1}}{\mathbf{b}_{s-1}}, \nabla f(\mathbf{x}_s) \right\rangle \right|$$

$$= \frac{\beta_1}{1-\beta_1} \cdot \left| \left\langle \left( \frac{\eta_s \phi_s}{\mathbf{b}_s} - \frac{\eta_{s-1} \phi_{s-1}}{\mathbf{b}_{s-1}} \right) \odot \mathbf{m}_{s-1}, \nabla f(\mathbf{x}_s) \right\rangle \right|$$

$$\leq \underbrace{\frac{\beta_1}{1-\beta_1} \cdot \left| \left\langle \left( \frac{\eta_s}{\mathbf{b}_s} - \frac{\eta_s}{\mathbf{a}_s} \right) \odot \phi_s \odot \mathbf{m}_{s-1}, \nabla f(\mathbf{x}_s) \right\rangle \right|}_{B.1.1}$$

$$+ \underbrace{\frac{\beta_1}{1-\beta_1} \cdot \left| \left\langle \left( \frac{\eta_s \phi_s}{\mathbf{a}_s} - \frac{\eta_s \phi_{s-1}}{\mathbf{b}_{s-1}} \right) \odot \mathbf{m}_{s-1}, \nabla f(\mathbf{x}_s) \right\rangle \right|}_{B.1.2}$$

$$+ \underbrace{\frac{\beta_1}{1-\beta_1} \cdot \left| (\eta_{s-1} - \eta_s) \left\langle \frac{\phi_{s-1} \odot \mathbf{m}_{s-1}}{\mathbf{b}_{s-1}}, \nabla f(\mathbf{x}_s) \right\rangle \right|}_{B.1.3}.$$

For $B.1.1$:

$$\frac{\beta_1}{1-\beta_1} \left| \left\langle \left( \frac{\eta_s}{\mathbf{b}_s} - \frac{\eta_s}{\mathbf{a}_s} \right) \odot \phi_s \odot \mathbf{m}_{s-1}, \nabla f(\mathbf{x}_s) \right\rangle \right|$$

$$= \frac{\beta_1}{1-\beta_1} \sum_{i=1}^{d} \eta_s \left| \left( \frac{1}{\mathbf{b}_{s,i}} - \frac{1}{\mathbf{a}_{s,i}} \right) \phi_{s,i} \mathbf{m}_{s-1,i} \nabla f(\mathbf{x}_s)_i \right|$$

$$\leq \frac{\beta_1}{1-\beta_1} \sum_{i=1}^{d} \eta_s \left| \left( \frac{1}{\mathbf{b}_{s,i}} - \frac{1}{\mathbf{a}_{s,i}} \right) \mathbf{m}_{s-1,i} \nabla f(\mathbf{x}_s)_i \right|$$

$$\overset{(\circ)}{\leq} \sum_{i=1}^{d} \frac{\beta_1}{1-\beta_1} \cdot \frac{G_T(s)\eta_s\sqrt{1-\beta_2}}{\mathbf{a}_{s,i}\mathbf{b}_{s,i}} \cdot |\nabla f(\mathbf{x}_s)_i \mathbf{m}_{s-1,i}|$$

$$\overset{(\star)}{\leq} \sum_{i=1}^{d} \frac{\eta_s}{2\beta} \cdot \frac{\nabla f(\mathbf{x}_s)_i^2}{\mathbf{a}_{s,i}} + \frac{\beta\eta_s\beta_1^2(1-\beta_2)}{2(1-\beta_1)^2} \sum_{i=1}^{d} \frac{(G_T(s))^2}{\mathbf{a}_{s,i}} \cdot \frac{\mathbf{m}_{s-1,i}^2}{\mathbf{b}_{s,i}^2}$$

$$\leq \frac{\eta_s}{2\beta} \left\| \frac{\nabla f(\mathbf{x}_s)}{\sqrt{\mathbf{a}_s}} \right\|^2 + \frac{\beta(G_T(t)+\epsilon)\eta\sqrt{1-\beta_2}}{2(1-\beta_1)^3} \left\| \frac{\mathbf{m}_{s-1}}{\mathbf{b}_s} \right\|^2,$$

where $(\circ)$ follows from Lemma E.8, and $(\star)$ follows from Young's inequality.

For $B.1.2$, let $\frac{\phi_{s,i}}{\phi_{s-1,i}} = c_i$ and for any $c_i \in \{1, \gamma, 1/\gamma\}$, let

$$\beta_3 = \max\left\{\frac{1-\beta_2}{\sqrt{1-\beta_2}}, \frac{2-\gamma^2(1+\beta_2)}{\gamma\sqrt{1-\beta_2}}, \frac{|\beta_2-\gamma^2|+1-\gamma^2}{\gamma\sqrt{1-\beta_2}}\right\}.$$

Then, we derive:

$$\frac{\beta_1}{1-\beta_1}\left|\left\langle\left(\frac{\eta_s\phi_s}{\mathbf{b}_{s-1}} - \frac{\eta_s\phi_{s-1}}{\mathbf{a}_s}\right)\mathbf{m}_{s-1}, \nabla f(\mathbf{x}_s)\right\rangle\right|$$

$$= \frac{\beta_1}{1-\beta_1}\sum_{i=1}^d \eta_s\phi_{s-1,i}\left|\left(\frac{c_i}{\mathbf{b}_{s-1,i}} - \frac{1}{\mathbf{a}_{s,i}}\right)\mathbf{m}_{s-1,i}\nabla f(\mathbf{x}_s)_i\right|$$

$$\overset{(\circ)}{\leq} \sum_{i=1}^d \frac{\beta_1}{1-\beta_1}\cdot\frac{G_T(s)\eta_s\beta_3}{\mathbf{a}_{s,i}\mathbf{b}_{s-1,i}}\cdot|\nabla f(\mathbf{x}_s)_i\mathbf{m}_{s-1,i}|$$

$$\overset{(\star)}{\leq} \sum_{i=1}^d \frac{\eta_s}{2\beta}\cdot\frac{\nabla f(\mathbf{x}_s)_i^2}{\mathbf{a}_{s,i}} + \frac{\beta\eta_s\beta_1^2\beta_3^2}{2(1-\beta_1)^2}\sum_{i=1}^d \frac{(G_T(s))^2}{\mathbf{a}_{s,i}}\cdot\frac{\mathbf{m}_{s-1,i}^2}{\mathbf{b}_{s,i}^2}$$

$$\leq \frac{\eta_s}{2\beta}\left\|\frac{\nabla f(\mathbf{x}_s)}{\sqrt{\mathbf{a}_s}}\right\|^2 + \frac{\beta(G_T(t)+\epsilon)\eta\beta_3^2}{2(1-\beta_1)^3}\left\|\frac{\mathbf{m}_{s-1}}{\mathbf{b}_s}\right\|^2,$$

where $(\circ)$ follows from Lemma E.8 and the condition $\phi_i \leq 1$, and $(\star)$ is due to Young's inequality.

For $B.1.3$, we derive:

$$\frac{\beta_1}{1-\beta_1}\cdot\left|(\eta_{s-1}-\eta_s)\left\langle\frac{\phi_{s-1}\odot\mathbf{m}_{s-1}}{\mathbf{b}_{s-1}}, \nabla f(\mathbf{x}_s)\right\rangle\right|$$

$$= \frac{\beta_1}{1-\beta_1}\left|\eta\left(\frac{\sqrt{1-\beta_2^s}}{1-\beta_1^s} - \frac{\sqrt{1-\beta_2^s}}{1-\beta_1^s} + \frac{\sqrt{1-\beta_2^s}}{1-\beta_2^{s-1}} - \frac{\sqrt{1-\beta_2^s}}{1-\beta_2^{s-1}}\right)\left\langle\frac{\phi_{s-1}\odot\mathbf{m}_{s-1}}{\mathbf{b}_{s-1}}, \nabla f(\mathbf{x}_s)\right\rangle\right|.$$

Thus,

$$B.1.3 \leq \underbrace{\frac{\eta\beta_1\sqrt{1-\beta_2^s}}{1-\beta_1}\left|\left(\frac{1}{1-\beta_1^{s-1}} - \frac{1}{1-\beta_1^s}\right)\left\langle\nabla f(\mathbf{x}_s), \frac{\phi_{s-1}\odot\mathbf{m}_{s-1}}{\mathbf{b}_{s-1}}\right\rangle\right|}_{B.1.3.1}$$

$$+ \underbrace{\frac{\eta\beta_1}{(1-\beta_1)(1-\beta_1^{s-1})}\left|\left(\sqrt{1-\beta_2^{s-1}} - \sqrt{1-\beta_2^s}\right)\left\langle\nabla f(\mathbf{x}_s), \frac{\phi_{s-1}\odot\mathbf{m}_{s-1}}{\mathbf{b}_{s-1}}\right\rangle\right|}_{B.1.3.2}.$$

Note that $\|\nabla f(\mathbf{x}_s)\| \leq G_s \leq G_t$ for all $s \leq t$. Then, by applying the Cauchy-Schwarz inequality, Lemma E.5, and the condition $\phi_i \leq 1$, we have:

$$\sqrt{1-\beta_2^s}\left|\left\langle\nabla f(\mathbf{x}_s), \frac{\phi_{s-1}\odot\mathbf{m}_{s-1}}{\mathbf{b}_{s-1}}\right\rangle\right|$$

$$\leq \sqrt{1-\beta_2^s}\|\nabla f(\mathbf{x}_s)\|\left\|\frac{\mathbf{m}_{s-1}}{\mathbf{b}_{s-1}}\right\| \leq \sqrt{d}G_t\sqrt{\frac{(1-\beta_1)(1-\beta_1^{s-1})}{(1-\beta_2)(1-\beta_1/\beta_2)}}.$$

Therefore, summing $B.1.3.1$ over $s \in [t]$, using $\beta_1 \in [0, 1]$, and noting that $B.1.3.1$ vanishes when $s = 1$:

$$\sum_{s=1}^t B.1.3.1 \leq \frac{\sqrt{d}\eta G_t}{1-\beta_1}\cdot\sqrt{\frac{1-\beta_1}{(1-\beta_2)(1-\beta_1/\beta_2)}}\sum_{s=2}^t\left(\frac{1}{1-\beta_1^{s-1}} - \frac{1}{1-\beta_1^s}\right)$$

$$\leq \frac{\sqrt{d}\eta G_t}{\sqrt{(1-\beta_1)^3(1-\beta_2)(1-\beta_1/\beta_2)}}.$$

Similarly, since $\|\nabla f(\mathbf{x}_s)\| \le G_s \le G_t$ for all $s \le t$, and $1 - \beta_1^{s-1} \ge 1 - \beta_1$, and $\phi_i \le 1$, we have:

$$\frac{1}{1 - \beta_1^{s-1}} \left| \left\langle \nabla f(\mathbf{x}_s), \frac{\phi_{s-1} \odot \mathbf{m}_{s-1}}{\mathbf{b}_{s-1}} \right\rangle \right|$$

$$\le \frac{1}{1 - \beta_1^{s-1}} \|\nabla f(\mathbf{x}_s)\| \left\| \frac{\mathbf{m}_{s-1}}{\mathbf{b}_{s-1}} \right\| \le \sqrt{d} G_t \sqrt{\frac{1}{(1 - \beta_2)(1 - \beta_1/\beta_2)}}.$$

Thus, we have:

$$\sum_{s=1}^{t} B.1.3.2 \le \frac{\sqrt{d} \eta G_t}{1 - \beta_1} \cdot \sqrt{\frac{1}{(1 - \beta_2)(1 - \beta_1/\beta_2)}} \sum_{s=2}^{t} \left( \sqrt{1 - \beta_2^{s-1}} - \sqrt{1 - \beta_2^{s}} \right)$$

$$\le \frac{\sqrt{d} \eta G_t}{(1 - \beta_1)\sqrt{(1 - \beta_2)(1 - \beta_1/\beta_2)}} \le \frac{\sqrt{d} \eta G_t}{\sqrt{(1 - \beta_1)^3 (1 - \beta_2)(1 - \beta_1/\beta_2)}}.$$

Therefore, combining all terms, we obtain:

$$B.1 \le \sum_{s=1}^{t} \frac{\eta_s}{2\beta} \left\| \frac{\nabla f(\mathbf{x}_s)}{\sqrt{\mathbf{a}_s}} \right\|^2 + \sum_{s=1}^{t} \frac{\beta G_T(t) \eta \sqrt{1 - \beta_2}}{2(1 - \beta_1)^3} \left\| \frac{\mathbf{m}_{s-1}}{\mathbf{b}_s} \right\|^2 + \sum_{s=1}^{t} \frac{\eta_s}{2\beta} \left\| \frac{\nabla f(\mathbf{x}_s)}{\sqrt{\mathbf{a}_s}} \right\|^2$$

$$+ \sum_{s=1}^{t} \frac{\beta G_T(t) \eta \beta_3^2}{2(1 - \beta_1)^3} \left\| \frac{\mathbf{m}_{s-1}}{\mathbf{b}_s} \right\|^2 + \frac{2\sqrt{d} \eta G_t}{\sqrt{(1 - \beta_1)^3 (1 - \beta_2)(1 - \beta_1/\beta_2)}}$$

$$= \sum_{s=1}^{t} \frac{\eta_s}{\beta} \left\| \frac{\nabla f(\mathbf{x}_s)}{\sqrt{\mathbf{a}_s}} \right\|^2 + \sum_{s=1}^{t} \frac{\beta G_T(t) \eta \sqrt{1 - \beta_2}}{2(1 - \beta_1)^3} \left\| \frac{\mathbf{m}_{s-1}}{\mathbf{b}_s} \right\|^2$$

$$+ \sum_{s=1}^{t} \frac{\beta G_T(t) \eta \beta_3^2}{2(1 - \beta_1)^3} \left\| \frac{\mathbf{m}_{s-1}}{\mathbf{b}_s} \right\|^2 + \frac{2\sqrt{d} \eta G_t}{\sqrt{(1 - \beta_1)^3 (1 - \beta_2)(1 - \beta_1/\beta_2)}}.$$

$\square$

## E.12 Lemma E.11

**Lemma E.11.** *Let $\beta_2 = 1 - 1/T$, and $\mathcal{F}_i(t)$ be given in Lemma E.3. We have $\log\left(\frac{\mathcal{F}_i(T)}{\beta_2^T}\right) \sim \mathcal{O}(\log(T))$.*

*Proof.* First, we have

$$-\log \beta_2 = \log\left(\frac{1}{\beta_2}\right) \le \frac{1 - \beta_2}{\beta_2} = \frac{1/T}{1 - 1/T} \le \frac{2}{T}, \tag{13}$$

where we apply $\log(1/a) \le (1 - a)/a, \forall a \in (0, 1)$.

It follows that

$$\log\left(\frac{\mathcal{F}(T)}{\beta_2^T}\right) \le \log(\mathcal{F}(T)) + 2 \le \log(e^2 \mathcal{F}(T)). \tag{14}$$

According to the definition of $\mathcal{F}_i(t)$ in Lemma E.3, we have

$$\sum_{s=1}^{t} \beta_2^{t-s} \mathbf{g}_{s,i}^2 \le 2 \sum_{s=1}^{t} \left( \nabla f(\mathbf{x}_s)_i^2 + \xi_{s,i}^2 \right) \le 2 \sum_{s=1}^{t} \left( \sigma_i^2 + \nabla f(\mathbf{x}_s)_i^2 \right)$$

$$\le 2 \left( \|\sigma\|_\infty^2 t + \sum_{s=1}^{t} \|\nabla f(\mathbf{x}_s)\|_\infty^2 \right). \tag{15}$$

Thus, we obtain

$$
\mathcal{F}_i(t) = 1 + \frac{1}{\epsilon^2} \sum_{s=1}^{t} \beta_2^{t-s} \mathbf{g}_{s,i}^2
$$

$$
\overset{(\circ)}{\leq} 1 + \frac{2}{\epsilon} \left[ \left( \|\sigma\|_\infty^2 + \left( \|\nabla f(\mathbf{x}_1)\|_\infty + tLC_0 \sqrt{\frac{d}{1 - \beta_1/\beta_2}} \right)^2 \right) t \right] \tag{16}
$$

$$
\overset{(\star)}{\leq} 1 + \frac{2}{\epsilon} \left[ \left( \|\sigma\|_\infty^2 + 2\|\nabla f(\mathbf{x}_1)\|_\infty^2 \right) t + \frac{2L^2 C_0^2 d}{1 - \beta_1/\beta_2} t^3 \right],
$$

where $(\circ)$ follows from using Lemma E.5; $(\star)$ is due to $(a + b)^2 \leq 2a^2 + 2b^b$.

Therefore combining (13), (14) and (16), we arrive at $\log\left( \frac{\mathcal{F}_i(T)}{\beta_2^T} \right) \sim \mathcal{O}(\log(T))$. $\qquad \square$

## F   Proofs of Theorem 4.2

*Proof.* Applying the descent Lemma to the algorithm, we have

$$
f(\mathbf{y}_{s+1}) \leq f(\mathbf{y}_s) + \langle \nabla f(\mathbf{y}_s), \mathbf{y}_{s+1} - \mathbf{y}_s \rangle + \frac{L}{2} \|\mathbf{y}_{s+1} - \mathbf{y}_s\|^2
$$

$$
= f(\mathbf{y}_s) + \left\langle \nabla f(\mathbf{y}_s), -\eta_s \phi_s \odot \frac{\mathbf{g}_s}{\mathbf{b}_s} + \frac{\beta_1}{1 - \beta_1} \Delta_s \odot (\mathbf{x}_s - \mathbf{x}_{s-1}) \right\rangle
$$

$$
+ \frac{L}{2} \left\| -\eta_s \phi_s \odot \frac{\mathbf{g}_s}{\mathbf{b}_s} + \frac{\beta_1}{1 - \beta_1} \Delta_s \odot (\mathbf{x}_s - \mathbf{x}_{s-1}) \right\|^2
$$

$$
= f(\mathbf{y}_s) - \eta_s \left\langle \nabla f(\mathbf{y}_s), \phi_s \odot \frac{\mathbf{g}_s}{\mathbf{b}_s} \right\rangle + \frac{\beta_1}{1 - \beta_1} \langle \nabla f(\mathbf{y}_s), \Delta_s \odot (\mathbf{x}_s - \mathbf{x}_{s-1}) \rangle
$$

$$
+ \frac{L}{2} \left\| -\eta_s \phi_s \odot \frac{\mathbf{g}_s}{\mathbf{b}_s} + \frac{\beta_1}{1 - \beta_1} \Delta_s \odot (\mathbf{x}_s - \mathbf{x}_{s-1}) \right\|^2
$$

$$
\leq f(\mathbf{x}_1) + \sum_{s=1}^{t} -\eta_s \left\langle \nabla f(\mathbf{y}_s), \phi_s \odot \frac{\mathbf{g}_s}{\mathbf{b}_s} \right\rangle + \sum_{s=1}^{t} \frac{\beta_1}{1 - \beta_1} \langle \nabla f(\mathbf{y}_s), \Delta_s \odot (\mathbf{x}_s - \mathbf{x}_{s-1}) \rangle
$$

$$
+ \sum_{s=1}^{t} \frac{L}{2} \left\| -\eta_s \phi_s \odot \frac{\mathbf{g}_s}{\mathbf{b}_s} + \frac{\beta_1}{1 - \beta_1} \Delta_s \odot (\mathbf{x}_s - \mathbf{x}_{s-1}) \right\|^2.
$$

Then, we define

$$
A = \sum_{s=1}^{t} -\eta_s \left\langle \nabla f(\mathbf{y}_s), \phi_s \odot \frac{\mathbf{g}_s}{\mathbf{b}_s} \right\rangle,
$$

$$
B = \sum_{s=1}^{t} \frac{\beta_1}{1 - \beta_1} \langle \nabla f(\mathbf{y}_s), \Delta_s \odot (\mathbf{x}_s - \mathbf{x}_{s-1}) \rangle,
$$

$$
C = \sum_{s=1}^{t} \frac{L}{2} \left\| -\eta_s \phi_s \odot \frac{\mathbf{g}_s}{\mathbf{b}_s} + \frac{\beta_1}{1 - \beta_1} \Delta_s \odot (\mathbf{x}_s - \mathbf{x}_{s-1}) \right\|^2.
$$

Now, decomposing $A$ and $B$,

$$
A = \sum_{s=1}^{t} -\eta_s \left\langle \nabla f(y_s), \phi_s \odot \frac{\mathbf{g}_s}{\mathbf{b}_s} \right\rangle
$$

$$
= \underbrace{\sum_{s=1}^{t} -\eta_s \left\langle \nabla f(\mathbf{x}_s), \phi_s \odot \frac{\mathbf{g}_s}{\mathbf{b}_s} \right\rangle}_{A.1} + \underbrace{\sum_{s=1}^{t} \eta_s \left\langle \nabla f(\mathbf{x}_s) - \nabla f(\mathbf{y}_s), \phi_s \odot \frac{\mathbf{g}_s}{\mathbf{b}_s} \right\rangle}_{A.2}.
$$

$$B = \sum_{s=1}^{t} \frac{\beta_1}{1-\beta_1} \langle \nabla f(\mathbf{y}_s), \Delta_s \odot (\mathbf{x}_s - \mathbf{x}_{s-1}) \rangle = \underbrace{\frac{\beta_1}{1-\beta_1} \sum_{s=1}^{t} \langle \nabla f(\mathbf{x}_s), \Delta_s \odot (\mathbf{x}_s - \mathbf{x}_{s-1}) \rangle}_{B.1}$$

$$+ \underbrace{\frac{\beta_1}{1-\beta_1} \sum_{s=1}^{t} \langle \nabla f(\mathbf{y}_s) - \nabla f(\mathbf{x}_s), \Delta_s \odot (\mathbf{x}_s - \mathbf{x}_{s-1}) \rangle}_{B.2}.$$

Subsequently, using the conclusions of Lemma E.9 and Lemma E.8, we have

$$A.1 = -\sum_{s=1}^{t} \eta_s \left\| \frac{\sqrt{\phi_s} \odot \nabla f(\mathbf{x}_s)}{\sqrt{\mathbf{a}_s}} \right\|^2$$

$$\underbrace{-\sum_{s=1}^{t} \eta_s \left\langle \nabla f(\mathbf{x}_s), \frac{\phi_s \odot \xi_s}{\mathbf{a}_s} \right\rangle}_{A.1.1} + \underbrace{\sum_{s=1}^{t} \eta_s \left\langle \nabla f(\mathbf{x}_s), \left( \frac{1}{\mathbf{a}_s} - \frac{1}{\mathbf{b}_s} \right) \odot \phi_s \odot \mathbf{g}_s \right\rangle}_{A.1.2}.$$

$$A.1 \leq -\sum_{s=1}^{t} \eta_s \gamma \left\| \frac{\nabla f(\mathbf{x}_s)}{\sqrt{\mathbf{a}_s}} \right\|^2 + \frac{1}{2\beta} \sum_{s=1}^{t} \eta_s \left\| \frac{\nabla f(\mathbf{x}_s)}{\sqrt{\mathbf{a}_s}} \right\|^2 + \frac{d}{\lambda} \log \left( \frac{T}{\delta} \right)$$

$$+ \frac{1}{2\beta} \sum_{s=1}^{t} \eta_s \left\| \frac{\nabla f(\mathbf{x}_s)}{\sqrt{\mathbf{a}_s}} \right\|^2 + \frac{\eta \beta G_T(t) \sqrt{1-\beta_2}}{2(1-\beta_1)} \sum_{s=1}^{t} \left\| \frac{\mathbf{g}_s}{\mathbf{b}_s} \right\|^2$$

$$= \left( \frac{1}{\beta} - \gamma \right) \sum_{s=1}^{t} \eta_s \left\| \frac{\nabla f(\mathbf{x}_s)}{\sqrt{\mathbf{a}_s}} \right\|^2 + \frac{d}{\lambda} \log \left( \frac{T}{\delta} \right) + \frac{\eta \beta G_T(t) \sqrt{1-\beta_2}}{2(1-\beta_1)} \sum_{s=1}^{t} \left\| \frac{\mathbf{g}_s}{\mathbf{b}_s} \right\|^2.$$

For $A.2$, applying Young's inequality, we have

$$\eta_s \left\langle \nabla f(\mathbf{x}_s) - \nabla f(\mathbf{y}_s), \frac{\phi_s \odot \mathbf{g}_s}{\mathbf{b}_s} \right\rangle \overset{(\bullet)}{\leq} \eta_s \|\nabla f(\mathbf{x}_s) - \nabla f(\mathbf{y}_s)\| \cdot \left\| \frac{\mathbf{g}_s}{\mathbf{b}_s} \right\|$$

$$\overset{(\circ)}{\leq} \frac{1}{2L} \|\nabla f(\mathbf{x}_s) - \nabla f(\mathbf{y}_s)\|^2 + \frac{L\eta_s^2}{2} \left\| \frac{\mathbf{g}_s}{\mathbf{b}_s} \right\|^2$$

$$\leq \frac{L\beta_1^2}{2(1-\beta_1)^2} \|\mathbf{x}_s - \mathbf{x}_{s-1}\|^2 + \frac{L\eta^2}{2(1-\beta_1)^2} \left\| \frac{\mathbf{g}_s}{\mathbf{b}_s} \right\|^2$$

$$\overset{(\star)}{\leq} \frac{L\eta^2}{2(1-\beta_1)^2} \left\| \frac{\hat{\mathbf{m}}_{s-1}}{\mathbf{b}_{s-1}} \right\|^2 + \frac{L\eta^2}{2(1-\beta_1)^2} \left\| \frac{\mathbf{g}_s}{\mathbf{b}_s} \right\|^2,$$

where $(\bullet)$ is based on $\phi_i \leq 1$ and applying the Cauchy-Schwarz inequality; $(\circ)$ follows from applying Young's inequality; $(\star)$ is due to

$$\|\mathbf{x}_s - \mathbf{x}_{s-1}\|^2 \leq \eta_{s-1}^2 \left\| \frac{\mathbf{m}_{s-1}}{\mathbf{b}_{s-1}} \right\|^2 \leq \eta^2 \left\| \frac{\hat{\mathbf{m}}_{s-1}}{\mathbf{b}_{s-1}} \right\|^2.$$

Thus, summing over $s \in [t]$, we obtain:

$$A.2 \leq \frac{L\eta^2}{2(1-\beta_1)^2} \sum_{s=1}^{t} \left\| \frac{\hat{\mathbf{m}}_{s-1}}{\mathbf{b}_{s-1}} \right\|^2 + \frac{L\eta^2}{2(1-\beta_1)^2} \sum_{s=1}^{t} \left\| \frac{\mathbf{g}_s}{\mathbf{b}_s} \right\|^2.$$

For $(B.1)$, using Lemma E.10, we have

$$B.1 \leq \sum_{s=1}^{t} \frac{\eta_s}{\beta} \left\| \frac{\nabla f(\mathbf{x}_s)}{\sqrt{\mathbf{a}_s}} \right\|^2 + \sum_{s=1}^{t} \frac{\beta(G_T(t))\eta\sqrt{1-\beta_2}}{2(1-\beta_1)^3} \left\| \frac{\mathbf{m}_{s-1}}{\mathbf{b}_s} \right\|^2$$

$$+ \sum_{s=1}^{t} \frac{\beta G_T(t)\eta\beta_3^2}{2(1-\beta_1)^3} \left\| \frac{\mathbf{m}_{s-1}}{\mathbf{b}_s} \right\|^2 + \frac{2\sqrt{d}\eta G_t}{\sqrt{(1-\beta_1)^3(1-\beta_2)(1-\beta_1/\beta_2)}}.$$

For $B.2$, applying vector inequalities and Lemma E.4, we have

$$B.2 \leq \frac{\beta_1}{1-\beta_1} \sum_{s=1}^{t} \|\Delta_s\|_{\infty} \|\mathbf{x}_s - \mathbf{x}_{s-1}\| \|\nabla f(\mathbf{y}_s) - \nabla f(\mathbf{x}_s)\|$$

$$\leq \frac{L\beta_1^2 \omega}{(1-\beta_1)^2} \sum_{s=1}^{t} \|\mathbf{x}_s - \mathbf{x}_{s-1}\|^2$$

$$\leq \frac{L\omega^2 \eta^2}{(1-\beta_1)^2} \sum_{s=1}^{t} \left\| \frac{\hat{\mathbf{m}}_{s-1}}{\mathbf{b}_{s-1}} \right\|^2.$$

Finally, for the upper bound of $C$, we have

$$C \leq L \sum_{s=1}^{t} \eta_s^2 \left\| \frac{\mathbf{g}_s}{\mathbf{b}_s} \right\|^2 + \frac{L\beta_1^2}{(1-\beta_1)^2} \sum_{s=1}^{t} \|\Delta_s\|_{\infty}^2 \|\mathbf{x}_s - \mathbf{x}_{s-1}\|^2$$

$$\leq \frac{L\eta^2}{(1-\beta_1)^2} \sum_{s=1}^{t} \left\| \frac{\mathbf{g}_s}{\mathbf{b}_s} \right\|^2 + \frac{L\eta^2 \omega^2}{(1-\beta_1)^2} \sum_{s=1}^{t} \left\| \frac{\hat{\mathbf{m}}_{s-1}}{\mathbf{b}_{s-1}} \right\|^2.$$

Therefore, we define

$$C_1 = \frac{\eta\beta G_T\sqrt{1-\beta_2}}{2(1-\beta_1)} + \frac{3L\eta^2}{2(1-\beta_1)^2},$$

$$C_2 = \frac{L\eta^2}{2(1-\beta_1)^2} + \frac{2L\eta^2\omega^2}{(1-\beta_1)^2},$$

$$C_3 = \frac{\beta G_T\eta\sqrt{1-\beta_2}}{2(1-\beta_1)^3} + \frac{\beta G_T\eta\beta_3^2}{2(1-\beta_1)^3},$$

$$C_4 = \frac{2\sqrt{d}\eta}{\sqrt{(1-\beta_1)^3(1-\beta_2)(1-\beta_1/\beta_2)}},$$

$$C_5 = \frac{L^2 C_0^2 d}{(1-\beta_1)^2(1-\beta_1/\beta_2)}.$$

Then, we have

$$f(\mathbf{y}_{s+1}) \leq f(\mathbf{x}_1) + \left(\frac{1}{\beta} - \gamma\right) \sum_{s=1}^{t} \eta_s \left\| \frac{\nabla f(\mathbf{x}_s)}{\sqrt{\mathbf{a}_s}} \right\|^2 + \frac{d}{\lambda} \log\left(\frac{T}{\delta}\right) + \frac{\eta\beta G_T(t)\sqrt{1-\beta_2}}{2(1-\beta_1)} \sum_{s=1}^{t} \left\| \frac{\mathbf{g}_s}{\mathbf{b}_s} \right\|^2$$

$$+ \frac{L\eta^2}{2(1-\beta_1)^2} \sum_{s=1}^{t} \left\| \frac{\hat{\mathbf{m}}_{s-1}}{\mathbf{b}_{s-1}} \right\|^2 + \frac{L\eta^2}{2(1-\beta_1)^2} \sum_{s=1}^{t} \left\| \frac{\mathbf{g}_s}{\mathbf{b}_s} \right\|^2 + \frac{1}{\beta} \sum_{s=1}^{t} \eta_s \left\| \frac{\nabla f(\mathbf{x}_s)}{\sqrt{\mathbf{a}_s}} \right\|^2$$

$$+ \frac{\beta G_T(t)\eta\sqrt{1-\beta_2}}{2(1-\beta_1)^3} \sum_{s=1}^{t} \left\| \frac{\mathbf{m}_{s-1}}{\mathbf{b}_s} \right\|^2 + \frac{\beta G_T(t)\eta\beta_3^2}{2(1-\beta_1)^3} \sum_{s=1}^{t} \left\| \frac{\mathbf{m}_{s-1}}{\mathbf{b}_s} \right\|^2$$

$$+ \frac{2\sqrt{d}\eta G_t}{\sqrt{(1-\beta_1)^3(1-\beta_2)(1-\beta_1/\beta_2)}}$$

$$+ \frac{L\eta^2}{(1-\beta_1)^2} \sum_{s=1}^{t} \left\| \frac{\mathbf{g}_s}{\mathbf{b}_s} \right\|^2 + \frac{2L\eta^2\omega^2}{(1-\beta_1)^2} \sum_{s=1}^{t} \left\| \frac{\hat{\mathbf{m}}_{s-1}}{\mathbf{b}_{s-1}} \right\|^2$$

$$\leq \left(\frac{2}{\beta} - \gamma\right) \sum_{s=1}^{t} \eta_s \left\| \frac{\nabla f(\mathbf{x}_s)}{\sqrt{\mathbf{a}_s}} \right\|^2 + \frac{d}{\lambda} \log\left(\frac{T}{\delta}\right) + C_1 \sum_{s=1}^{t} \left\| \frac{\mathbf{g}_s}{\mathbf{b}_s} \right\|^2 + C_2 \sum_{s=1}^{t} \left\| \frac{\hat{\mathbf{m}}_{s-1}}{\mathbf{b}_{s-1}} \right\|^2$$

$$+ C_3 \sum_{s=1}^{t} \left\| \frac{\mathbf{m}_{s-1}}{\mathbf{b}_s} \right\|^2 + C_4 G_T.$$

Then, according to Lemma E.6, we have

$$\|\nabla f(\mathbf{x}_{s+1})\|^2 \leq 2\|\nabla f(\mathbf{y}_{s+1})\|^2 + 2C_5$$
$$\leq 4L(f(\mathbf{y}_{s+1}) - f^*) + 2C_5.$$

Thus, we have

$$\|\nabla f(\mathbf{x}_{s+1})\|^2 \leq 4L(f(\mathbf{x}_1) - f^*) + 4L\left(\frac{2}{\beta} - \gamma\right)\sum_{s=1}^{t}\eta_s\left\|\frac{\nabla f(\mathbf{x}_s)}{\sqrt{\mathbf{a}_s}}\right\|^2 + \frac{4Ld}{\lambda}\log\left(\frac{T}{\delta}\right) + 2C_5$$

$$+ 4LC_1\sum_{s=1}^{t}\left\|\frac{\mathbf{g}_s}{\mathbf{b}_s}\right\|^2 + 4LC_2\sum_{s=1}^{t}\left\|\frac{\hat{\mathbf{m}}_{s-1}}{\mathbf{b}_{s-1}}\right\|^2 + 4LC_3\sum_{s=1}^{t}\left\|\frac{\mathbf{m}_{s-1}}{\mathbf{b}_s}\right\|^2 + 4LC_4G_T.$$

Recall Lemma E.3, we have:

$$\sum_{s=1}^{t}\left\|\frac{\mathbf{g}_s}{\mathbf{b}_s}\right\|^2 \leq \frac{1}{1-\beta_2}\sum_{i=1}^{d}\log\left(\frac{\mathcal{F}_i(t)}{\beta_2^t}\right),$$

$$\sum_{s=1}^{t}\left\|\frac{\mathbf{m}_s}{\mathbf{b}_s}\right\|^2 \leq \frac{1-\beta_1}{(1-\beta_2)(1-\beta_1/\beta_2)}\sum_{i=1}^{d}\log\left(\frac{\mathcal{F}_i(t)}{\beta_2^t}\right),$$

$$\sum_{s=1}^{t}\left\|\frac{\mathbf{m}_s}{\mathbf{b}_{s+1}}\right\|^2 \leq \frac{1-\beta_1}{\beta_2(1-\beta_2)(1-\beta_1/\beta_2)}\sum_{i=1}^{d}\log\left(\frac{\mathcal{F}_i(t)}{\beta_2^t}\right),$$

$$\sum_{s=1}^{t}\left\|\frac{\hat{\mathbf{m}}_s}{\mathbf{b}_s}\right\|^2 \leq \frac{1}{(1-\beta_2)(1-\beta_1/\beta_2)}\sum_{i=1}^{d}\log\left(\frac{\mathcal{F}_i(t)}{\beta_2^t}\right).$$

Let

$$D_1 = \left(\frac{2L\eta\beta G_T\sqrt{1-\beta_2}}{(1-\beta_1)^2} + \frac{6L\eta^2}{(1-\beta_1)^3}\right)d\log\left(\frac{\mathcal{F}(T)}{\beta_2^T}\right),$$

$$D_2 = \frac{2L^2\eta^2(1+4\omega^2)}{(1-\beta_1)^2(1-\beta_2)(1-\beta_1/\beta_2)}d\log\left(\frac{\mathcal{F}(T)}{\beta_2^T}\right),$$

$$D_3 = \frac{2L\eta\beta G_T(\sqrt{1-\beta_2}+\beta_3^2)}{\beta_2(1-\beta_1)^2(1-\beta_2)(1-\beta_1/\beta_2)}d\log\left(\frac{\mathcal{F}(T)}{\beta_2^T}\right),$$

$$D_4 = \frac{8LG_T\sqrt{d}\eta}{\sqrt{(1-\beta_1)^3(1-\beta_2)(1-\beta_1/\beta_2)}},$$

$$D_5 = \frac{4Ld}{\lambda}\log\left(\frac{T}{\delta}\right),$$

$$\lambda = \frac{2(1-\beta_1)\sqrt{1-\beta_2}}{3\eta G_T\beta}.$$

Finally, given $0 \leq \beta_1 < \beta_2 < 1$, $\eta = C_0\sqrt{1-\beta_2}$, $\gamma \in \left(\frac{2}{\beta}, 1\right)$, and $\beta > 2$, we define $\beta_3 = \max\left\{\frac{1-\beta_2}{\sqrt{1-\beta_2}}, \frac{2-\gamma^2(1+\beta_2)}{\gamma\sqrt{1-\beta_2}}, \frac{|\beta_2-\gamma^2|+1-\gamma^2}{\gamma\sqrt{1-\beta_2}}\right\}$ and $\omega = (\sqrt{1+1/\beta_2}+1)\max\{1, \gamma, 1/\gamma\}$. With these definitions, we can derive that $G^2$ satisfies

$$G^2 = 4L(f(x_1) - f^*) + D_1 + D_2 + D_3 + D_4 + D_5 + 2C_5. \tag{17}$$

Next, we proceed with a proof by mathematical induction. First, we assume that $G_t \leq G, G_T(t) \leq G_T, \forall t \in [t]$. Thus,

$$\|\nabla f(\mathbf{x}_{t+1})\|^2 \leq G^2 + 4L\left(\frac{2}{\beta} - \gamma\right)\sum_{s=1}^{t}\eta_s\left\|\frac{\nabla f(\mathbf{x}_s)}{\sqrt{\mathbf{a}_s}}\right\|^2 \leq G^2.$$

Thus, $G_{t+1} = \max\{G_t, \|\nabla f(\mathbf{x}_{t+1})\|\} \leq G$, which confirms the validity of the initial hypothesis.

Since Lemma E.7 and Lemma E.9 each hold with probability at least 1-$\delta$, they hold simultaneously with probability at least 1-2$\delta$. We have,

$$\|\mathbf{a}_s\|_\infty = \max_{i \in [d]} \sqrt{\beta_2 \mathbf{v}_{s-1,i} + (1-\beta_2)(G_T(s))^2} + \epsilon$$

$$\leq \max_{i \in [d]} \sqrt{(1-\beta_2)\left[\sum_{j=1}^{s-1} \beta_2^{s-j} \mathbf{g}_{j,i}^2 + (G_T(s))^2\right]} + \epsilon$$

$$\leq \sqrt{(1-\beta_2)\sum_{j=1}^{s} \beta_2^{s-j} G_T^2} + \epsilon = G_T\sqrt{1-\beta_2^s} + \epsilon, \quad \forall s \in [T].$$

Then, combining $\eta = C_0\sqrt{1-\beta_2}$, $\epsilon > \epsilon\sqrt{(1-\beta_2^s)(1-\beta_2)}$, for any $s \in [T]$, we can obtain

$$\frac{\eta_s}{\|\mathbf{a}_s\|_\infty} \geq \frac{C_0\sqrt{(1-\beta_2^s)(1-\beta_2)}}{G_T\sqrt{1-\beta_2^s} + \epsilon\sqrt{(1-\beta_2^s)(1-\beta_2)}} \cdot \frac{1}{1-\beta_1^s} \geq \frac{C_0\sqrt{1-\beta_2}}{G_T + \epsilon\sqrt{1-\beta_2}}.$$

Thus,

$$\frac{\|\mathbf{a}_s\|_\infty}{\eta_s} \leq \frac{G_T + \epsilon\sqrt{1-\beta_2}}{C_0\sqrt{1-\beta_2}}.$$

Therefore, letting $\gamma > \frac{2}{\beta}$,

$$L\sum_{s=1}^{T} \frac{\eta_s}{\|\mathbf{a}_s\|_\infty}\|\nabla f(\mathbf{x}_s)\|^2 \leq L\sum_{s=1}^{T} \eta_s \left\|\frac{\nabla f(\mathbf{x}_s)}{\sqrt{\mathbf{a}_s}}\right\|^2 \leq \frac{G^2 - \|\nabla f(\mathbf{x}_{T+1})\|^2}{4(\gamma - 2/\beta)} \leq \frac{G^2}{4(\gamma - 2/\beta)}.$$

$$\frac{1}{T}\sum_{s=1}^{T} \|\nabla f(\mathbf{x}_s)\|^2 \leq \frac{\sqrt{2}G^2}{4(\gamma - 2/\beta)TLC_0}\left(\sqrt{\frac{\|\sigma\|^2 + G^2}{1-\beta_2}} + \epsilon\right)\sqrt{\log\left(\frac{eT}{\delta}\right)}$$

$$= \frac{\sqrt{2}G^2}{4(\gamma - 2/\beta)LC_0}\left(\sqrt{\frac{\|\sigma\|^2 + G^2}{T}} + \frac{\epsilon}{T}\right)\sqrt{\log\left(\frac{eT}{\delta}\right)} \overset{(\circ)}{=} \tilde{\mathcal{O}}(T^{-1/2}),$$

where $(\circ)$ is due to Lemma E.11, we have $G^2 \sim \mathcal{O}(\text{poly}(\log(T)))$.

$\square$

## G    Experimental Details

### G.1    Pretraining on CIFAR10

The ViT-27M model [43] undergoes pretraining on the CIFAR-10 dataset with comprehensive hyperparameter specifications provided in Table 3. Our training protocol employs a base learning rate of $1.5 \times 10^{-4}$ coupled with a cosine decay schedule over 200 training epochs. The optimization configuration utilizes AdamW parameters with weight decay coefficient $\lambda = 0.05$, numerical stability constants $\epsilon = 1 \times 10^{-8}$, and momentum terms $\beta_1 = 0.9$, $\beta_2 = 0.95$. To maintain training stability while processing large-scale inputs, we implement a batch size of 4096 through gradient accumulation with a step size of 20, ensuring memory efficiency without compromising convergence dynamics.

Table 3: Hyperparameters used for training ViT

| # Params | $\beta_1$ | $\beta_2$ | Learning Rate | Weight Decay | Batch Size | Warmup Epochs |
|---|---|---|---|---|---|---|
| 27.6M | 0.9 | 0.95 | 1.5e-4 | 0.05 | 4096 | 20 |

### G.2    Pretraining on WikiText-103

The LLaMA2-71M model[46] and Qwen2.5-150M model[47] were pre-trained on the Wikitext-103 dataset. Identical learning rates and scheduling protocols were systematically implemented across all optimizers during the training process. Comprehensive experimental specifications are tabulated in Table 4 and Table 5.

Table 4: Hyperparameters used for training LLaMA2-71M on WikiText-103

| | Adam-Type | Lion-Type | Muon-Type |
|---|---|---|---|
| Model Size | | 71M | |
| Hidden Size | | 512 | |
| Head | | 8 | |
| Depth | | 12 | |
| Training Steps | | 2034 | |
| Warmup Steps | | 203 | |
| Maximum Length | | 1024 | |
| Batch Size | | 480 | |
| Learning Rate | | 3e-4 | |
| Warmup Scheduling | | linear from 3e-5 | |
| Learning Rate Scheduling | | cosine to 10% | |
| Numerical precision | | bfloat16 | |
| Weight Decay | | 0.01 | |
| $\beta_1$ | 0.9 | 0.9 | 0.95 |
| $\beta_2$ | 0.999 | 0.99 | 0.95 |
| Momentum | ✗ | ✗ | 0.95 |

Table 5: Hyperparameters used for training Qwen2.5-150M on WikiText-103

| | Adam-Type | Muon-Type |
|---|---|---|
| Model Size | | 150M |
| Hidden Size | | 640 |
| Head | | 10 |
| Depth | | 12 |
| Training Steps | | 1525 |
| Warmup Steps | | 154 |
| Maximum Length | | 1024 |
| Batch Size | | 160 |
| Learning Rate | | 1e-3 |
| Warmup Scheduling | | linear from 6e-5 |
| Learning Rate Scheduling | | cosine to 10% |
| Numerical precision | | bfloat16 |
| Weight Decay | | 0.01 |
| $\beta_1$ | 0.9 | 0.95 |
| $\beta_2$ | 0.95 | 0.95 |
| Momentum | ✗ | 0.95 |

### G.3 Fine-Tuning on GLUE

We fine-tune the pre-trained RoBERTa-Base model[48] on the GLUE benchmark using the Hugging Face implementation[23]. For all tasks except QQP, we employ a batch size of 32, while QQP uses a larger batch size of 128 due to its dataset characteristics. The model is trained uniformly for 3 epochs across all tasks with a maximum sequence length of 512. For each task, we perform a grid search over learning rates. For most optimizers, the learning rate range is $\{1e\text{-}5, 2e\text{-}5, 3e\text{-}5, 4e\text{-}5, 5e\text{-}5\}$ and the weight decay is set to 0.01. For Lion-type optimizers, the learning rate range is $\{1e\text{-}6, 2e\text{-}6, 3e\text{-}6, 4e\text{-}6, 5e\text{-}6\}$ and the weight decay is set to 0.1. The complete hyperparameter configurations are summarized in Table 6. For MGUP-AdamW, AdamW, Adam-mini, and C-AdamW, we use $\beta_1 = 0.9$ and $\beta_2 = 0.999$. For LDAdam and Galore, we use $\beta_1 = 0.9$ and $\beta_2 = 0.99$. For Lion, C-Lion, and MGUP-Lion, we use $\beta_1 = 0.95$ and $\beta_2 = 0.98$.

---

[2]https://huggingface.co/transformers/model_doc/roberta.html
[3]https://huggingface.co/datasets/nyu-mll/glue

Table 6: Hyperparameters used for fine-tuning on GLUE.

| Hyperparameter | MRPC | STS-B | CoLA | RTE | SST-2 | QNLI | QQP |
|---|---|---|---|---|---|---|---|
| Batch Size | 32 | 32 | 32 | 32 | 32 | 32 | 128 |
| Weight Decay (Most) | | | | 0.01 | | | |
| Weight Decay (Lion-type) | | | | 0.1 | | | |
| Epochs | | | | 3 | | | |
| Max Seq Len | | | | 512 | | | |

## G.4 Fine-Tuing on GSM8K

We fine-tune the pre-trained LLaMA2-7B model [46] using the llm-foundry codebase[4] with evaluation via standardized lm-evaluation-harness[5] on the GSM8K benchmark with the Hugging Face implementation[6]. The fine-tuning process employs consistent hyperparameters across all optimizers, including MGUP-AdamW, Adam-8bit, AdamW, and C-AdamW. Specifically, we train for 3 epochs with a total of 702 training steps, including 20 warm-up steps. The batch size is set to 32, and the maximum sequence length is 512. We use a learning rate of 5e-5 and optimizer parameters $\beta_1 = 0.9$ and $\beta_2 = 0.999$. The complete hyperparameter configurations are summarized in Table 7.

Table 7: Hyperparameter configurations for fine-tuning LLaMA2-7B on GSM8K.

| Hyperparameter | Value |
|---|---|
| Epochs | 3 |
| Training Steps | 702 |
| Warm-up Steps | 20 |
| Batch Size | 32 |
| Maximum Length | 512 |
| Learning Rate | 5e-5 |
| $\beta_1$ | 0.9 |
| $\beta_2$ | 0.999 |

# H  Other MGUP-type Algorithms

---
**Algorithm 3 MGUP-Lion**
---

**Input:** Learning rate $\eta_t > 0$, initial parameters $\mathbf{x}_0 \in \mathbb{R}^d$, loss function $f(\mathbf{x})$, momentum factors $\beta_1, \beta_2 \in [0, 1)$, weight decay coefficient $\lambda$, stability term $\epsilon > 0$, ratio $\tau \in (0, 1)$.
**for** $t = 1$ **to** $T$ **do**
  Compute the stochastic gradient $\mathbf{g}_t = \nabla f(\mathbf{x}_t; \xi_t)$
  $\mathbf{u}_t = \text{sign}(\beta_1 \mathbf{m}_{t-1} + (1 - \beta_1)\mathbf{g}_t)$
  $\mathbf{m}_t = \beta_2 \mathbf{m}_{t-1} + (1 - \beta_2)\mathbf{g}_t$
  $\phi_t = \mathbf{MGUP}(\mathbf{u}_t \odot \mathbf{g}_t)$
  $\mathbf{x}_t = (1 - \eta_t \lambda) \odot \mathbf{x}_t$
  $\mathbf{x}_{t+1} = \mathbf{x}_t - \eta_t \phi_t \odot \mathbf{u}_t$
**end for**

---

[4] https://github.com/hiyouga/LLaMA-Factory
[5] https://github.com/EleutherAI/lm-evaluation-harness
[6] https://huggingface.co/datasets/openai/gsm8k

**Algorithm 4 MGUP-Muon**

---

**Input:** Learning rate $\eta_t > 0$, initial parameters $\mathbf{X}_0 \in \mathbb{R}^{m \times n}$, loss function $f(\mathbf{X})$, momentum factors $\beta \in [0, 1)$, weight decay coefficient $\lambda$, ratio $\tau \in (0, 1)$.
**for** $t = 1$ **to** $T$ **do**
    Compute the stochastic gradient $\mathbf{G}_t = \nabla f(\mathbf{X}_t; \xi_t)$
    $\mathbf{M}_t = \beta \mathbf{M}_{t-1} + \mathbf{G}_t$
    $\phi_t = \mathbf{MGUP}(\mathbf{M}_t \odot \mathbf{G}_t)$
    $\mathbf{U}_t = \text{Newton-Schulz}(\mathbf{M}_t)$
    $\mathbf{X}_t = (1 - \eta_t \lambda) \odot \mathbf{X}_t$
    $\mathbf{X}_{t+1} = \mathbf{X}_t - \eta_t \phi_t \odot \mathbf{U}_t$
**end for**

---

# I More Results

As shown in Figure 7, which depicts the training curves under a learning rate of 5e-5, MGUP-AdamW achieves lower training loss per epoch, outperforming baseline optimizers.

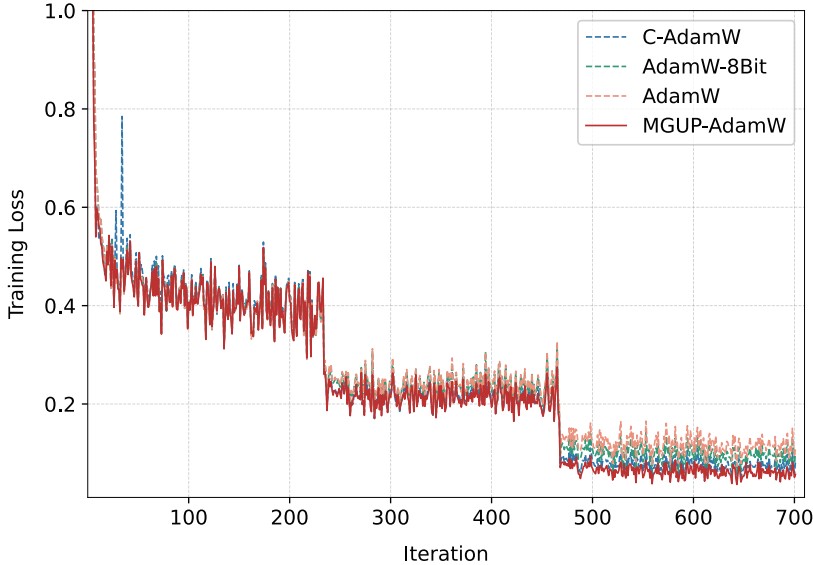

Figure 7: Training curves of LLaMA2-7B on GSM-8K.

## I.1 Memory cost

Table 8 reports memory for fine-tuning LLaMA-7B on two NVIDIA V100 32GB GPUs. We use a micro batch size of 1 for GSM8K fine-tuning. Our algorithm introduces transient elevation in peak memory overhead while maintaining unaltered static memory allocation throughout the computational process.

Table 8: Memory for FineTuing LLaMA2-7B on GSM8K

|  | Adam-8bit | AdamW | C-AdamW | MGUP-AdamW |
|---|---|---|---|---|
| Peak Reserved Memory | 25.38GB | 32.09GB | 32.98GB | 33.82 GB |

## I.2 Time cost

Table 9 documents the runtime measurements for both fine-tuning and pre-training processes conducted primarily on a single NVIDIA RTX 4090 24GB GPU, with one exceptional case: LLaMA2-7B fine-tuning utilizing two NVIDIA V100-32GB GPUs. For GSM8K fine-tuning experiments, we maintained a micro-batch size of 1 throughout the process. In pre-training configurations, gradient accumulation strategy was implemented to optimize memory utilization.

**Remark I.1.** *In all experiments, we documented the duration from initiation to completion rather than the algorithm's execution time. Discrepancies may arise due to variations in GPU operational states.*

Table 9: Runtime for FineTuing(PT) and PreTraining(PT) tasks

| Model | Task | AdamW | C-AdamW | MGUP-AdamW |
|---|---|---|---|---|
| ViT-28M | CIFAR10(PT) | 1h5m | 1h 6m | 1h 6m |
| LLaMA2-71M | WikiText-103(PT) | 5h 36m | 5h 37m | 5h 37m |
| Qwen2.5-150M | WikiText-103(PT) | 1h 54m | 1h55m | 1h56m |
| RoBERTa-Base | QQP(FT) | 25m | 30m | 34m |
| LLaMA2-7B | GSM8K(FT) | 15h 53m | 21h 37m | 22h 58m |

