# OpenReview forum: "MGUP: A Momentum-Gradient Alignment Update Policy for Stochastic Optimization"
_NeurIPS.cc/2025/Conference — NeurIPS 2025 spotlight_

### Official Review · Reviewer_go7i · 2025-06-21

**Clarity:** 3
**Significance:** 2
**Originality:** 2
**Rating:** 4
**Confidence:** 3

**Summary:**

This paper proposes a new monentum-gradient alignment policy for stochastic optimization.
It provides theoretical guarantees of the proposed methods, and evaluates the performance on a few benchmark NN training tasks.

**Questions:**

1. In equation (3), the scaling factor is 1/tau if signs are aligned and tau if signs are not aligned. What's the intuition here to choose these scaling factors as the reverse of each other, instead of choosing in a more empirical/data-driven way?

2. In Assumption 4.2, it is assumed that function f is L-smooth. Is it possible to show a convergence result for with relaxed (non-smooth) conditions? For example for relu activation.

3. In Theorem 4.2, it is better to give a specific probability upper bound (at least mention it in a remark) instead of just stating it is 'with high probability'. It can help the reader understand how the probability bound depends on the parameters.

**Ethical Concerns:**

["NO or VERY MINOR ethics concerns only"]

**Final Justification:**

Thank you for the response. I will keep my rating.

**Limitations:**

See weakness and questions.

**Quality:**

3

**Strengths And Weaknesses:**

Strength: This paper have a clean introduction of the main idea -- I enjoy reading the paper a lot. It has sound theoretical guarantees for the proposed algorithm, and demonstrates the efficiency with extensive numerical experiments.

Weakness: As mentioned by the authors, the proposed algorithm combines the idea of a few existing methods. The contribution is a bit incremental.

---

> ### Author Rebuttal · Authors · 2025-07-30
>
> We thank the reviewer for his/her thoughtful comments and suggestions.
>
> ---
>
> **Q1.** In equation (3), the scaling factor is $1/\tau$ if signs are aligned and tau if signs are not aligned. What's the intuition here to choose these scaling factors as the reverse of each other, instead of choosing in a more empirical/data-driven way?
>
> **A1.** We will now elaborate on the intuition for choosing these scale factors.
>
> (1) MGUP increases the average update magnitude. In MGUP, a proportion of $\tau$ parameters execute updates with an increased learning rate of $1/\tau \cdot \text{lr}$, while a proportion of $1-\tau$ parameters execute updates with a reduced learning rate of $\tau \cdot \text{lr}$. This means that when $0 < \tau < 1$, the average learning rate is $\tau \cdot \text{lr} \cdot (1/\tau) + (1-\tau) \cdot \text{lr} \cdot \tau = \text{lr} \cdot (1+\tau-\tau^2) > \text{lr}$. If the step size for each parameter is roughly equivalent, for example, when Adam's updates are approximately $\text{sign}(g_t)$ in the initial phase, this implies that the overall magnitude of MGUP-Adam's updates is increased by a factor of $(1+\tau-\tau^2)$ compared to Adam. This is a potential underlying intuition behind MGUP-Adam's acceleration: MGUP can accelerate updates by an approximate factor of $(1+\tau-\tau^2)$.
>
> (2) With the goal of reducing hyperparameter tuning, we recognize that introducing two separate hyperparameters for adjustment $\alpha$ for increasing the step size and $\gamma$ for decreasing it—could add to the burden of practical application. In our experiments, however, we found that simply choosing a $\tau$ in the range of 0.3 to 0.7 is sufficient to achieve good results.
>
> ---
>
> **Q2.** In Assumption 4.2, it is assumed that function f is L-smooth. Is it possible to show a convergence result for with relaxed (non-smooth) conditions? For example for relu activation.
>
> **A2.** Extending our convergence results to non-smooth scenarios is a highly valuable but also challenging theoretical task. We are confident that this approach can be extended to our method, though other issues may arise in the process.
>
> ---
>
> **Q3.** In Theorem 4.2, it is better to give a specific probability upper bound (at least mention it in a remark) instead of just stating it is 'with high probability'. It can help the reader understand how the probability bound depends on the parameters.
>
> **A3.** Thank you for your suggestion, which helps to improve the clarity of our theoretical results. However, we would like to clarify that we have already stated in the **Theorem 4.2** that the convergence holds with a probability of at least $1−2\delta$, where $\delta\in(0,1/2)$ is a user-selected confidence parameter. We use the notation $\tilde{O}$ to hide polylogarithmic factors. For the specific formula, please refer to the result above line 626 in the appendix.

---

> > ### Comment · Reviewer_go7i · 2025-08-05
> >
> > Thank you for the detailed response. I will keep my rating.

---

### Official Review · Reviewer_kuKg · 2025-07-02

**Clarity:** 3
**Significance:** 2
**Originality:** 3
**Rating:** 4
**Confidence:** 3

**Summary:**

This paper proposed a new plug-in method, MGUP, for improving momentum updates. This new strategy can be viewed as a smoother version of the cautious optimizer. Theoretical convergence guarantees and empirical results are provided to show the effectiveness of the algorithm.

**Questions:**

1. I am concerned with the computation burden of MGUP, since it requires sorting during the update, which basically has $O(d\log d)$ complexity.

2. MGUP-Muon seems to have a different implementation compared to MGUP-Lion and MGUP-Adam, since it does MGUP to compute $\phi_t$ before computing the update $u_t$. Could you explain this difference?

**Ethical Concerns:**

["NO or VERY MINOR ethics concerns only"]

**Final Justification:**

I am generally satisfied with the rebuttal, thus raise my score to 4.

**Limitations:**

Yes.

**Paper Formatting Concerns:**

No.

**Quality:**

3

**Strengths And Weaknesses:**

**Strengths**:
1. The paper explains the idea well, and to smooth the cautious optimizer sounds to be a reasonable idea.

2. The paper provides theoretical proof for the convergence of the algorithm and illustrates the importance of employing a nonzero update even when the gradient and momentum are not well aligned.

**Weakness**:
1. The theoretical illustration for using nonzero $\gamma$ doesn't seem to be convincing enough. We basically need a lower bound, or maybe even a simple concrete case study for why setting $\gamma=0$ can lead to unstable performance of the algorithm. Also, I think it would be better to also talk about work related to the convergence theory.

2. The empirical improvement doesn't seem to be significant enough, especially for the language model tasks. The improvement upon the baseline is not that significant while the experiment scale is not large, at least insufficient to justify the effectiveness of the approach. This is important because we need strong empirical evidence to support MGUP, since the motivation of the MGUP idea isn't that convincing theoretically. I would be happy to raise the score if you could explain the benefits of doing MGUP compared to the cautious optimizer and the original Adam.

---

> ### Author Rebuttal · Authors · 2025-07-30
>
> We sincerely thank the reviewer for his/her constructive feedback.
>
> ---
>
> **Q1.** The theoretical illustration for using nonzero doesn't seem to be convincing enough. We basically need a lower bound, or maybe even a simple concrete case study for why setting $\gamma=0$ can lead to unstable performance of the algorithm. Also, I think it would be better to also talk about work related to the convergence theory.
>
> **A1.** Thank you for raising this important point.
> To show non-convergence when $\gamma=0$, we'll use a counterexample from the **literature [1]** section 3.2.
>
> Consider $f(x) = \sum_{i=0}^{n-1} f_{i}(x)$ for $x\in \mathbb{R}$, and then we define $f_i(x)$ as
>
> $\begin{align*}
> f_i(x) &=
> \begin{cases}
> nx, & x\geq-1 ;\\
> \frac{n}{2}(x+2)^2-\frac{3n}{2}, & x<-1
> \end{cases}
> & \text{for } i=0,\end{align*}
> $
>
>
> $\begin{align*}
> f_i(x) &=
> \begin{cases}
> -x, & x\geq-1 ;\\
> -\frac{1}{2}(x+2)^2+\frac{3}{2}, & x<-1
> \end{cases}
> & \text{for } i>0.
> \end{align*}$
>
> so
>
> $
> f(x)=
> \begin{cases}
> x, & x\geq-1 ;\\
> \frac{1}{2}(x+2)^2-\frac{3}{2}, & x<-1
> \end{cases}
> $
>
> The optimum is $x^*=-2$.
> We initialize at $x = -0.5$. Let's analyze what happens when $x\ge-1$ for Cautious-Adam.
>
> 1.Dynamic Analysis of Momentum ($m_t$)
> In this counterexample, momentum ($m_t$) follows a "pulse-decay" model: rare, large positive gradients push $m_t$ high, followed by frequent negative gradients causing it to decay, inevitably crossing zero. This oscillation leads to $P(m_t>0)\approx 0.5$.
>
> 2.Decomposing the Behavior of Cautious-Adam in This Scenario
> Based on the assumption of momentum oscillation, we analyze the four core behaviors of Cautious-Adam:
>
> (1)Case A: A positive gradient $g_t = n$ is sampled (low probability of $1/n$)
>
> (1.1)A.1 ($m_t > 0$, probability $\approx 0.5$):The directions align, and a correct update is performed. (Total probability $\approx 0.5/n$)
>
> (1.2)A.2 ($m_t < 0$, probability $\approx 0.5$): The directions are opposed, and the update is skipped. (Total probability $\approx 0.5/n$)
>
> (2)Case B: A negative gradient $g_t = -1$ is sampled (high probability of $(n-1)/n \approx 1$)
>
> (2.1)B.1 ($m_t > 0$, probability $\approx 0.5$):The directions are opposed, and the update is skipped. (Total probability $\approx 0.5$)
>
> (2.2)B.2 ($m_t < 0$, probability $\approx 0.5$):The directions align, and an incorrect update (moving away from the optimum) is performed. (Total probability $\approx 0.5$)
>
> 3.Conclusion
>
> The behavior of Cautious-Adam is dominated by high-probability events. Ultimately, the dynamics of the algorithm are as follows:
>
> (1)With a probability of $\approx 50\%$, the update is skipped.
>
> (2)With a probability of $\approx 50\%$, the algorithm updates in the wrong direction.
>
> (3)Only with an extremely low probability of $\approx 0.5/n$ does the algorithm update in the correct direction.
>
> Therefore, Cautious-Adam not only stagnates but also diverges from the optimal point with high probability, demonstrating its non-convergence. In contrast, MGUP avoids this by setting $\gamma>0$, ensuring continued progress towards the optimum. Our empirical study with n=10 shows Cautious-Adam diverging from the optimal point of -2, while MGUP-Adam converges to -2, consistent with our analysis. (Due to character limits, you can view partial results in our response to the second Reviewer Vt5g.)
>
> Then, regarding your suggestion :**'We basically need a lower bound'** .
>
> Our theoretical results indicate that $\gamma$ only needs to be a strictly positive constant, thus making this infimum unnecessary. We believe that this concrete counterexample more clearly reveals the inherent instability of setting $\gamma=0$ than a generalized theoretical lower bound would.
>
> Regarding your suggestion: **'Also, I think it would be better to also talk about work related to the convergence theory’**.
>
> Thank you for your suggestion. We'll summarize the development of Adam's convergence theory and additionally cite articles [1], [2], [3], [4], [5], and others. Unlike existing work that commonly relies on specific treatments of $\beta_2$ (e.g., decay), our Theorem 4.1 establishes convergence by decaying the first moment parameter $\beta_1$, which constitutes a unique aspect of our proof.
>
> [1] Zhang, Y., Chen, C., Shi, N., Sun, R., & Luo, Z. (2022). Adam Can Converge Without Any Modification on Update Rules.
>
> [2] Zou, F., Shen, L., Jie, Z., Zhang, W., & Liu, W. (2019). A sufficient condition for convergences of adam and rmsprop.
>
> [3] Wang, B., Fu, J., Zhang, H., Zheng, N., & Chen, W. (2023). Closing the Gap Between the Upper Bound and the Lower Bound of Adam's Iteration Complexity.
>
> [4] Chen, X., Liu, S., Sun, R., & Hong, M. (2018). On the Convergence of A Class of Adam-Type Algorithms for Non-Convex Optimization.
>
> [5] He, M., Liang, Y., Liu, J., & Xu, D. (2023). Convergence of adam for non-convex objectives: Relaxed hyperparameters and non-ergodic case.
>
> ---
>
> **Q2.** The empirical improvement doesn't seem to be significant enough, especially for the language model tasks. The improvement upon the baseline is not that significant while the experiment scale is not large, at least insufficient to justify the effectiveness of the approach. This is important because we need strong empirical evidence to support MGUP, since the motivation of the MGUP idea isn't that convincing theoretically. I would be happy to raise the score if you could explain the benefits of doing MGUP compared to the cautious optimizer and the original Adam.
>
> **A2.** We emphasize MGUP's empirical superiority through its efficiency, robustness, and extensive validation. MGUP significantly enhances robustness and stability, reducing learning rate sensitivity and maintaining training stability, outperforming methods like C-Lion. Our findings are broadly and diversely validated across diverse model architectures (ViT, LLaMA, MoE), scales (27M-7B parameters), and tasks (pre-training, NLU, mathematical reasoning fine-tuning). This multifaceted improvement and consistent validation offer compelling evidence for our method.
>
> Next, we will focus on elaborating the core advantages of MGUP over the Cautious Optimizer and Adam in response to your follow-up questions.
>
> **1.MGUP-Adam vs. Cautious Adam.**
>
> (1) Cautious Adam lacks theoretical results for stochastic optimization, whereas we ensure global convergence by employing a safeguard parameter.
>
> (2) The inability of cautious optimizers' methods to guarantee convergence under extreme conditions accounts for their lack of theoretical results, as illustrated by the counterexample in **A1**.
>
> (3) MGUP-Adam sets a threshold $\tau$, to ensure that the number of parameters being updated does not reach extreme cases. For instance, an extreme case could involve updates to merely 1% of the parameters.
>
> **2.MGUP-Adam vs. Adam.**
>
> (1)MGUP's mechanism uses a greedy strategy (due to sorting). Greedy approaches play a crucial accelerating role in heuristic algorithms. When the stochastic gradient and the current update have the same sign, their product is positive, making them more likely to be selected for a large step-size update. Conversely, when they have opposite signs, their product is negative, making them more likely to be selected for a small step-size update. Large step-size updates drive acceleration, while small step-size updates ensure convergence.
>
> (2)MGUP increases the average update magnitude. In MGUP, a proportion of $\tau$ parameters execute updates with an increased learning rate of $1/\tau \cdot \text{lr}$, while a proportion of $1-\tau$ parameters execute updates with a reduced learning rate of $\tau \cdot \text{lr}$. This means that when $0 < \tau < 1$, the average learning rate is $\tau \cdot \text{lr} \cdot (1/\tau) + (1-\tau) \cdot \text{lr} \cdot \tau = \text{lr} \cdot (1+\tau-\tau^2) > \text{lr}$. If the step size for each parameter is roughly equivalent, for example, when Adam's updates are approximately $\text{sign}(g_t)$ in the initial phase, this implies that the overall magnitude of MGUP-Adam's updates is increased by a factor of $(1+\tau-\tau^2)$ compared to Adam. This is a potential underlying intuition behind MGUP-Adam's acceleration: MGUP can accelerate updates by an approximate factor of $(1+\tau-\tau^2)$.
>
> ---
>
> **Q3.** I am concerned with the computation burden of MGUP, since it requires sorting during the update, which basically has $O(d\log d)$ complexity.
>
> **A3.** We understand your concern. However:
>
> (1) The complexity is approximately linear, and the sorting for TopK is O(dlogK). We can set K to an acceptably small value, in which case the complexity remains nearly unchanged.
>
> (2) Our experiments in Qwen2.5 show that despite 4 random trials with the same iterations, sorting has no significant impact on total execution time. Crucially, our approach yields a superior objective function result. Our setup utilizes a single NVIDIA RTX 4090 GPU.
>
> |            | average time   |
> | ---------- | ----------- |
> | MGUP-AdamW | 4h20m49s |
> | Cautious-AdamW    | 4h20m40s |
> | AdamW      | 4h20m03s |
>
> ---
>
> **Q4.** MGUP-Muon seems to have a different implementation compared to MGUP-Lion and MGUP-Adam, since it does MGUP to compute  before computing the update . Could you explain this difference?
>
> **A4.** All three of our algorithms adhere to the same framework: parameter selection is based on the dot product of the mini-batch gradient and the EMA  of the gradient. In Adam and Lion, the element-wise direction of the momentum and the update vector are consistent. However, this is not the case in Muon, because Muon introduces a gradient whitening step, $(G G^\top)^{-1/2}$. Specifically, the process $M_t \stackrel{SVD}{=}V_1SV_2^\top \Rightarrow U_t = V_1V_2^\top$ is likely to change the element-wise direction of the update vector, which is inconsistent with our momentum-gradient alignment concept. Therefore, we use $s=M_t \odot G_t$ instead of $s=U_t \odot G_t$.

---

> > ### Comment · Reviewer_kuKg · 2025-08-06
> >
> > Thanks for the reply. I think the justification example is intuitive and clearly illustrates why we need $ \gamma > 0 $$. I would suggest the authors incorporate this example in the revised version of the paper. In general, I am satisfied with the results and will raise my score to 4.

---

### Official Review · Reviewer_Vt5g · 2025-07-03

**Clarity:** 2
**Significance:** 3
**Originality:** 3
**Rating:** 4
**Confidence:** 4

**Summary:**

This paper proposes an adaptive mechanism that adjusts the step size based on the alignment of the momentum and the gradient. The paper provides a convergence analysis of the proposed method, MGUP, with Adam under mild conditions. The theoretical analysis also shows the necessity of keeping all step sizes non-zero for convergence. Numerical experiments demonstrate that the proposed method outperforms its baseline algorithms without MGUP.

**Questions:**

Please address the weakness.

It is unclear to me how the theoretical analysis reflects the unique properties of the proposed algorithm.

**Ethical Concerns:**

["NO or VERY MINOR ethics concerns only"]

**Final Justification:**

The response of the authors addressed my initial concern. Therefore, I would raise my socre.

**Quality:**

3

**Strengths And Weaknesses:**

Strength:
1. The paper provides a rigorous convergence analysis for the Adam-type algorithm with different step sizes.
2. Numerical results on ViT training and LLaMa fine-tuning show that the proposed algorithm outperforms its baselines without MGUP.

Weakness:
1. On the necessity of using $\gamma >0$. Remark 4.1 and 4.3 only show that under the current analysis, $\gamma>0$ is a necessary condition. However, it is not sufficient to show whether $\gamma>0$ is necessary due to the analysis or the algorithm design. If the authors can provide an example that demonstrates that choosing $\gamma=0$ leads to non-convergence, the claim would be significantly strengthened.
2. Regarding remark 4.2, the authors failed to provide further discussion on the existence of such an $M_T < \infty$. Since the convergence results only show that with high probability, or in expectation, the gradient norm is diminishing. Proving that the per-sample gradient is bounded along the trajectory is also a non-trivial extension.
3. The experiments failed to report the standard deviation. For LLaMA2 and Qwen2.5 training tasks, the accuracy and loss are relatively close to each other. It is challenging to determine whether the performance improvement is due to the random seed or if there is a statistically significant difference.
4. The analysis uses constant $\alpha$ and $\gamma$, independent of $s=u_t g_t$. It does not explain why MGUP has better performance. Also, it does not provide insight on how to choose $\phi_i$ to accelerate the convergence of MGUP.

Minor point:
4. Why choose $\alpha = 1/\tau$ and $\gamma = \tau$? Is there any specific reason? It is also a bit confusing since $\gamma, \alpha$ do not appear in Algorithms 1 and 2.

---

> ### Author Rebuttal · Authors · 2025-07-30
>
> Thank you for the reviewer’s valuable comments. Below we reply point‑by‑point, using the reviewer’s wording for clarity.
>
> ---
>
> **Q1.** On the necessity of using $\gamma >0$. Remark 4.1 and 4.3 only show that under the current analysis, $\gamma>0$ is a necessary condition. However, it is not sufficient to show whether $\gamma>0$ is necessary due to the analysis or the algorithm design. If the authors can provide an example that demonstrates that choosing $\gamma=0$ leads to non-convergence, the claim would be significantly strengthened.
>
> **A1.** First, we provide below a counterexample, based on Section 3.2 of (Zhang, Y., Chen, C., Shi, N., Sun, R., & Luo, Z. Adam Can Converge Without Any Modification on Update Rules. NeurIPS 2022), which demonstrates that setting $\gamma=0$ can lead to non-convergence in practice.
>
> Consider $f(x) = \sum_{i=0}^{n-1} f_{i}(x)$ for $x\in \mathbb{R}$, and then we define $f_i(x)$ as
>
> $\begin{align*}
> f_i(x) &=
> \begin{cases}
> nx, & x\geq-1 ;\\
> \frac{n}{2}(x+2)^2-\frac{3n}{2}, & x<-1
> \end{cases}
> & \text{for } i=0,\end{align*}
> $
>
>
> $\begin{align*}
> f_i(x) &=
> \begin{cases}
> -x, & x\geq-1 ;\\
> -\frac{1}{2}(x+2)^2+\frac{3}{2}, & x<-1
> \end{cases}
> & \text{for } i>0.
> \end{align*}$
>
> so
>
> $
> f(x)=
> \begin{cases}
> x, & x\geq-1 ;\\
> \frac{1}{2}(x+2)^2-\frac{3}{2}, & x<-1
> \end{cases}
> $
>
>
> The optimum is $x^*=-2$.
>
> We construct a counterexample to demonstrate the non-convergence of Cautious-Adam in a specific stochastic environment. We initialize at $x = -0.5$. Let's analyze what happens when $x\ge-1$.
>
> 1.Dynamic Analysis of Momentum ($m_t$)
>
> In this counterexample, the behavior of the momentum $m_t$ follows a "pulse-decay" model:
>
> （1）Pulse:A positive gradient $g_t = n$ with an extremely low probability pushes $m_t$ to a large positive value.
>
> （2）Decay: A negative gradient $g_t = -1$ with a high probability causes $m_t$ to continuously decay towards its steady state of -1. This process inevitably leads to $m_t$ crossing zero into the negative region.
>
> This oscillatory cycle results in the momentum spending roughly equal amounts of time on either side of zero. Therefore, we can reasonably approximate that **$P(m_t > 0) \approx 0.5$.**
>
> 2.Decomposing the Behavior of Cautious-Adam in This Scenario
>
> Based on the assumption of momentum oscillation, we analyze the four core behaviors of Cautious-Adam:
>
> （1）Case A: A positive gradient $g_t = n$ is sampled (low probability of $1/n$)
>
> （1.1）A.1 ($m_t > 0$, probability $\approx 0.5$): The directions align, and a correct update is performed. (Total probability $\approx 0.5/n$)
>
> （1.2）A.2 ($m_t < 0$, probability $\approx 0.5$): The directions are opposed, and the update is skipped. (Total probability $\approx 0.5/n$)
>
> （2）Case B: A negative gradient $g_t = -1$ is sampled (high probability of $(n-1)/n \approx 1$)
>
> （2.1）B.1 ($m_t > 0$, probability $\approx 0.5$): The directions are opposed, and the update is skipped. (Total probability $\approx 0.5$)
>
> （2.2）B.2 ($m_t < 0$, probability $\approx 0.5$): The directions align, and an incorrect update (moving away from the optimum) is performed. (Total probability $\approx 0.5$)
>
> 3.Conclusion
>
> The behavior of Cautious-Adam is dominated by high-probability events. Ultimately, the dynamics of the algorithm are as follows:
>
> (1)With a probability of $\approx 50\%$, the update is skipped.
>
> (2)With a probability of $\approx 50\%$, the algorithm updates in the wrong direction.
>
> (3)Only with an extremely low probability of $\approx 0.5/n$ does the algorithm update in the correct direction.
>
> Therefore, not only does Cautious-Adam's convergence stagnate, but it also actively moves away from the optimal point with high probability. This provides strong evidence for its non-convergence. In contrast, MGUP avoids this issue by setting $\gamma > 0$, which ensures that it still progresses in the correct direction in case B.1.
>
> We set n=10 and conducted an empirical study. The data we recorded are shown below (due to character limits, we only display some of the updates, up to a total of 900 steps). As can be seen, Cautious-Adam gradually diverges from the optimal point of -2, whereas MGUP-Adam is able to converge to -2. This is consistent with our analysis.
>
> | Step | Cautious-AdamW x           | MGUP-AdamW x |
> | ---- | ------------------- | ------------ |
> | 0    | -0.5                | -0.5         |
> | 50   | -0.178418443        | -0.710841    |
> | 100  | -0.005119875        | -0.767780244 |
> | 150  | 0.35917407274246216 | -0.48992154  |
> | 200  | 0.583101749420166   | -0.627167225 |
> | …    |                     |              |
> | 700  | 2.395268678665161   | -1.837388873 |
> | 750  | 2.733309268951416   | -1.848099828 |
> | 800  | 2.945746898651123   | -1.914497256 |
> | 850  | 3.069368839263916   | -1.959119558 |
> | 900  | 3.2514820098876953  | -1.992537737 |
>
> ---
>
> **Q2.** Regarding remark 4.2, the authors failed to provide further discussion on the existence of such an $M_T<\infty$. Since the convergence results only show that with high probability, or in expectation, the gradient norm is diminishing. Proving that the per-sample gradient is bounded along the trajectory is also a non-trivial extension.
>
> **A2.** Thank you for pointing out the imprecision. This assumption is mild as $|| \nabla f(x;\xi)|| \le || \nabla f(x;\xi) - \nabla f(x) || + || \nabla f(x)||  < +\infty$ and $f(x)$ is a continous function.
>
> ---
>
> **Q3.** The experiments failed to report the standard deviation. For LLaMA2 and Qwen2.5 training tasks, the accuracy and loss are relatively close to each other. It is challenging to determine whether the performance improvement is due to the random seed or if there is a statistically significant difference.
>
> **A3.** Thank you for your suggestion. We conducted an additional four sets of experiments with different random seeds for AdamW, Cautious-AdamW, and MGUP-AdamW on the Qwen2.5 training task, doubling the number of iterations and calculating their standard deviations. Please refer to the table below. Our setup utilizes a single NVIDIA RTX 4090 GPU.
>
> |            | Avg loss | Std         |
> | ---------- | -------- | ----------- |
> | MGUP-AdamW | 2.748    | 0.005223205 |
> | Cautious-AdamW    | 2.787    | 0.003388492 |
> | AdamW      | 2.778    | 0.006125102 |
>
> ---
>
> **Q4.** The analysis uses constant and , independent of $s=u_t g_t$ . It does not explain why MGUP has better performance. Also, it does not provide insight on how to choose $\phi_i$ to accelerate the convergence of MGUP. Minor point: 4. Why choose $\alpha = 1/\tau$ and $\gamma=\tau$? Is there any specific reason? It is also a bit confusing since $\alpha,\gamma$ do not appear in Algorithms 1 and 2.
>
> **A4.** Thank you for raising these important questions.
>
> 1.We will first address the first question: **"It does not explain why MGUP has better performance. Also, it does not provide insight on how to choose to accelerate the convergence of MGUP."**
>
> (1)**MGUP's mechanism uses a greedy strategy (due to sorting).** Greedy approaches play a crucial accelerating role in heuristic algorithms. When the stochastic gradient and the current update have the same sign, their product is positive, making them more likely to be selected for a large step-size update. Conversely, when they have opposite signs, their product is negative, making them more likely to be selected for a small step-size update. Large step-size updates drive acceleration, while small step-size updates ensure convergence.
>
> (2)**MGUP increases the average update magnitude.** In MGUP, a proportion of $\tau$ parameters execute updates with an increased learning rate of $1/\tau \cdot \text{lr}$, while a proportion of $1-\tau$ parameters execute updates with a reduced learning rate of $\tau \cdot \text{lr}$. This means that when $0 < \tau < 1$, the average learning rate is $\tau \cdot \text{lr} \cdot (1/\tau) + (1-\tau) \cdot \text{lr} \cdot \tau = \text{lr} \cdot (1+\tau-\tau^2) > \text{lr}$. If the step size for each parameter is roughly equivalent, for example, when Adam's updates are approximately $\text{sign}(g_t)$ in the initial phase, this implies that the overall magnitude of MGUP-Adam's updates is increased by a factor of $(1+\tau-\tau^2)$ compared to Adam. This is a potential underlying intuition behind MGUP-Adam's acceleration: MGUP can accelerate updates by an approximate factor of $(1+\tau-\tau^2)$.
>
> 2.Next, we address the second minor point: **"Why choose $\alpha = 1/\tau$ and $\gamma=\tau$? Is there any specific reason? It is also a bit confusing since $\alpha,\gamma$ do not appear in Algorithms 1 and 2."**
>
> $\alpha$ represents the factor for increasing the step size, and $\gamma$ represents the factor for decreasing the step size. In experiments, once the threshold is determined, the choice of $\gamma$ is robust, as shown in **Figure 3(b)** of the paper. Designing $\alpha$ and $\gamma$ to be related to $\tau$ aims to reduce hyperparameter tuning and simplify the design, resulting in the final algorithm having only one hyperparameter.
>
> ---
>
> **Q5.** It is unclear to me how the theoretical analysis reflects the unique properties of the proposed algorithm.
>
> **A5.** Thank you for raising this important question.
>
> (1) MGUP incorporates a protective mechanism where $\gamma>0$ ensures convergence, a feature absent in existing algorithms like Cautious AdamW.
>
> (2) Furthermore, our approach to expected convergence and the underlying proof methodology differ from current convergence proofs, representing an additional contribution. Unlike existing work that commonly relies on specific treatments of $\beta_2$ (e.g., decay), our Theorem 4.1 establishes convergence by decaying the first moment parameter $\beta_1$, which constitutes a unique aspect of our proof.

---

> > ### Comment · Reviewer_Vt5g · 2025-08-04
> >
> > Thank the author for addressing my concerns. The example and std of the experiment really strengthen the paper.
> > I am still not quite satisified with the answer to Q2 but I don't have further comments.

---

> > > ### Author Response · Authors · 2025-08-05
> > >
> > > We have carefully considered your final concern, raised in Q2, regarding our Remark 4.2. We would like to add a little to our previous response. Our initial high-level intuition was that the per-sample function $f(x;\xi)$ is M-Lipschitz continuous. To make this foundation explicit and resolve any ambiguity, we have added a formal statement of this standard assumption to our manuscript. We thank you again for your valuable feedback, which has significantly strengthened the rigor and clarity of our paper.

---

### Official Review · Reviewer_xAKK · 2025-07-03

**Clarity:** 3
**Significance:** 3
**Originality:** 3
**Rating:** 4
**Confidence:** 3

**Summary:**

This paper introduces MGUP, a novel mechanism designed to enhance momentum-based optimizers like AdamW, Lion, and Muon. The core idea is to selectively and differentially adjust learning rates within each layer. Specifically, for a fixed proportion of parameters with the highest momentum-gradient alignment score, MGUP applies an amplified step size. For the remaining parameters, it applies a smaller, nonzero step size. The authors provide theoretical convergence guarantees for a variant, MGUP-AdamW, under standard stochastic non-convex optimization assumptions. Through extensive experiments on tasks including image pretraining, language model pretraining, and fine-tuning, the paper demonstrates that MGUP-enhanced optimizers consistently achieve faster convergence and/or better generalization compared to their original counterparts and other state-of-the-art methods.

**Questions:**

Regarding the hyperparameter τ, have you identified any general heuristics for its selection? For instance, does its optimal value correlate with the task (e.g., pretraining vs. fine-tuning), model size, or the base optimizer being used?

**Ethical Concerns:**

["NO or VERY MINOR ethics concerns only"]

**Limitations:**

I do not see any potential negative societal impact in addition to the development of LLMs themselves.

**Paper Formatting Concerns:**

No concern

**Quality:**

3

**Strengths And Weaknesses:**

Pros:

- The paper is well-motivated by the limitations of existing selective update methods. It identifies a key problem with methods like Cautious Optimizers, which can discard too much gradient information by nullifying updates (setting step size to zero) for misaligned parameters. MGUP's approach of applying a decayed, non-zero step size (γ>0) is a simple yet principled solution that is well-justified by the theoretical analysis.

- The empirical evaluation is thorough and provides strong evidence for MGUP's effectiveness.

- The design of MGUP as a lightweight, nearly plug-and-play module that can be integrated with various optimizers is a major practical advantage. The paper successfully demonstrates this by creating and evaluating MGUP-AdamW, MGUP-Lion, and MGUP-Muon.

Cons:

- MGUP introduces a key hyperparameter, τ, which defines the proportion of parameters to receive amplified updates. While the authors provide a sensitivity analysis (Figure 3b), the choice of τ appears to be data and model-dependent, which could add to the tuning burden for practitioners.

- he convergence guarantees are formally provided for MGUP-AdamW (without weight decay). While the empirical results for MGUP-Lion and MGUP-Muon are strong, the paper notes that their theoretical properties require further study. This limits the theoretical scope of the paper's claims, though the empirical success across different optimizers is a mitigating factor.

---

> ### Author Rebuttal · Authors · 2025-07-30
>
> We sincerely appreciate the reviewer’s thoughtful comments and suggestions.
>
> ---
>
> **Q1.** MGUP introduces a key hyperparameter, $\tau$, which defines the proportion of parameters to receive amplified updates. While the authors provide a sensitivity analysis (Figure 3b), the choice of $\tau$ appears to be data and model-dependent, which could add to the tuning burden for practitioners.
>
> **A1.** We would like to highlight MGUP's robustness to the hyperparameter $\tau$. As depicted in the sensitivity analysis in Figure 3b, simply setting $\tau=0.5$ provides strong performance across all tasks.
>
> ---
>
> **Q2.** The convergence guarantees are formally provided for MGUP-AdamW (without weight decay). While the empirical results for MGUP-Lion and MGUP-Muon are strong, the paper notes that their theoretical properties require further study. This limits the theoretical scope of the paper's claims, though the empirical success across different optimizers is a mitigating factor.
>
> **A2.** Our theoretical analysis focuses on MGUP-Adam, as Adam remains one of the most popular optimizers, making the convergence guarantees for MGUP-Adam meaningful and timely.
>
> We believe the analysis may serve as a foundation for future theoretical investigations into MGUP-Lion and MGUP-Muon.
>
> At the same time, this focus helps preserve the clarity and conciseness of the theoretical exposition in the paper.
>
> ---
>
> **Q3.** Regarding the hyperparameter $\tau$, have you identified any general heuristics for its selection? For instance, does its optimal value correlate with the task (e.g., pretraining vs. fine-tuning), model size, or the base optimizer being used?
>
> **A3.** In some tasks, we observed that MGUP is not highly sensitive to the choice of $\tau$. A fixed value of $\tau=0.5$ generally performs well across tasks.
>
> While the optimal value may vary slightly depending on the specific setting (e.g., task type, model size, or base optimizer), strong performance is typically maintained within the range $\tau \in [0.3,0.7]$, suggesting that extensive tuning is unnecessary in practice.

---

### Note · Authors · 2025-08-12

Dear Area Chair and Reviewers,

Thank you for your insightful feedback. We are grateful for the reviewers' positive assessment of our work and its contributions. We were pleased to see acknowledgement of:

* Reviewer xAKK: The practical advantage of MGUP as a lightweight, plug-and-play module.
* Reviewer Vt5g: The rigorous convergence analysis and the method's proven effectiveness on Vision Transformers and LLaMA.
* Reviewer kuKg: The clarity of the core idea, the soundness of the research direction, and the theoretical proof.
* Reviewer go7i: The paper's clear presentation, supported by strong theory and extensive experiments.

We address the core issues as follows:

1. **On the necessity of $\gamma > 0$** (raised by Reviewer Vt5g, kuKg):  We cite a classic construction from the literature to provide a clear counterexample demonstrating why $\gamma=0$ (as in Cautious Adam) may fail to converge. Both our analysis and experiments indicate that in this scenario, the algorithm is highly prone to updating in incorrect directions or ceasing updates altogether due to momentum oscillation, ultimately leading to divergence. This highlights the critical role of the safeguarding design of $\gamma > 0$ in MGUP.

2. **On empirical improvements and intuition** (raised by Reviewer Vt5g, kuKg, go7i):
* Statistical Significance: We have added multi-seed experiments, demonstrating that MGUP's performance improvement is consistent and statistically significant.
* Source of Performance Gain: The acceleration is twofold. (1) MGUP is a greedy strategy which accelerates parameters with consistent update directions via larger steps while using smaller steps to stabilize inconsistent ones. (2) MGUP increases the average update magnitude by a factor of $(1+\tau-\tau^2)$, providing an inherent drive for acceleration.

In the final version, we will:
* Incorporate the complete counterexample analysis and corresponding experiments for the non-convergence issue when $\gamma=0$.
* Add a literature review on the convergence proofs for the Adam optimizer.
* To address the concern raised by Reviewer Vr5g regarding Q2, we will add the assumption that $f(x;\xi)$ is continuous.

We once again sincerely thank the reviewers and the Area Chair for your diligent work. We have addressed all core concerns raised by the reviewers and are confident in the contributions of this paper.

---

### Decision · Program_Chairs · 2025-09-17

**Decision:**

Accept (spotlight)

**Comment:**

This paper introduces MGUP, a mechanism for intra-layer selective updates which comes with strong theoretical guarantees as well. The proposed method is a plug-and-play module, and the authors combine it with AdamW, Muon, and Lion and demonstrate its effectiveness in various experiments.

Reviewers all agree that the theory is very strong. Even though there are some concerns about the results not being much better than AdamW, I do not think that is a major issue. There was a concern about the work being incremental (reviewer g07i), but I do not think it is a valid reason to reject this paper. It is an interesting method with strong theoretical guarantees. I recommend accepting this paper.